# On the Data Heterogeneity in Adaptive Federated Learning

**Yujia Wang**  *yjw5427@psu.edu*
*College of Information Sciences and Technology*
*Pennsylvania State University*

**Jinghui Chen**  *jzc5917@psu.edu*
*College of Information Sciences and Technology*
*Pennsylvania State University*

**Reviewed on OpenReview:** *https: // openreview. net/ forum? id= hv7iXsiBZE*

## Abstract

Adaptive federated learning, which benefits from the characteristic of both adaptive optimizer and federated training paradigm, has recently gained lots of attention. Despite achieving outstanding performances on tasks with heavy-tail stochastic gradient noise distributions, adaptive federated learning also suffers from the same data heterogeneity issue as standard federated learning: heterogeneous data distribution across the clients can largely deteriorate the convergence of adaptive federated learning. In this paper, we propose a novel adaptive federated learning framework with local gossip averaging to address this issue. Particularly, we introduce a client re-sampling mechanism and peer-to-peer gossip communications between local clients to mitigate the data heterogeneity without requiring additional gradient computation costs. We theoretically prove the fast convergence for our proposed method under non-convex stochastic settings and empirically demonstrate its superior performances over vanilla adaptive federated learning with client sampling. Moreover, we extend our framework to a communication-efficient variant, in which clients are divided into disjoint clusters determined by their connectivity or communication capabilities. We exclusively perform local gossip averaging within these clusters, leading to an enhancement in network communication efficiency for our proposed method.

## 1 Introduction

Federated learning (FL) (McMahan et al., 2017) has gained tons of attraction recently with the development of edge computing and edge devices such as IoT devices and smartphones. It enables clients to collaboratively learn a machine learning model by iteratively synchronizing with the central server without sharing their local private data. Standard SGD-based federated learning methods such as FedAvg (McMahan et al., 2017) work by aggregating the local-updated models via stochastic gradient descent. Recently, as the demand for training large-scale models such as BERT (Devlin et al., 2018), GPT-3 (Brown et al., 2020), and ViT (Dosovitskiy et al., 2021), adaptive optimizations such as Adam (Kingma & Ba, 2014) and AMSGrad (Reddi et al., 2018) show their efficiency compared to stochastic gradient descent (SGD). This led to the development of adaptive federated learning methods such as FedAdam (Reddi et al., 2020) and FedAMS (Wang et al., 2022b) which take the advantage of efficient iterative synchronization and stable adaptive optimization methods.

Despite achieving superior model training performances on tasks with heavy-tail stochastic gradient noise distributions, adaptive federated learning still suffers from the same data heterogeneity issue as standard SGD-based federated learning. Specifically, the *statistical heterogeneity* of data distribution across clients can lead to overfitting issues of local models and degradation of global model convergence for adaptive federated learning. This data heterogeneity issue becomes noticeable, especially in practical cases where clients are not able to participate in each local training iteration due to system heterogeneity such as computational capabilities. Despite that various attempts have been made in solving the data heterogeneity issue for

standard federated learning (Karimireddy et al., 2020b;a; Khanduri et al., 2021), few studies are tackling this issue in adaptive federated learning, especially for partial participation case where only selected clients have participated in each round of the training process.

In this work, we aim to develop a novel federated learning framework, **A**daptive **F**ederated learning with local **G**ossip **A**veraging (AFGA), that addresses the challenges of statistical heterogeneity in the adaptive federated learning setting. AFGA introduces a client re-sampling strategy and peer-to-peer gossip communication between clients during local training steps to reduce the dissimilarity between local models, thus tackling the data heterogeneity issue. Note that AFGA does not incur extra communication between the central server and the sampled clients, and it also does not result in extra local gradient computations. Our contributions can be summarized as follows:

- We propose a novel adaptive federated optimization method, which benefits from both the client re-sampling strategy and decentralized gossip averaging, to mitigate the impact of data heterogeneity in adaptive federated optimization methods.

- We theoretically provide convergence guarantees of our proposed method in the stochastic non-convex settings with data heterogeneity under partial client participation cases. Specifically, we prove that our proposed method can achieve a faster convergence rate than FedAMSGrad (Wang et al., 2022b) in partial participation settings.

- Moreover, we also extend our framework to a communication-efficient variant, CAFGA, where clients are divided into disjoint clusters and the local gossip communications are only performed within the clusters, thus leading to an overall efficient communication network. We demonstrate that CAFGA can achieve comparable performance and final accuracy to AFGA while simultaneously enhancing overall communication efficiency.

- Extensive experiments on several benchmark datasets demonstrate the proposed AFGA and CAFGA achieving outstanding performance with heterogeneous data in low client participation ratios.

## 2 Related Work

**Federated learning.** Federated learning (Konečnỳ et al., 2016) play a critical role in collaboratively training models at edge devices with potential privacy protections. Basic optimization methods for federated learning include SGD-based global optimizer, e.g., FedAvg (McMahan et al., 2017) (a.k.a. Local SGD (Stich, 2018) and its variants (Li et al., 2019a; Yang et al., 2021), adaptive gradient optimization based global optimizer such as FedAdam, FedAdagrad, FedYogi (Reddi et al., 2020), FedAGM (Tong et al., 2020) and FedAMS (Wang et al., 2022b). While these optimization methods for federated learning show their ability on achieving stable results when data are heterogeneously distributed, they rarely study data heterogeneity itself. Recently, several works address the data heterogeneity issue through several aspects. For example, FedProx (Li et al., 2020a) adds a proximate term to align the local model with the global one, and FedDyn (Acar et al., 2021) involves dynamic regularization term for local and global model consistency. FedNova (Wang et al., 2020b) proposes a normalized averaging mechanism that reduces objective inconsistency with heterogeneous data. Moreover, several works study to eliminate the client drift caused by data heterogeneity from the aspect of variance reduction such as Karimireddy et al. (2020b;a); Khanduri et al. (2021); Cutkosky & Orabona (2019). They introduce additional control variables to track and correct the local model shift during local training, but they require extra communication costs for synchronizing these control variables. Besides, FedDC (Gao et al., 2022) involves both dynamic regularization terms and local drift variables for model correction.

Recent studies extend the decentralized training paradigm to federated learning with various adaption. For example, Guo et al. (2021) considered heterogeneous communications for modern communication networks that improve communication efficiency, and hierarchical federated learning algorithms (Liu et al., 2020; Abad et al., 2020; Castiglia et al., 2020) develop frameworks by aggregating client models to edge servers first before synchronizing them to the central server. [1]

---

[1] Due to space limitations, we leave the detailed related work for decentralized learning in Appendix.

# 3 Proposed Method

## 3.1 Preliminaries

In federated learning, we aim to minimize the following objective through $N$ local clients:

$$\min_{\boldsymbol{x} \in \mathbb{R}^d} f(\boldsymbol{x}) := \frac{1}{N}\sum_{i=1}^{N} f_i(\boldsymbol{x}) = \frac{1}{N}\sum_{i=1}^{N} \mathbb{E}_{\mathcal{D}_i}[f_i(\boldsymbol{x};\xi_i)], \tag{1}$$

where $\boldsymbol{x}$ denotes the model parameters, $d$ denotes the dimension of model parameters $\boldsymbol{x}$, $f_i(\boldsymbol{x}) = \mathbb{E}_{\xi_i \sim \mathcal{D}_i} f_i(\boldsymbol{x}, \xi_i)$ is the local loss function corresponding to client $i$ and let $\mathcal{D}_i$ denotes the local data distribution associated with client $i$. In this work, we focus on the non-convex optimization problem with heterogeneous data distributions, i.e., $f_i$ are non-convex and the local data distribution $\mathcal{D}_i$ are non-i.i.d. distributed among clients. FedAvg (McMahan et al., 2017) is a basic optimization algorithm to solve equation 1, with the sequential implementation of local SGD updates and global averaging.

**Adaptive federated optimizations.** Adaptive federated learning is proposed to incorporate adaptive optimization methods (such as Adam (Kingma & Ba, 2014) and AMSGrad (Reddi et al., 2018)) to global optimizer by replacing the global averaging step in FedAvg. Reddi et al. (2020) summarizes several adaptive federated learning algorithms, and Jin et al. (2022) proposed FedDA, a momentum decoupling adaptive optimization method from the perspective of the dynamics of ODEs. FAFED (Wu et al., 2022) also studied adaptive federated learning but in the context of full participation. FedAMSGrad (Tong et al., 2020; Wang et al., 2022b) considers local SGD updates and global AMSGrad (Reddi et al., 2018) update on the central server. Specifically, at global round $r \in [R]$, the server broadcasts the model $\boldsymbol{x}_r$ to selected clients in the set $\mathcal{S}_r$. The selected client $i$ conducts $\mathcal{I}$ steps of local SGD updates with local learning rate $\eta_l$ and obtains the local model $\boldsymbol{x}_{r,\mathcal{I}}^i$. Then for the selected client $i$, it obtains a model difference $\Delta_r^i = \boldsymbol{x}_{r,\mathcal{I}}^i - \boldsymbol{x}_r$ and sends to the server. The server aggregates $\Delta_r^i$ then updates the global model $\boldsymbol{x}_{r+1}$ by taking $\Delta_r$ as a pseudo gradient for calculating momentum $\boldsymbol{m}_r$ and variance $\boldsymbol{v}_r$ for AMSGrad optimizer, and performs one step AMSGrad update with global learning rate $\eta$, i.e.,

$$\boldsymbol{m}_r = \beta_1 \boldsymbol{m}_{r-1} + (1-\beta_1)\Delta_r, \boldsymbol{v}_r = \beta_2 \boldsymbol{v}_{r-1} + (1-\beta_2)\Delta_r^2,$$

$$\widehat{\boldsymbol{v}}_r = \max\{\widehat{\boldsymbol{v}}_{r-1}, \boldsymbol{v}_r\}, \boldsymbol{x}_{r+1} = \boldsymbol{x}_r + \eta\frac{\boldsymbol{m}_r}{\sqrt{\widehat{\boldsymbol{v}}_r + \epsilon}}, \tag{2}$$

the server finally obtains model $\boldsymbol{x}_{r+1}$ by global round $r$. It's worth mentioning that if the set of selected clients $\mathcal{S}_r$ contains all clients, i.e., $|\mathcal{S}_r| = N$, it is known as full participation or without client sampling, and if $|\mathcal{S}_r| = M < N$, we denote it as partial participation or with client sampling.

**Heterogeneous and inconsistency.** Previous works show that FedAvg suffers from convergence degradation when data is non-i.i.d. distributed on local clients (Karimireddy et al., 2020b;a; Wang et al., 2020b). Several works on adaptive federated learning (Reddi et al., 2020; Wang et al., 2022b) empirically demonstrate that the unbalanced data distribution across clients may lead to worse performance, which implies these adaptive federated methods, unfortunately, do not escape from the convergence degradation as well. Theoretically, it has been shown that when the local data are heterogeneously distributed among clients, FedAMS/FedAMSGrad (Wang et al., 2022b) under partial participation setting is proved with convergence rate $\mathcal{O}(\sqrt{\mathcal{I}}/\sqrt{RM})$ w.r.t. global communication rounds $R$, local update iterations $\mathcal{I}$ and the number of participated clients $M$ which has a certain gap between the desired rate $\mathcal{O}(1/\sqrt{R\mathcal{I}M})$. Although several works apply variance reduction techniques to show their ability to reduce the effect of data heterogeneity in federated learning (Karimireddy et al., 2020a;b), they are less compatible with the global adaptive optimizer. This motivates us to develop a new framework for mitigating the model inconsistency caused by data heterogeneity in adaptive federated learning.

## 3.2 AFGA: Adaptive Federated Learning with Local Gossip Averaging

To reduce the inconsistency between local models and achieve a better convergence rate under heterogeneous data in partial participation settings, we proposed a novel Adaptive Federated Learning with Local Gossip

Averaging (AFGA) method with peer-to-peer gossip communication and client re-sampling framework. The peer-to-peer gossip communication implies that clients are able to communicate their local model without help from the server. Suppose there are in total $N$ clients, we study the same objective function as other adaptive federated learning methods with a similar global framework but a different local updating process.

---

**Algorithm 1** AFGA: Adaptive Federated Learning with Local Gossip Averaging

**Input:** initial point $\boldsymbol{x}_1$, local step size $\eta_l$ and global step size $\eta$, optimizer hyperparameter $\beta_1, \beta_2, \epsilon$, doubly stochastic mixing matrix $W$

1: $\boldsymbol{m}_0 \leftarrow 0$, $\boldsymbol{v}_0 \leftarrow 0$
2: **for** $r = 1$ to $R$ **do**
3:     Randomly sample a subset of clients $\mathcal{S}_r$ for collecting local updates in round $r$
4:     Clients Init: clients in $\mathcal{S}_r$ receive $\boldsymbol{x}_r$ from the server and broadcast to all clients with local communications
5:     **for** $t = 0, ..., \mathcal{I} - 1$ **do**
6:         Randomly **re-sample** a subset of clients $\mathcal{S}_{r,t}$ for gradient computation
7:         **for** each client $i \in [N]$ in parallel **do**
8:             **if** $i \in \mathcal{S}_{r,t}$ **then**
9:                 Compute $\boldsymbol{g}_{r,t}^i = \nabla F_i(\boldsymbol{x}_{r,t}^i; \xi_{r,t}^i)$
10:                 $\boldsymbol{x}_{r,t'}^i = \boldsymbol{x}_{r,t}^i - \eta_l \boldsymbol{g}_{r,t}^i$
11:             **else**
12:                 $\boldsymbol{x}_{r,t'}^i = \boldsymbol{x}_{r,t}^i$
13:             **end if**
14:             **Gossip:**   $\boldsymbol{x}_{r,t+1}^i = \sum_{j \in \mathcal{N}^i} (W)_{i,j} \boldsymbol{x}_{r,t'}^j$
15:         **end for**
16:     **end for**
17:     Clients $i \in \mathcal{S}_r$ send $\Delta_r^i = \boldsymbol{x}_{r,\mathcal{I}}^i - \boldsymbol{x}_r$ to the server
18:     Aggregate model updates: $\Delta_r = \frac{1}{|\mathcal{S}_r|} \sum_{i \in \mathcal{S}_r} \Delta_r^i$
19:     $\boldsymbol{m}_r = \beta_1 \boldsymbol{m}_{r-1} + (1 - \beta_1)\Delta_r$
20:     $\boldsymbol{v}_r = \beta_2 \boldsymbol{v}_{r-1} + (1 - \beta_2)\Delta_r^2$
21:     $\widehat{\boldsymbol{v}}_r = \max(\widehat{\boldsymbol{v}}_{r-1}, \boldsymbol{v}_r)$ and $\widehat{\boldsymbol{V}}_r = \text{diag}(\widehat{\boldsymbol{v}}_r + \epsilon)$
22:     Server update $\boldsymbol{x}_{r+1} = \boldsymbol{x}_r + \eta \frac{\boldsymbol{m}_r}{\sqrt{\widehat{\boldsymbol{v}}_r} + \epsilon}$
23: **end for**

---

If we take a deeper look at the local update steps (Lines 5-16), the major difference between AFGA and FedAMS/FedAMSGrad is the *re-sample* step (Line 5) and the *gossip communication* step (Line 14), which we will discuss in detail in the following.

**Client re-sampling.** Note that for each global training round, we have already sampled a subset of participating clients $\mathcal{S}_r$. Normally (e.g., in FedAMS/FedAMSGrad), this will be the fixed chosen subset of clients who actually performs local gradient computations throughout this global training round. Yet for AFGA, we perform client re-sampling at each local iteration to obtain a new subset $\mathcal{S}_{r,t}$ and only the selected clients in the subset $\mathcal{S}_{r,t}$ are active for gradient computation in that local iteration, while the other clients will stay idle. Note that such a design does not incur extra local gradient computations as the size of $\mathcal{S}_{r,t}$ is the same as $\mathcal{S}_r$.

**Gossip communications.** After finishing the local gradient computations and model update with respect to the selected subset of clients, AFGA introduces a gossip communication step that allow each client to communicate their model weights with their connected neighbors (the selected subset of clients are connected in a graph $\mathcal{G}$ with a corresponding mixing matrix $W$). This gossip communication step is conducted by all local clients despite being selected in $\mathcal{S}_{r,t}$ or not. While in practice, clients are not required to know the whole mixing matrix $W$. Instead, client $i$ only needs to maintain the weights corresponding to those it receives from other clients.

It's important to note that AFGA does not necessitate the sampling of every client in specific rounds or iterations. In practical cases, if a client is unavailable, then it would not be sampled during local steps. If we remove the re-sampling step by setting $\mathcal{S}_{r,t} = \mathcal{S}_t$ and remove the gossip communication step by setting $\boldsymbol{x}^i_{r,t+1} = \boldsymbol{x}^j_{r,t'}$, AFGA will reduce to standard FedAMSGrad algorithm.

*Remark* 3.1. Our AFGA algorithm benefits from both gossip communications and client re-sampling while preserving the stable behavior of adaptive gradient methods. The steps of re-sampling in each local iteration help reduce the impact of data heterogeneity. It allows more clients to be included and participate in local training, which results in training a global model with a more balanced data distribution than without re-sampling. The peer-to-peer gossip communication is inspired by the recent advancement in decentralized optimization (Lian et al., 2017; Koloskova et al., 2020; Chen et al., 2021b), which has shown the ability to reduce the data heterogeneity issues between clients. By involving gossip communications in adaptive federated learning, it enables local models to average with their neighbors, thus preventing over-fitting on local data. The frequent re-sampling and gossip communications make AFGA effectively reduce the inconsistency between local clients, thus accelerating the overall convergence especially when the number of local steps increases. AFGA is also crucial to practical scenarios as it is compatible with low client participation ratios and limited local gradient computation capability while addressing the challenge of statistical heterogeneity in adaptive federated methods.

## 4 Convergence Analysis

In this section, we provide the theoretical convergence analysis of the proposed AFGA method. Before starting with the main theoretical results, let us first state the following assumptions based on stochastic optimization and adaptive gradient methods. For vector $\boldsymbol{x}$ and matrix $A$, we let $\|\boldsymbol{x}\| = \|\boldsymbol{x}\|_2$ and $\|A\| = \|A\|_F$. and $\|A\|_2$ represents the spectral norm of $A$. We denote $\mathbf{1}$ as vector with all elements equal to 1, and $\mathbf{I}$ as the identity matrix, with appropriate dimension.

**Assumption 4.1.** Each local objective function is $L$-smooth, i.e., $\forall \boldsymbol{x}, \boldsymbol{y} \in \mathbb{R}^d$, $\left| f_i(\boldsymbol{x}) - f_i(\boldsymbol{y}) - \langle \nabla f_i(\boldsymbol{y}), \boldsymbol{x} - \boldsymbol{y} \rangle \right| \leq \frac{L}{2} \|\boldsymbol{x} - \boldsymbol{y}\|^2, \forall i \in [N]$. This also implies the $L$-gradient Lipschitz condition, i.e., $\|\nabla f_i(\boldsymbol{x}) - \nabla f_i(\boldsymbol{y})\| \leq L\|\boldsymbol{x} - \boldsymbol{y}\|$.

**Assumption 4.2.** The stochastic gradient on each client has a bounded local variance, i.e., $\forall \boldsymbol{x} \in \mathbb{R}^d, i \in [N]$, there is $\mathbb{E}\left[\|\nabla f_i(\boldsymbol{x}, \xi) - \nabla f_i(\boldsymbol{x})\|^2\right] \leq \sigma^2$.

Assumption 4.1 to 4.2 are standard assumptions in centralized and federated non-convex stochastic optimization problems (Kingma & Ba, 2014; Li et al., 2019a; Yang et al., 2021; Reddi et al., 2020).

**Assumption 4.3.** Each local objective function $f_i(\boldsymbol{x})$ has $G$-bounded stochastic gradient on $\ell_2$, i.e., for all $\xi$, we have $\|\nabla f_i(\boldsymbol{x}, \xi)\| \leq G, \forall i \in [N]$.

Note that Assumption 4.3 is a standard assumption in non-convex *adaptive* optimization problems for under centralized and federated learning settings (Kingma & Ba, 2014; Reddi et al., 2018; Chen et al., 2020; Reddi et al., 2020; Wang et al., 2022a;b; 2024b)

**Assumption 4.4.** The dissimilarity between client's objective function and the global objective function is bounded, i.e., $\forall \boldsymbol{x} \in \mathbb{R}^d$, there is $\frac{1}{N} \sum_{i=1}^{N} \|\nabla f_i(\boldsymbol{x}) - \nabla f(\boldsymbol{x})\|^2 \leq \sigma_g^2$.

Assumption 4.4 captures the objective dissimilarity between the local and global objectives. Similar data heterogeneity assumption, which considers the variance between local clients, is common in federated learning (Reddi et al., 2020; Yang et al., 2021; Wang et al., 2022b; 2024a) and decentralized learning (Lian et al., 2017; Li et al., 2019b; Koloskova et al., 2020).

**Assumption 4.5** (Spectral gap)**.** For the gossip communications, clients are connected in the graph $\mathcal{G}$, and the corresponding weighting matrix $W$ is a *doubly stochastic matrix*: $W \in [0,1]^{n \times n}$, $W\mathbf{1} = \mathbf{1}$, $\mathbf{1}^\top W = \mathbf{1}^\top$ and null$(\mathbf{I} - W) = \text{span}(\mathbf{1})$. We assume the spectral gap $\rho$ of matrix $W$ satisfies: there exists $\rho \in [0,1)$ such that $\|W - \frac{1}{n}\mathbf{1}\mathbf{1}^\top\|_2 \leq \rho$.

Assumption 4.5 is highly related to the gossip communication update process and is usually assumed for decentralized learning framework (Koloskova et al., 2020; Chen et al., 2021b; Guo et al., 2021). For a doubly

stochastic matrix $W$, $\|W - \frac{1}{n}\mathbf{1}\mathbf{1}^\top\|_2 \leq 1$ naturally established [2], and the spectral gap $\rho \in [0,1)$ describes the connectivity of the clients: a smaller spectral gap $\rho$ indicates a denser connectivity between clients. Specifically, $\rho = 0$ indicates that all elements in the matrix $W$ are $\frac{1}{n}$, and this means that each client would be connected and communicated with other clients in the network with a mixing weight of $\frac{1}{n}$. $\rho \to 1$ means the matrix $W$ tends to be elements with either 0 or 1, corresponding to a graph that is nearly disconnected. We assume that there exists $\rho \in [0,1)$ to satisfy $\|W - \frac{1}{n}\mathbf{1}\mathbf{1}^\top\|_2 \leq \rho$ since our proposed method is contributed by gossip communications between clients. Several works (Lian et al., 2017; Li et al., 2019b) alternatively assume the spectral gap $\rho$ of a weighting matrix $W$ as the second largest eigenvalue of a doubly stochastic matrix $W$, i.e., $\rho = |\lambda_2(W)|$, and this spectral gap holds the same role for revealing the connectivity of the graph.

**Theorem 4.6** (Convergence analysis for Algorithm 1). *Under Assumptions 4.1-4.5, if the local learning rate $\eta = \Theta(N\sqrt{\mathcal{I}/M})$ and $\eta_l = \Theta(1/\sqrt{R\mathcal{I}^2})$, and the network spectral gap satisfies $\rho \leq \frac{M}{M+N}$, then the iterates of Algorithm 1 satisfy*

$$\frac{1}{R}\sum_{r=1}^{R}\mathbb{E}[\|\nabla f(\boldsymbol{x}_r)\|^2]$$
$$= \mathcal{O}\left(\frac{1}{\sqrt{R\mathcal{I}M}}\left[\Delta f + L(\sigma^2 + \sigma_g^2)\right]\right) + \mathcal{O}\left(\frac{1}{R}\left[G^2 + L^2(1+\rho^2)\sigma_g^2 + \frac{\rho^2\sigma^2}{\mathcal{I}}\right]\right) + \widetilde{\mathcal{O}}\left(\frac{1}{R^{3/2}}\right), \qquad (3)$$

*where $\Delta f = f_0 - f_*$, $\widetilde{\mathcal{O}}(\cdot)$ hides all the absolute constants and problem dependent constants including $\rho, \sigma, \sigma_g^2, \mathcal{I}, M, N, G$.*

**Corollary 4.7.** *Theorem 4.6 suggests that with sufficient amounts of global communication rounds $R$, i.e., $R \geq \mathcal{I}M$, our proposed method achieves a convergence rate of $\mathcal{O}\left(\frac{1}{\sqrt{R\mathcal{I}M}}\right)$. This improves the rate $\mathcal{O}\left(\frac{\sqrt{\mathcal{I}}}{\sqrt{RM}}\right)$ of adaptive federated optimization methods under partial client participation (Wang et al., 2022b), which suggests that AFGA can indeed bring accelerated convergence through local gossip communications.*

*Remark* 4.8. The sufficient amounts of global communication rounds that $R \geq \mathcal{I}M$ is a commonly-used condition to obtain the convergence rate in FL (Wang et al., 2022b; Yang et al., 2021). This condition is also practical as the algorithm usually converges after sufficient global rounds.

*Remark* 4.9. The impact of data heterogeneity is reflected in the variance $\sigma_g$ within the convergence rate. From equation 3, the variance $\sigma_g$ appears in $O\left(\frac{1}{\sqrt{R\mathcal{I}M}}\right)$ and smaller order terms w.r.t. $R$, $\mathcal{I}$, and $M$ for AFGA. The partially participated FedAvg in Yang et al. (2021) and adaptive FL models like FedAMS (Wang et al., 2022b) obtain the rate of $O\left(\frac{\sqrt{\mathcal{I}}}{\sqrt{RM}}\right)$, and the dominant $O\left(\frac{\sqrt{\mathcal{I}}}{\sqrt{RM}}\right)$ term directly relates to the variance $\sigma_g$. This demonstrates improvements over partially participated adaptive FL models like FedAMS (Wang et al., 2022b).

*Remark* 4.10. The second term of the convergence rate in equation 3 contains terms with spectral gap $\rho$ of the gossip communication network. A larger value of $\rho$ corresponds to a sparser network, which results in a larger variance term in the convergence rate, indicating that the dissimilarity variance has not been completely eliminated.

## 5 Communication-efficiency: Clustered-clients AFGA and Further Adaptation

**CAFGA.** While the frequent gossip communications enhance the overall performance of heterogeneous federated learning, it indeed involves extra peer-to-peer communication overhead, which makes our proposed AFGA less efficient in communication especially when clients are densely connected. Recent studies (Guo et al., 2021; Yuan et al., 2020) show that clients can be gathered into neighboring clusters based on locations or network capabilities, in which gossip communications are less expensive than communicating with the whole network. A similar idea of dividing clients into clusters has recently been studied in federated learning (Guo et al., 2021; Malinovsky et al., 2022; Long et al., 2022) and receives a lot of attention. Note that under a cluster-clients design, part of the network clients are grouped in a cluster, and clients within the

---

[2]Theoretical analysis is provided in the Appendix C.

---

**Algorithm 2** CAFGA: Clustered-Client Adaptive Federated Learning with Local Gossip Averaging

---

**Input:** initial point $\boldsymbol{x}_1$, local step size $\eta_l$ and global step size $\eta$, , optimizer hyperparameter $\beta_1, \beta_2, \epsilon$, doubly stochastic mixing matrix $W$

1: $\boldsymbol{m}_0 \leftarrow 0$, $\boldsymbol{v}_0 \leftarrow 0$
2: **for** $r = 1$ to $R$ **do**
3:     **for** each cluster $k \in [K]$ in parallel **do**
4:         Randomly sample a subset $\mathcal{S}_r^k$ for collecting local updates in round $r$
5:         Init: clients in $\mathcal{S}_r^k$ receive $\boldsymbol{x}_r$ from the server and broadcast $\boldsymbol{x}_r$ to all local neighbors
6:         **for** $t = 0, ..., \mathcal{I} - 1$ **do**
7:             Randomly **re-sample** a subset of clients $\mathcal{S}_{r,t}^k$ for gradient computation
8:             **for** each client $i \in \mathcal{V}_k$ in parallel **do**
9:                 **if** $i \in \mathcal{S}_{r,t}^k$ **then**
10:                     Compute $\boldsymbol{g}_{r,t}^i = \nabla F_i(\boldsymbol{x}_{r,t}^i; \xi_{r,t}^i)$
11:                     $\boldsymbol{x}_{r,t'}^i = \boldsymbol{x}_{r,t}^i - \eta_l \boldsymbol{g}_{r,t}^i$
12:                 **else**
13:                     $\boldsymbol{x}_{r,t'}^i = \boldsymbol{x}_{r,t}^i$
14:                 **end if**
15:                 **Gossip** Comm: $\boldsymbol{x}_{r,t+1}^i = \sum_{j \in \mathcal{N}_k^i} (W)_{i,j} \boldsymbol{x}_{r,t'}^j$
16:             **end for**
17:         **end for**
18:         Clients $i \in \mathcal{S}_r^k$ send $\Delta_r^i = \boldsymbol{x}_{r,\mathcal{I}}^i - \boldsymbol{x}_r$ to the server
19:     **end for**
20:     Aggregate: $\Delta_r = \frac{1}{K} \sum_{k \in [K]} \frac{1}{|\mathcal{S}_r^k|} \sum_{i \in S_r^k} \Delta_r^i$
21:     Server update follows Lines 19-22, Algorithm 1
22: **end for**

---

cluster can be connected through high-bandwidth peer-to-peer communications, leading to an efficient gossip communication network and a relatively smaller spectral gap. This leverages the communication efficiency for applying gossip communications while maintaining comparable performance for our proposed AFGA.

Suppose there are still in total $N$ clients, we study the same objective as Eq. equation 1 but we partition the clients into $K$ disjoint clusters where each of them has $n$ clients ($N = Kn$) [3]. We denote $\mathcal{V}_k$ as the set of local clients in cluster $k, k \in [K]$ and denote the neighbors for client $i \in \mathcal{V}_k$ as $\mathcal{N}_k^i$. Similar to Algorithm 1, we denote the weighted matrix of gossip averaging as $W_k$, and the corresponding spectral gap $\rho_k$. We then refer $\rho_{\max}$ as the maximum spectral gap among all clusters to represent the overall density in the network.

Algorithm 2 summarizes the proposed **C**lustered-clients paradigm **AFGA** (CAFGA). At the beginning of global round $r$, the server sample total $M$ clients (for convenience, uniformly sampled $m$ clients in each cluster) for global synchronization. The update rule *inside each cluster* follows the similar local update rule as Algorithm 1 with clients *re-sampling* and *gossip communications* in each local iteration, all clusters perform the training process parallelly. To be specific, at the $t$-th local iteration in cluster $k$, clients in the re-sampled subset $\mathcal{S}_{r,t}^k$ are active for gradient computation, while the unselcted clients stay idle. All clients in cluster $k$ then perform a gossip communication step with mixing matrix $W_k$. The leftover global update process of the clustered-client framework is the same as Algorithm 1 and FedAMS.

We also provide a complete theoretical convergence analysis for the Clustered-clients paradigm of AFGA (CAFGA); due to space limits, we referred interested readers to Appendix D for more details. In a nutshell, our theoretical analysis suggests that the convergence of CAFGA is related to $\rho_{\max}$ which aligns with the convergence rate of Algorithm 1, and implies that more densely connected gossip communications can help reduce the impact of data heterogeneity. Empirically we observe that under the same gossip communication structure (e.g., ring topology), the clustered-clients paradigm obtains performance improvement since

---

[3]We omit the clustering process in the algorithm for simplicity. The algorithm is compatible with various clustering methods, including clustering based on locations and network conditions, clustering based on client similarities, and random clustering.

grouping the whole network into disjoint clusters making the local clients more densely connected thus have smaller $\rho$ values. The clustered-clients paradigm enables efficient and dense connections for adequate model averaging, which keeps the benefits of further mitigating the effects of data heterogeneity.

**Communication-adapted: reduce the communication frequency.** Despite AFGA achieves a faster convergence rate, one noticeable drawback is that it requires all clients to stay online to conduct frequent gossip averaging, even if some of the clients do not participate in local training (gradient computation). We want to emphasize that this design is mainly for the ease of theoretical analysis. In practice, we can avoid this issue by enforcing gossip communications only on selected clients in each round[4]. As shown in the next Section, such a communication-adapted AFGA actually enjoys similar model training performances compared to the original AFGA algorithm without requiring dense gossip communications for all clients. Also, note that this communication-adapted can also be applied here to CAFGA for further improving communication efficiency while achieving similar model performances.

## 6 Experiments

**Datasets and models.** We conduct experiments on CIFAR-10 (Krizhevsky et al., 2009), CIFAR-100 (Krizhevsky et al., 2009) and Shakespeare (Caldas et al., 2018) dataset with various data sampling levels and client participation settings. We evaluate experiments on non-i.i.d. data distributions by a Dirichlet distribution partitioned strategy with parameter $\alpha = 0.6$ similar to Wang et al. (2020a;b). For image classification tasks on CIFAR-10 and CIFAR-100 datasets, we adopt a ConvMixer-256-8 network (Trockman & Kolter, 2022), which shares similar ideas to vision transformer (Dosovitskiy et al., 2021) to use patch embeddings to preserve locality and similarly, and is trained via adaptive gradient methods by default. For the next-character prediction task on Shakespeare, we adopt a 2-layer LSTM network, with 80-dimensional word embedding and 256 hidden units per layer, and follow a dropout layer with dropout rate 0.05.

**Baselines and methods.** We compare our method with several federated learning and adaptive federated learning baselines including FedAvg (McMahan et al., 2017), FedAdam (Reddi et al., 2020), FedAMS-Grad(Wang et al., 2022b)[5], SCAFFOLD (Karimireddy et al., 2020b) FedProx (Li et al., 2020b) and FedDyn (Acar et al., 2021).

**Implementation overview.** The number of local training iterations $\mathcal{I}$ on each client is set to 24 for experiments on CIFAR-10 and CIFAR-100 datasets, and $\mathcal{I} = 100$ for experiments on the Shakespeare dataset, and the batch size is set to 50 for all experiments by default. We report 500 rounds (denoted as #R or # Rounds in tables and figures) for the CIFAR 10 and the Shakespeare datasets and 600 rounds for the CIFAR-100 dataset. For each dataset and setting, the number of local training iterations $\mathcal{I}$ and the total rounds of training #R are fixed across all baseline methods to ensure a fair comparison. For local update, we use the SGD optimizer with a learning rate from {0.1, 1} for SGD-based global optimization methods (FedAvg, SCAFFOLD, FedProx, and FedDyn), and use SGD optimizer with a learning rate from {1,2,10} for adaptive global optimization methods. For a fair comparison, the local SGD updates apply no momentum and no gradient clipping steps for all methods. We set the global learning rate as 1 for SGD-based global update, and set the global learning rate as 0.01 for global adaptive optimization, FedAdam, FedAMSGrad, and our proposed AFGA. See Appendix B for more details about the experimental setup including datasets, models, and hyperparameter details.

### 6.1 Main Results

We summarize the performance of our proposed methods and other federated learning baselines in Table 1, Table 2 and 3. Due to space limits, we leave the learning curves and most ablation studies in Appendix B. Our experiments based on two settings, `Setting 1`: 100 clients with 5% participation ratio and `Setting 2`: 50 clients with 10% participation ratio. For the Clustered-clients AGFA (CAFGA), we evenly partition clients into 5 clusters for both settings, i.e., for `Setting 1`: there are 20 clients in each cluster with participate

---

[4]For example, suppose we train AFGA with a ring topology and select $M$ out of $N$ clients to participate in each round. We can form a new ring topology over the $M$ selected clients and only ask them to communicate over the new ring topology. In this way, the gossip communications only include these active clients while other unselected clients do not need to stay online and participate in the training process.

[5]FedAMSGrad is one of the variants of the FedAMS algorithm introduced in Wang et al. (2022b).

Table 1: The test accuracy of different methods on CIFAR-10 datasets. `Setting 1`: 100 clients, 5% participation. `Setting 2`: 50 clients, 10% participation. We report the average and the standard derivation over 3 runs with different random seeds.

| Method | Setting 1 | | Setting 2 | |
|---|---|---|---|---|
| | Acc. & std | R# (78%) | Acc.& std | R# (78%) |
| FedAvg | $75.57 \pm 1.10$ | 313 | $76.85 \pm 1.69$ | 180 |
| FedAdam | $77.07 \pm 0.05$ | 425 | $78.46 \pm 0.19$ | 157 |
| FedAMSGrad | $77.53 \pm 0.60$ | 388 | $79.59 \pm 0.76$ | 154 |
| SCAFFOLD | $76.94 \pm 1.17$ | 273 | $76.46 \pm 3.95$ | 146 |
| FedProx | $75.63 \pm 1.24$ | 500+ | $76.91 \pm 1.39$ | 180 |
| FedDyn | $77.68 \pm 0.06$ | 297 | $78.55 \pm 0.36$ | 160 |
| AFGA | $78.45 \pm 0.58$ | 302 | $80.02 \pm 2.00$ | 152 |
| CAFGA | $\mathbf{79.18} \pm 1.02$ | **233** | $\mathbf{82.10} \pm 0.67$ | **112** |

Table 2: The test accuracy of different methods on CIFAR-100 datasets. `Setting 1`: 100 clients, 5% participation. `Setting 2`: 50 clients, 10% participation. We report the average and the standard derivation over 3 runs with different random seeds.

| Method | Setting 1 | | Setting 2 | |
|---|---|---|---|---|
| | Acc. & std | R# (52%) | Acc.& std | R# (52%) |
| FedAvg | $49.88 \pm 0.33$ | 600+ | $51.21 \pm 0.23$ | 600+ |
| FedAdam | $50.40 \pm 0.53$ | 600+ | $51.99 \pm 0.81$ | 375 |
| FedAMSGrad | $50.58 \pm 0.99$ | 600+ | $52.15 \pm 0.58$ | 278 |
| SCAFFOLD | $51.71 \pm 0.21$ | 396 | $\mathbf{55.12} \pm 1.01$ | 235 |
| FedProx | $49.75 \pm 0.13$ | 600+ | $51.28 \pm 0.10$ | 600+ |
| FedDyn | $49.93 \pm 0.22$ | 396 | $51.93 \pm 0.52$ | 600+ |
| AFGA | $52.08 \pm 0.17$ | 436 | $53.87 \pm 0.68$ | 369 |
| CAFGA | $\mathbf{53.08} \pm 0.72$ | **380** | $54.41 \pm 0.24$ | **230** |

ratio 5%, and for `Setting 2:` there are 10 clients in each cluster with participate ratio 10%. We set ring topology as the default gossip communication topology.

**Results on CIFAR-10 and CIFAR-100.** Table 1 and Table 2 show the overall performance on training CIFAR-10 and CIFAR-100 datasets with ConvMixer-256-8 model. We observe that AFGA shows improvement upon other baselines, and the proposed CAFGA achieves better performance than AFGA.

For `Setting 1` on CIFAR-10, based on the results on three random seeds, AFGA shows an average 1.5% improvement in accuracy compared to SCAFFOLD, nearly 0.8% improvement compared to FedDyn, and nearly 1% improvement compared to FedAMSGrad, The proposed CAFGA, our extension on clustered-clients setting, further improves 0.7% accuracy based over AFGA. For `Setting 2` on CIFAR-10, CAFGA demonstrated around 3.5% increase in accuracy over FedDyn and 2% increase over FedAMSGrad. Note that in both settings, AFGA and CAFGA show their superior performance in achieving desired test accuracy. This demonstrates our proposed AFGA and CAFGA achieve overall better performance than adaptive federated learning methods and other federated learning baselines in both settings.

Table 2 shows that in experiments on CIFAR-100, for `Setting 1`, AFGA and CAFGA obtain higher test accuracy among other baselines including SCAFFOLD and FedDyn. Specifically, CAFGA significantly outperforms all baselines with more than 1.3% increase over SCAFFOLD and more than 2% increase over FedAdam and FedAMSGrad. For `Setting 2`, our proposed AFGA and CAFGA still outperform other federated learning baselines expect for SCAFFOLD.

Table 3: The test accuracy of different methods on Shakespeare datasets. `Setting 1`: 100 clients, 5% participation. `Setting 2`: 50 clients, 10% participation. We report the average and the standard derivation over 3 runs with different random seeds.

| Method | Setting 1 | | Setting 2 | |
|---|---|---|---|---|
| | Acc. | R#(52%) | Acc. | R# (52%) |
| FedAvg | $48.66 \pm 0.01$ | 500+ | $49.59 \pm 0.56$ | 500+ |
| FedAdam | $52.35 \pm 0.10$ | 391 | $52.23 \pm 0.08$ | 189 |
| FedAMSGrad | $52.09 \pm 0.23$ | 239 | $51.96 \pm 0.06$ | 220 |
| SCAFFOLD | $51.39 \pm 0.02$ | 500+ | $52.76 \pm 0.06$ | 166 |
| FedProx | $48.55 \pm 0.01$ | 252 | $49.33 \pm 0.48$ | 229 |
| FedDyn | $51.87 \pm 0.03$ | 500+ | $50.81 \pm 0.08$ | 500+ |
| AFGA | $53.08 \pm 0.08$ | **121** | $53.13 \pm 0.04$ | 108 |
| CAFGA | $\mathbf{53.20} \pm 0.05$ | 152 | $\mathbf{53.57} \pm 0.02$ | **91** |

**Results on Shakespeare.** Table 3 shows the overall performance of training the Shakespeare dataset with a 2-layer LSTM network. For `Setting 1` on Shakespeare, AFGA shows approximately 2.5% improvement in accuracy compared to SCAFFOLD, and 0.5% improvement compared to FedAMSGrad. The proposed CAFGA achieves even higher final accuracy than AFGA and significantly outperforms other baselines. For `Setting 2` on Shakespeare, AFGA addresses increasing in accuracy over FedAMSGrad, while CAFGA outperforms AFGA in final results. In both settings, AFGA or CAFGA show their superior performance in achieving desired test accuracy.

## 6.2 Ablation Studies

**Sensitivity of gossip averaging and client re-sampling.** We conduct experiments studying how the individual components, gossip averaging and re-sampling, and the clustered-clients framework contribute to the proposed AFGA and CAFGA. Table 4 presents the ablation study of the contribution of individual components, which indicates that the gossip averaging and client re-sampling simultaneously contribute to the accuracy improvements of AFGA. Furthermore, by the results from Table 4 (also with the observation of the learning curves in Appendix B), it shows that the clustered-clients paradigm further improves overall accuracy. These results show our intuition of utilizing gossip averaging and client re-sampling can effectively mitigate data heterogeneity, and also address the benefit of the clustered-client framework that consistently helps improve the performance. In addition to the aforementioned ablation studies, we have also conducted further ablation studies to investigate the effect of data heterogeneity, examine different gossip averaging topologies to understand the impact of the spectral gap on model performance, and explore the effects of varying the number of local iterations. Due to constraints on space, we provide detailed ablation studies and results in Appendix.

Table 4: Ablation of components at the last 5 rounds (in total 500 rounds) in training CIFAR-10 on ConvMixer-256-8 model.

| Methods (FedAMSGrad) | Acc. |
|---|---|
| FedAMSGrad Only | $79.59 \pm 0.76$ |
| +Gossip | $77.39 \pm 0.01$ |
| +Gossip + Re-sampling (**AFGA**) | $80.02 \pm 2.00$ |
| +Gossip + Re-sampling + Clustered (**CAFGA**) | $\mathbf{82.10} \pm 0.67$ |

**Ablation of gossip averaging topology.** We also conduct ablation studies on how the gossip averaging topology affects the overall performance in (C)AFGA. Figure 1 shows the ablation study on (a) spectral gap in AFGA and (b) clusters' maximum spectral gap $\rho$ for 5 clusters in CAFGA.

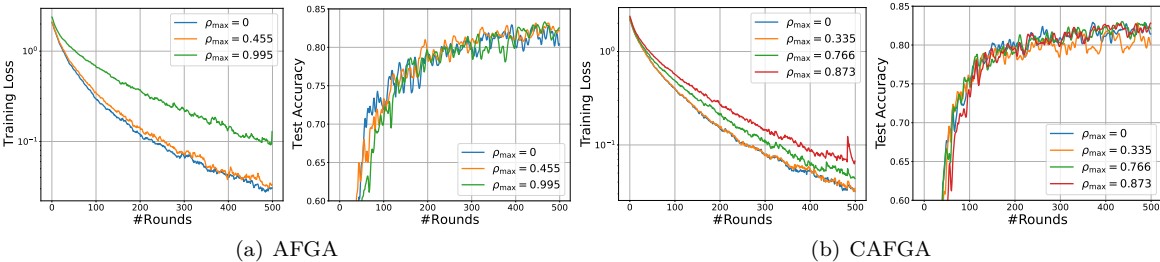

(a) AFGA

(b) CAFGA

Figure 1: Ablation study with different heterogeneity degree of (C)AFGA in training CIFAR-10 on ConvMixer-256-8 model.

We follow the `Setting 2` in Table 1, i.e., 50 clients with a participation ratio of 0.1. For AFGA, we compare various of $\rho$ from $\rho = \{0, 0.455, 0.995\}$ calculated by balanced fully-connected, random, and ring typologies correspondingly. We observe that the balanced fully-connected topology with $\rho = 0$ contributes to faster convergence, which aligns with the theoretical result that smaller $\rho$ can help reduce the impact of data heterogeneity. Similar to CAFGA, we compare various of $\rho_{\max}$ from $\rho_{\max} = \{0, 0.335, 0.766, 0.873\}$ calculated by balanced fully-connected, unbalanced fully-connected [6], random, and ring typologies correspondingly. It shows that the fully-connected topology (relatively small $\rho_{\max}$ value) results in faster convergence as well.

**Ablation on data heterogeneity.** We further conduct experiments to investigate the impact of data

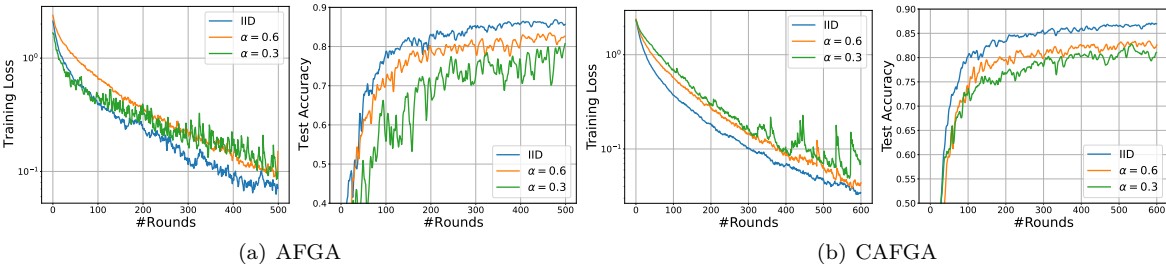

(a) AFGA

(b) CAFGA

Figure 2: Ablation study with different heterogeneity degree of AFGA and CAFGA in training CIFAR-10 on ConvMixer-256-8 model.

heterogeneity as theoretically the proposed (C)AFGA show that the convergence rate is highly related to the model dissimilarity. We use Dirichlet($\alpha$) distribution for data partitioned in experiments, where $\alpha$ represents the degree of heterogeneity (smaller $\alpha$ implies more heterogeneous data distribution), and we choose $\alpha$ from $\{0.3, 0.6\}$ together with the i.i.d. data partitioned setting for ablation study. The rest of the experimental setup is the same as `Setting 2` in Table 1. Figure 2 shows the learning curves for different non-i.i.d. degrees. We observe that the data heterogeneity across clients still significantly affects the convergence and generalization performance for our proposed (C)AFGA, as a more balanced data distribution attains faster convergence and higher accuracy.

## 6.3 Communication-Adapted AFGA and CAFGA

Table 5 shows the test accuracy and the total global round to reach target test accuracy of communication-adapted AFGA and communication-adapted CAFGA which we have discussed in the previous Section. We can observe that both communication-adapted methods achieve similar test accuracy compared with their original version. This suggests that in practice we can still solve the data heterogeneity issue without requiring all clients to participate in gossip communications. Due to space limitations, we left additional results of the CIFAR-100 dataset in Appendix B.

---

[6]This means that clients in the cluster are connected with all their neighbors but with random weighted averaging elements.

Table 5: The test accuracy (Acc.) and the total global rounds (R#) to reach 78% test accuracy of different methods when training ConvMixer-256-8 model on CIFAR-10 dataset, where (a) denotes the communication-adaptive version. `Setting 1`: 100 clients, 5% participation. `Setting 2`: 50 clients, 10% participation. `Setting 3`: 100 clients, 10% participation. `Setting 4`: 50 clients, 20% participation. To mitigate the effect of randomness and fluctuation of the accuracy, we take the average of the last 5 global rounds to represent final accuracy.

| Method | Setting 1 | | Setting 2 | |
|---|---|---|---|---|
| | Acc. | R# | Acc. | R# |
| AFGA | 78.80 | 302 | 82.61 | 152 |
| **AFGA (a)** | 76.59 | 419 | 81.51 | 259 |

| Method | Setting 3 | | Setting 4 | |
|---|---|---|---|---|
| | Acc. | R# | Acc. | R# |
| CAFGA | 81.56 | 142 | 83.15 | 69 |
| **CAFGA (a)** | 81.15 | 194 | 82.99 | 136 |

## 7 Conclusions and Future Works

In this paper, we propose a novel adaptive federated optimization algorithm, AFGA, that addresses data heterogeneity across clients and mitigates local model inconsistency by introducing gossip communications and client re-sampling during local training steps. We present a completed theoretical convergence analysis for the proposed AFGA. We prove that AFGA achieves a faster convergence rate than the previous adaptive federated optimization method for partial participation scenarios with heterogeneous data under non-convex stochastic settings. We extend AFGA to a more communication-efficient clustered-clients paradigm, where clients are divided into disjoint clusters and we only perform local gossip averaging within the clusters. The extended CAFGA algorithm is aimed at reducing the communication overhead introduced by gossip communications while maintaining the benefits of client re-sampling and gossip communications under heterogeneous data. Experiments on several benchmarks and ablation studies backup our theory.

Despite successfully tackling the data heterogeneity issue among clients by introducing gossip communications and client re-sampling, our current proposed methods also have certain limitations. First, extending the theoretical analysis to the communication-adapted versions is challenging and highly non-trivial. Moreover, gossip communications also incur extra challenges if attempting to further apply secure aggregation schemes to our method. Also if not all clients are trusted and there exist malicious clients, the frequent gossip communications between clients may increase the risks of model poisoning or privacy attacks. We leave those new challenges as future works.

## Acknowledgments

We thank the anonymous reviewers for their helpful comments. This work is partially supported by the National Science Foundation under Grant No. 2348541. The views and conclusions contained in this paper are those of the authors and should not be interpreted as representing any funding agencies.

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

## A  Related Work

**Federated learning.** Federated learning (Konečnỳ et al., 2016) play a critical role in collaboratively training models at edge devices with potential privacy protections. Basic optimization methods for federated learning include SGD-based global optimizer, e.g., FedAvg (McMahan et al., 2017) (a.k.a. Local SGD (Stich, 2018) and its variants (Li et al., 2019a; Yang et al., 2021), adaptive gradient optimization based global optimizer such as FedAdam, FedAdagrad, FedYogi (Reddi et al., 2020), FedAGM (Tong et al., 2020) and FedAMS-Grad (Wang et al., 2022b). While these optimization methods for federated learning show their ability on achieving stable results when data are heterogeneously distributed, they rarely study data heterogeneity

itself. Recently, several works address the data heterogeneity issue through several aspects. For example, FedProx (Li et al., 2020a) adds a proximate term to align the local model with the global one, and FedDyn (Acar et al., 2021) involves dynamic regularization term for local and global model consistency. FedNova (Wang et al., 2020b) proposes a normalized averaging mechanism that reduces objective inconsistency with heterogeneous data. Moreover, several works study to eliminate the client drift caused by data heterogeneity from the aspect of variance reduction such as (Karimireddy et al., 2020b;a; Khanduri et al., 2021; Cutkosky & Orabona, 2019). They introduce additional control variables to track and correct the local model shift during local training, but they require extra communication costs for synchronizing these control variables. Besides, FedDC (Gao et al., 2022) involves both dynamic regularization terms and local drift variables for model correction.

**Decentralized learning and beyond.** Decentralized learning studies a distributed machine learning paradigm without a central server. It can been tracked back from gossip averaging techniques (Tsitsiklis, 1984; Boyd et al., 2006). Decentralized (gossip) SGD algorithms (Lian et al., 2017; Li et al., 2019b; Boyd et al., 2006; Tang et al., 2018) are then proposed that consider client-to-client communications after each step of SGD update on the client for decentralized learning. Lu & De Sa (2021) proves a tight lower bound for decentralized training under the non-convex setting. Teng et al. (2019) proposes a leader-distributed SGD algorithm that pulls workers to the currently best-performing model among all models. There are recent studies generalized various distributed SGD algorithms under unified frameworks, where Wang & Joshi (2021) included reducing communication costs and decentralized training in i.i.d. settings, and Koloskova et al. (2020) studied a general network topology-changing gossip SGD methods that summarize several algorithms in distributed and federated learning.

Recent studies extend the decentralized training paradigm to federated learning with various adaption. For example, Guo et al. (2021) considered heterogeneous communications for modern communication networks that improve communication efficiency, and hierarchical federated learning algorithms (Liu et al., 2020; Abad et al., 2020; Castiglia et al., 2020) develop frameworks by aggregating client models to edge servers first before synchronizing them to the central server.

# B  Additional Experiments

In this section, we present additional empirical results for our proposed algorithm AFGA and CAFGA in training ConvMixer-256-8 model (Trockman & Kolter, 2022) on CIFAR-10/100 (Krizhevsky et al., 2009) datasets, and in training LSTM model on Shakespeare (Caldas et al., 2018) dataset. All experiments in this paper are conducted on 4 NVIDIA RTX A6000 GPUs.

## B.1  Additional Experimental Results

**Additional Experimental Results on CIFAR-10.**  Figure 3 shows the overall test accuracy curves of experiments on CIFAR-10. It demonstrates that our proposed AFGA and CAFGA achieve overall better performance than adaptive federated learning methods and other federated learning baselines in both settings.

**Additional Experimental Results on training ResNet-18 on CIFAR-10.**  Table 6 presents the empirical result for our proposed AFGA and CAFGA together with several federated learning baselines on training CIFAR-10 with ResNet-18 model. It shows that the proposed CAFGA outperforms other federated learning baselines, achieving a 0.4% improvement over FedAMSGrad and an enhancement of more than 1% compared to other baselines.

**Additional Experimental Results on CIFAR-100.**  Figure 4 shows the empirical result for our proposed AFGA and CAFGA together with several federated learning baselines on training CIFAR-100 ConvMixer-256-8 model. They demonstrate that our proposed AFGA and CAFGA achieve overall better performance than adaptive federated learning methods and other federated learning baselines in both settings.

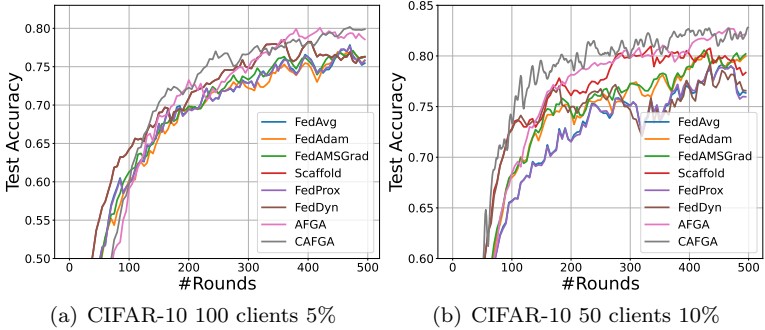

(a) CIFAR-10 100 clients 5%      (b) CIFAR-10 50 clients 10%

Figure 3: The test accuracy for AFGA and CAFGA with several federated learning baselines in training CIFAR-10 data on ConvMixer-256-8 model.

Table 6: The test accuracy of training ResNet-18 model on CIFAR-10 datasets considering 50 clients and 10% participation ratio.

| Method | Acc. & std. |
|---|---|
| FedAvg | $70.32 \pm 0.44$ |
| FedAdam | $73.80 \pm 0.58$ |
| FedAMSGrad | $75.59 \pm 0.73$ |
| SCAFFOLD | $74.60 \pm 0.67$ |
| FedProx | $70.26 \pm 0.45$ |
| FedDyn | $74.15 \pm 2.23$ |
| AFGA | $74.72 \pm 0.32$ |
| CAFGA | $\mathbf{75.96} \pm 0.67$ |

### B.1.1 Ablation Studies and Other Comparisons

**Ablation of local iteration.** We further study how the local iteration affects the convergence of our proposed CAFGA algorithm. Figure 6 shows the ablation study about local iterations, we compare the number of local iterations $\mathcal{I}$ from $\mathcal{I} = \{12, 24, 48\}$. We observe that larger $\mathcal{I}$ indeed helps accelerate convergence on training loss and helps to obtain a higher test accuracy. This result backs up our theory that the increasing number of local steps would help the overall performance.

**Communication run-time simulations.** Table 7 presents a simulation study as a substitution of the real-world measurement similar to Guo et al. (2021). Consider a limited bandwidth setting where the average time of client-to-client communication cost is 1.8 seconds, and the average time of client-to-server communication cost is 18 seconds. Table 7 suggests that even when considering client-to-client communication costs, our proposed AFGA and CAFGA can still efficiently achieve high accuracy with less overall communication costs. This implies that though our proposed methods incur extra local gossip communications, it helps mitigate the impact of data heterogeneity thus improve the overall performance.

Table 7: The communication time under for CIFAR-10. Setting 1: 100 clients, 5% participation ratio.

| Test Accuracy | 70% | 75% | 78% | 80% |
|---|---|---|---|---|
| FedAMSGrad time (h) | **76.0** | 128.0 | 194.0 | 281.0 |
| AFGA time (h) | 91.76 | 130.98 | 223.48 | 250.86 |
| CAFGA time (h) | 76.22 | **102.86** | **172.42** | **179.82** |

**Comparisons to Decentralized Methods.** We have briefly discussed decentralized learning in the related work in the main paper, here we provide more discussion about our proposed methods and the decentralized algorithms. Decentralized learning can certainly prevent single point failure without a central

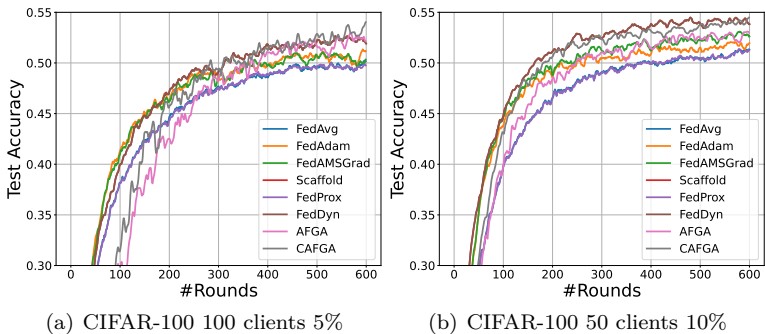

(a) CIFAR-100 100 clients 5%          (b) CIFAR-100 50 clients 10%

Figure 4: The test accuracy for AFGA and CAFGA with several federated learning baselines in training CIFAR-100 data on ConvMixer-256-8 model.

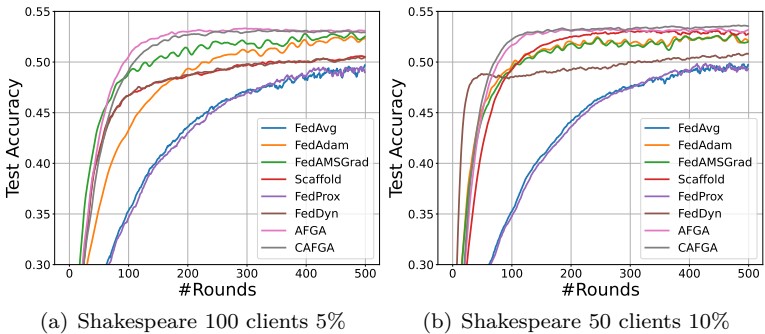

(a) Shakespeare 100 clients 5%          (b) Shakespeare 50 clients 10%

Figure 5: The test accuracy for AFGA and CAFGA with several federated learning baselines in training Shakespeare data on LSTM model.

server, its performance is not on par with the conventional server-client FL setup, especially when data are heterogeneous distributed. In sharp contrast, the periodic synchronization between server and clients in our proposed method can help align local models for better convergence and ease the data heterogeneity issue in adaptive federated learning, which is mainly focus of this paper. Moreover, the central server setting allow us to easily apply adaptive optimizer for stable performances, while decentralized learning methods are mostly limited to SGD-based update as the adaptive optimizer needs the alignment of gradient update, otherwise suffers from some divergence issue (Chen et al., 2021a).

We provide some experimental results comparing our proposed methods with decentralized algorithms including DSGD (Lian et al., 2017), DAdam (Nazari et al., 2019; Chen et al., 2021a) and PGA (Chen et al., 2021b) under the same training settings. The following table shows the comparison result for several decentralized methods and our proposed AFGA and CAFGA. It shows that our proposed AFGA and CAFGA indeed attains better test accuracy results comparing to other decentralized baselines.

Table 8: Comparison to decentralized algorithms.

| Method | Test Accuracy(%) | Rounds (78%) |
|--------|------------------|--------------|
| DSGD   | 80.23            | **50**       |
| DAdam  | 70.37            | 227          |
| PGA    | 80.93            | 210          |
| AFGA   | 82.16            | 152          |
| CAFGA  | **83.03**        | 112          |

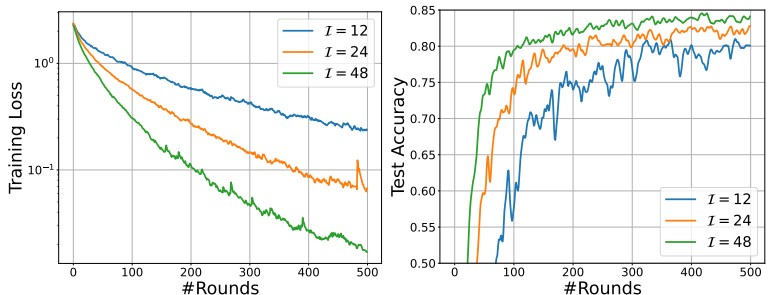

Figure 6: Ablation study with different heterogeneity degree of CAFGA in training CIFAR-10 on ConvMixer-256-8 model.

## B.2 Hyper-parameters Details

**Hyper-parameter Settings.** We conduct detailed hyper-parameter searches to find the best hyper-parameter for each baseline. We grid over the local learning rate $\eta_l \in \{0.001, 0.01, 0.1, 1.0\}$, and the global learning rate $\eta \in \{0.001, 0.01, 0.1, 1.0, 2.0, 5.0, 10.0\}$ for each method. For the global AMSGrad optimizer, we set $\beta_1 = 0.9$, $\beta_1 = 0.99$, and we search the best $\epsilon$ from $\{10^{-10}, 10^{-8}, 10^{-6}, 10^{-4}\}$. Table 9 summarizes the hyper-parameter details in our experiments.

Experiments are set up with `Setting 1`: 100 total clients, and `Setting 2`: 50 total clients in the network. For CAFGA, clients are equally divided into 5 clusters. The partial participation ratio is set to $p = 0.05$ for `Setting 1` and $p = 0.1$ for `Setting 2`, and the gossip communication topology is ring topology by default. For each method, we conduct $\mathcal{I} = 24$ iterations of local training with a batch size of 50 by default.

Table 9: Hyper-parameters details.

| | Setting 1 (100 clients 5% participation) | | | | | | | | | | | | | | | |
|---|---|---|---|---|---|---|---|---|---|---|---|---|---|---|---|---|
| | FedAvg | | FedAdam | | FedAMSGrad | | SCAFFOLD | | FedProx | | FedDyn | | AFGA | | CAFGA | |
| Data | $\eta_l$ | $\eta$ | $\eta_l$ | $\eta$ | $\eta_l$ | $\eta$ | $\eta_l$ | $\eta$ | $\eta_l$ | $\eta$ | $\eta_l$ | $\eta$ | $\eta_l$ | $\eta$ | $\eta_l$ | $\eta$ |
| CIFAR-10 | 0.1 | 1.0 | 1.0 | 0.01 | 1.0 | 1.0 | 0.1 | 1.0 | 0.1 | 1.0 | 0.1 | 1.0 | 0.01 | 2.0 | 0.01 | 2.0 |
| CIFAR-100 | 0.1 | 1.0 | 1.0 | 0.01 | 1.0 | 1.0 | 0.1 | 1.0 | 0.1 | 1.0 | 0.1 | 1.0 | 0.01 | 1.0 | 0.01 | 1.0 |
| Shakespeare | 1.0 | 1.0 | 1.0 | 0.01 | 1.0 | 1.0 | 0.1 | 1.0 | 0.1 | 1.0 | 0.1 | 1.0 | 0.01 | 10.0 | 0.01 | 10.0 |
| | Setting 2 (50 clients 10% participation) | | | | | | | | | | | | | | | |
| | FedAvg | | FedAdam | | FedAMSGrad | | SCAFFOLD | | FedProx | | FedDyn | | AFGA | | CAFGA | |
| Data&Model | $\eta_l$ | $\eta$ | $\eta_l$ | $\eta$ | $\eta_l$ | $\eta$ | $\eta_l$ | $\eta$ | $\eta_l$ | $\eta$ | $\eta_l$ | $\eta$ | $\eta_l$ | $\eta$ | $\eta_l$ | $\eta$ |
| CIFAR-10 | 0.1 | 1.0 | 1.0 | 0.01 | 1.0 | 1.0 | 0.1 | 1.0 | 0.1 | 1.0 | 0.1 | 1.0 | 0.01 | 2.0 | 0.01 | 2.0 |
| CIFAR-100 | 0.1 | 1.0 | 1.0 | 0.01 | 1.0 | 1.0 | 0.1 | 1.0 | 0.1 | 1.0 | 0.1 | 1.0 | 0.01 | 1.0 | 0.01 | 1.0 |
| Shakespeare | 1.0 | 1.0 | 1.0 | 0.01 | 1.0 | 1.0 | 0.1 | 1.0 | 0.1 | 1.0 | 0.1 | 1.0 | 0.01 | 10.0 | 0.01 | 10.0 |

# C Preliminaries

**About weighted matrix:**

$$\text{null}(I_n - W_k) = \text{span}(\mathbf{x}|\mathbf{x} \in \mathbb{R}^n : (I_n - W_k)\mathbf{x} = \mathbf{0}) \tag{4}$$

If we have $\text{null}(I_n - W_k) = \text{span}(\mathbf{1})$, that means the following equation holds

$$\begin{pmatrix} 1-w_{11} & -w_{12} & ... & -w_{1n} \\ -w_{21} & 1-w_{22} & ... & -w_{2n} \\ \vdots & & & \\ -w_{n1} & ... & ... & 1-w_{nn} \end{pmatrix} \cdot \begin{pmatrix} x_1 \\ x_2 \\ \vdots \\ x_n \end{pmatrix} = \mathbf{0} \tag{5}$$

if and only if $x_1 = x_2 = ... = x_n$. Since we assume $w_{ij} \in [0,1]$, there is a counter-example if $W_k = (W_2, 0; 0, I_{n-2})$, then we have

$$\begin{pmatrix} 1 - w_{11} & -w_{12} & ... & 0 \\ -w_{21} & 1 - w_{22} & ... & 0 \\ \vdots & & & \\ 0 & & I_{n-2} & \end{pmatrix} \cdot \begin{pmatrix} x_1 \\ x_2 \\ \vdots \\ x_n \end{pmatrix} = \mathbf{0}, \tag{6}$$

then $(x_1, x_2, ..., x_n) = (c_1, c_2, 0, ..., 0)(c_1, c_2 \neq 0)$ can be a solution to the equation.

For the eigenvalues and eigenvectors of matrix $W_k - (1/n)\mathbf{1}\mathbf{1}^T$, we have

$$\begin{aligned} W_k - (1/n)\mathbf{1}\mathbf{1}^T &= W_k - J, \\ (W_k - J)(W_k - J) &= W_k - J, \\ \lambda \boldsymbol{b} = (W_k - J)\boldsymbol{b} &= (W_k - J)(W_k - J)\boldsymbol{b} = (W_k - J)\lambda \boldsymbol{b} = \lambda^2 \boldsymbol{b}, \\ (\lambda^2 - \lambda)\boldsymbol{b} &= 0, \quad \lambda = 1 \text{ or } \lambda = 0. \end{aligned} \tag{7}$$

The eigenvectors of $W_k - J$ are $(1, -1, 0, ..., 0)$, $(1, 0, -1, ..., 0)$,..., $(1, 1, 1, ..., 1)$.

The maximum of $\|W_k - (1/n)\mathbf{1}\mathbf{1}^T\|_2$ is obtained when $W_k$ is equivalent to $I_n$, and we have $\max \|W_k - (1/n)\mathbf{1}\mathbf{1}^T\|_2 = 1$, which implies $\|W_k - (1/n)\mathbf{1}\mathbf{1}^T\|_2 \leq 1$.

## D    Convergence Analysis for Clustered-clients framework

We re-state two cluster related assumptions, the assumption of inter-client dissimilarity and the assumption of gossip mixing spectral gap, and the theorem in the following. First, we denote the local objective function of cluster $k$, i.e., $\bar{f}_k(\boldsymbol{x}) = \frac{1}{n}\sum_{i \in \mathcal{V}_k} f_i(\boldsymbol{x})$, then we state the following assumptions.

**Assumption D.1.** The dissimilarity between client's objective function and corresponding cluster's objective function is bounded, i.e., for all $\boldsymbol{x}, k \in [K]$, there is $\frac{1}{n}\sum_{i \in \mathcal{V}_k} \|\nabla f_i(\boldsymbol{x}) - \nabla \bar{f}_k(\boldsymbol{x})\|^2 \leq \sigma_k^2$. Similarly, clusters' objective function the global objective has a bounded dissimilarity variance: for $\alpha \geq 1$ and $\sigma_c \geq 0$, there is $\frac{1}{K}\sum_{k \in [K]} \|\nabla \bar{f}_k(\boldsymbol{x})\|^2 \leq \alpha^2 \|\nabla f(\boldsymbol{x})\|^2 + \sigma_c^2$.

**Assumption D.2** (Intra-cluster spectral gap)**.** Local clients in cluster $k \in [K]$ are connected in the graph $\mathcal{G}_k$ with weighting matrix $W_k$ satisfies same characteristic as Assumption 4.5. We assume the spectral gap $\rho_k$ satisfies: there exists $\rho_k \in [0,1)$ such that $\|W_k - \frac{1}{n}\mathbf{1}\mathbf{1}^\top\|_2 \leq \rho_k$.

We further denote $\bar{\sigma}_l^2 = \frac{1}{K}\sum_{k=1}^K \sigma_k^2$ as the average dissimilarity between local clients in the same cluster, and denote $\rho_{\max} = \max_{k \in [K]} \rho_k$ as the maximum spectral gap among all $K$ clusters.

Note that Assumption D.1 and Assumption D.2 are general assumptions for the clustered-client AFGA (CAFGA) framework. Specifically, if $K = 1$, i.e., there is only one cluster containing all clients $[N]$ in the network:

- Assumption D.1 reduces to Assumption 4.4: the cluster's objective function in Assumption D.1 is exactly the global objective, thus $\sigma_c = 0$, and there is a bounded dissimilarity between client's objective and global objective: $\frac{1}{n}\sum_{i \in [N]} \|\nabla f_i(\boldsymbol{x}) - \nabla f(\boldsymbol{x})\|^2 \leq \sigma_g^2$, which is consistent with Assumption 4.4.

- Assumption D.2 reduces to Assumption 4.5: the only cluster contains all $i \in [N]$, clients $i \in [N]$ are connected in the graph $\mathcal{G}$ with weighting matrix $W$ and there exists $\rho \in [0,1)$ such that $\|W - \frac{1}{n}\mathbf{1}\mathbf{1}^\top\|_2 \leq \rho$, which is consistent with Assumption 4.5.

In the following, we state the convergence rate for Algorithm 2, (CAFGA).

**Theorem D.3.** *Under Assumptions 4.1-4.3, D.1 and D.2, if the local learning rate satisfies specific constraints, and the maximum spectral gap satisfies $\rho_{\max} \leq \frac{m}{m+n}$, then the iterates of Algorithm 2 in partial participation scenarios satisfy*

$$\frac{1}{R}\sum_{r=1}^{R}\mathbb{E}[\|\nabla f(\boldsymbol{x}_r)\|^2] = \mathcal{O}\left(\frac{1}{\sqrt{R\mathcal{I}m}}\left[[f_0 - f_*] + \left(\widetilde{\sigma}^2 + \frac{\sigma^2}{K}\right)L\right]\right)$$
$$+ \mathcal{O}\left(\frac{1}{R}\left[G^2 + L^2[\sigma_c^2 + (1 + \rho_{\max}^2)\bar{\sigma}_l^2] + \frac{\rho_{\max}^2\sigma^2}{\mathcal{I}}\right]\right) + \widetilde{\mathcal{O}}\left(\frac{1}{R^{3/2}}\right), \tag{8}$$

where $\widetilde{\mathcal{O}}(\cdot)$ hides all the absolute constants and problem dependent constants including $\rho, \sigma, \sigma_c^2, \bar{\sigma}_l^2, \mathcal{I}, M, N$, and additionally we denote $\widetilde{\sigma}^2 = \sigma^2 + \bar{\sigma}_l^2 + \sigma_c^2$ as the variance summation.

## E  Proof of Theorem D.3 and Theorem 4.6

**Preliminaries for the proof.**    We define the following auxiliary sequences, w.r.t. $\boldsymbol{x}_{r,t}$, $\boldsymbol{x}_{r,t}^k$. Firstly, we denote the average model on cluster $k$ as

$$\bar{\boldsymbol{x}}_{r,t+1}^k = \bar{\boldsymbol{x}}_{r,t}^k - \eta_l \bar{\boldsymbol{g}}_{r,t}^k, \tag{9}$$

where $\bar{\boldsymbol{g}}_{r,t}^k = \frac{1}{n}\sum_{i\in\mathcal{V}_k}\bar{\boldsymbol{g}}_{r,t}^i$, where $\bar{\boldsymbol{g}}_{r,t}^i = 0$ as a **virtual gradient** if client $i$ in $\mathcal{V}_k$ has not been selected in the set $\mathcal{S}_{t,r}^k$, and $\bar{\boldsymbol{g}}_{r,t}^i$ is equal to the **real gradient** $\boldsymbol{g}_{r,t}^i$. if $i \in \mathcal{S}_{t,r}^k$ We also define the global average model

$$\bar{\boldsymbol{x}}_{r,t+1} = \bar{\boldsymbol{x}}_{r,t} - \eta_l \frac{1}{N}\sum_{i=1}^{N}\bar{\boldsymbol{g}}_{r,t}^i. \tag{10}$$

We next define sequences related to model differences, we denote the average model difference on cluster $k$ as $\bar{\Delta}_r^k$, and the average global model difference $\bar{\Delta}_r$ without sampling consideration.

$$\bar{\Delta}_r^k = \frac{1}{n}\sum_{i\in\mathcal{V}_k}\Delta_r^i = \frac{1}{n}\sum_{i\in\mathcal{V}_k}(\boldsymbol{x}_{r,\mathcal{I}}^i - \boldsymbol{x}_r) = \bar{\boldsymbol{x}}_{r,\mathcal{I}}^k - \boldsymbol{x}_r = \bar{\boldsymbol{x}}_{r,0}^k - \eta_l\sum_{t=0}^{\mathcal{I}-1}\bar{\boldsymbol{g}}_{r,t}^k - \boldsymbol{x}_r = -\eta_l\sum_{t=0}^{\mathcal{I}-1}\bar{\boldsymbol{g}}_{r,t}^k,$$
$$\bar{\Delta}_r = \frac{1}{K}\sum_{k\in[K]}\frac{1}{n}\sum_{i\in\mathcal{V}_k}(\boldsymbol{x}_{r,\mathcal{I}}^i - \boldsymbol{x}_r) = \frac{1}{K}\sum_{k\in[K]}\bar{\Delta}_r^k = -\eta_l\frac{1}{K}\frac{1}{n}\sum_{t=0}^{\mathcal{I}-1}\sum_{k\in[K]}\sum_{i\in\mathcal{V}_k}\bar{\boldsymbol{g}}_{r,t}^i. \tag{11}$$

Since we have two sampling process: sampling clients for global communication per global round $r$, and sampling selected clients for local gradient update per local iteration $t$. Thus we state the following auxiliary equations

$$\mathbb{E}_{\mathcal{S}_{t,r}^k}[\bar{\boldsymbol{g}}_{t,r}^k] = \mathbb{E}_{\mathcal{S}_{t,r}^k}\left[\frac{1}{n}\sum_{i\in\mathcal{V}_k}\bar{\boldsymbol{g}}_{t,r}^i\right] = \mathbb{E}_{\mathcal{S}_{t,r}^k}\left[\frac{1}{n}\sum_{i\in\mathcal{S}_{t,r}^k}\boldsymbol{g}_{t,r}^i\right],$$
$$\mathbb{E}_{\mathcal{S}_{t,r}^k}[\bar{\Delta}_r^k] = \mathbb{E}_{\mathcal{S}_{t,r}^k}\left[-\eta_l\sum_{t=0}^{\mathcal{I}-1}\bar{\boldsymbol{g}}_{t,r}^k\right] = \mathbb{E}_{\mathcal{S}_{t,r}^k}\left[-\frac{\eta_l}{n}\sum_{t=0}^{\mathcal{I}-1}\sum_{i\in\mathcal{S}_{t,r}^k}\boldsymbol{g}_{t,r}^i\right],$$
$$\mathbb{E}_{\mathcal{S}_r}[\bar{\Delta}_r] = \mathbb{E}_{\mathcal{S}_r}\left[\frac{1}{K}\sum_{k=1}^{K}\bar{\Delta}_r^k\right] = \frac{1}{K}\sum_{k=1}^{K}\mathbb{E}_{\mathcal{S}_r^k}\left[\mathbb{E}_{\mathcal{S}_{t,r}^k}\left[-\frac{\eta_l}{n}\sum_{t=0}^{\mathcal{I}-1}\sum_{i\in\mathcal{S}_{t,r}^k}\boldsymbol{g}_{t,r}^i\right]\right]$$
$$= \frac{1}{K}\sum_{k=1}^{K}\mathbb{E}_{\mathcal{S}_r^k}\left[-\frac{1}{m}\sum_{i\in\mathcal{S}_r^k}\frac{\eta_l}{n}\sum_{t=0}^{\mathcal{I}-1}\mathbb{E}_{\mathcal{S}_{t,r}^k}\left[\sum_{i\in\mathcal{S}_{t,r}^k}\boldsymbol{g}_{t,r}^i\right]\right] = \mathbb{E}_{\mathcal{S}_r}[\Delta_r]. \tag{12}$$

Thus $\bar{\Delta}_r$ is the unbiased estimate of $\Delta_r$.

*Proof of Theorem D.3.* Similar to previous works (Zhou et al., 2018; Chen et al., 2020), we introduce a Lyapunov sequence $\boldsymbol{z}_r$: assume $\boldsymbol{x}_0 = \boldsymbol{x}_1$, for each $r \geq 1$, we have

$$\boldsymbol{z}_r = \boldsymbol{x}_r + \frac{\beta_1}{1-\beta_1}(\boldsymbol{x}_r - \boldsymbol{x}_{r-1}) = \frac{1}{1-\beta_1}\boldsymbol{x}_r - \frac{\beta_1}{1-\beta_1}\boldsymbol{x}_{r-1}. \tag{13}$$

For the difference of two adjacent element in sequence $\boldsymbol{z}_r$, we have

$$
\begin{aligned}
\boldsymbol{z}_{r+1} - \boldsymbol{z}_r &= \frac{1}{\beta_1}(\boldsymbol{x}_{r+1} - \boldsymbol{x}_r) - \frac{\beta_1}{1-\beta_1}(\boldsymbol{x}_r - \boldsymbol{x}_{r-1}) \\
&= \frac{1}{1-\beta_1}(\eta \widehat{\boldsymbol{V}}_r^{-1/2} \boldsymbol{m}_r) - \frac{\beta_1}{1-\beta_1}\eta \widehat{\boldsymbol{V}}_{r-1}^{-1/2} \boldsymbol{m}_{r-1} \\
&= \frac{1}{1-\beta_1}\eta \widehat{\boldsymbol{V}}_r^{-1/2}\Big[\beta_1 \boldsymbol{m}_{r-1} + (1-\beta_1)\Delta_r\Big] - \frac{\beta_1}{1-\beta_1}\eta \widehat{\boldsymbol{V}}_{r-1}^{-1/2} \boldsymbol{m}_{r-1} \\
&= \eta \widehat{\boldsymbol{V}}_r^{-1/2}\Delta_r - \eta \frac{\beta_1}{1-\beta_1}\Big(\widehat{\boldsymbol{V}}_{r-1}^{-1/2} - \widehat{\boldsymbol{V}}_r^{-1/2}\Big)\boldsymbol{m}_{r-1}.
\end{aligned}
$$

By Assumption 4.1, with the property of $L$-smoothness, for $r \in [R]$, taking conditional expectation at global round $r$, we have

$$
\begin{aligned}
\mathbb{E}[f(\boldsymbol{z}_{r+1})] - f(\boldsymbol{z}_r) &\leq \mathbb{E}[\langle \nabla f(\boldsymbol{z}_r), \boldsymbol{z}_{r+1} - \boldsymbol{z}_r \rangle] + \frac{L}{2}\mathbb{E}[\|\boldsymbol{z}_{r+1} - \boldsymbol{z}_r\|^2] \\
&\leq \eta \mathbb{E}\Big[\Big\langle \nabla f(\boldsymbol{z}_r), \widehat{\boldsymbol{V}}_r^{-1/2}\Delta_r \Big\rangle\Big] - \eta \mathbb{E}\Big[\Big\langle \nabla f(\boldsymbol{z}_r), \frac{\beta_1}{1-\beta_1}\Big(\widehat{\boldsymbol{V}}_{r-1}^{-1/2} - \widehat{\boldsymbol{V}}_r^{-1/2}\Big)\boldsymbol{m}_{r-1} \Big\rangle\Big] \\
&\quad + \frac{\eta^2 L}{2}\mathbb{E}\Big[\Big\|\widehat{\boldsymbol{V}}_r^{-1/2}\Delta_r - \frac{\beta_1}{1-\beta_1}\Big(\widehat{\boldsymbol{V}}_{r-1}^{-1/2} - \widehat{\boldsymbol{V}}_r^{-1/2}\Big)\boldsymbol{m}_{r-1}\Big\|^2\Big] \\
&= \underbrace{\eta \mathbb{E}\Big[\Big\langle \nabla f(\boldsymbol{x}_r), \widehat{\boldsymbol{V}}_r^{-1/2}\Delta_r \Big\rangle\Big]}_{I} \underbrace{-\eta \mathbb{E}\Big[\Big\langle \nabla f(\boldsymbol{z}_r), \frac{\beta_1}{1-\beta_1}\Big(\widehat{\boldsymbol{V}}_{r-1}^{-1/2} - \widehat{\boldsymbol{V}}_r^{-1/2}\Big)\boldsymbol{m}_{r-1} \Big\rangle\Big]}_{II} \\
&\quad + \underbrace{\frac{\eta^2 L}{2}\mathbb{E}\Big[\Big\|\widehat{\boldsymbol{V}}_r^{-1/2}\Delta_r - \frac{\beta_1}{1-\beta_1}\Big(\widehat{\boldsymbol{V}}_{r-1}^{-1/2} - \widehat{\boldsymbol{V}}_r^{-1/2}\Big)\boldsymbol{m}_{r-1}\Big\|^2\Big]}_{III} \\
&\quad + \underbrace{\eta \mathbb{E}\Big[\Big\langle \nabla f(\boldsymbol{z}_r) - \nabla f(\boldsymbol{x}_r), \widehat{\boldsymbol{V}}_r^{-1/2}\Delta_r \Big\rangle\Big]}_{IV},
\end{aligned}
\tag{14}
$$

### E.1 Bounding $I$

We have

$$
\begin{aligned}
I &= \eta \mathbb{E}\Big[\Big\langle \nabla f(\boldsymbol{x}_r), \frac{\Delta_r}{\sqrt{\widehat{\boldsymbol{v}}_r} + \epsilon} \Big\rangle\Big] \\
&= \eta \mathbb{E}\Big[\Big\langle \frac{\nabla f(\boldsymbol{x}_r)}{\sqrt{\widehat{\boldsymbol{v}}_r} + \epsilon}, \bar{\Delta}_r \Big\rangle\Big] \\
&= -\eta \eta_l \mathbb{E}\Big[\Big\langle \frac{\nabla f(\boldsymbol{x}_r)}{\sqrt{\widehat{\boldsymbol{v}}_r} + \epsilon}, \frac{1}{Kn}\sum_{t=0}^{\mathcal{I}-1}\sum_{k\in[K]}\sum_{i\in\mathcal{V}_k}\bar{\boldsymbol{g}}_{r,t}^i \Big\rangle\Big] \\
&= -\eta \eta_l \mathbb{E}\Big[\Big\langle \frac{\nabla f(\boldsymbol{x}_r)}{\sqrt{\widehat{\boldsymbol{v}}_r} + \epsilon}, \frac{1}{Kn}\sum_{t=0}^{\mathcal{I}-1}\sum_{k\in[K]}\sum_{i\in\mathcal{S}_{r,t}^k}\boldsymbol{g}_{r,t}^i \Big\rangle\Big] \\
&= -\frac{\eta \eta_l m}{n}\sum_{t=0}^{\mathcal{I}-1}\mathbb{E}\Big[\Big\langle \frac{\nabla f(\boldsymbol{x}_r)}{\sqrt{\widehat{\boldsymbol{v}}_r} + \epsilon}, \frac{1}{N}\sum_{i=1}^{N}\boldsymbol{g}_{r,t}^i \Big\rangle\Big] \\
&= -\frac{\eta \eta_l m}{n}\sum_{t=0}^{\mathcal{I}-1}\mathbb{E}\Big[\Big\langle \frac{\nabla f(\boldsymbol{x}_r)}{\sqrt{\widehat{\boldsymbol{v}}_r} + \epsilon}, \frac{1}{N}\sum_{i=1}^{N}\nabla f_i(\boldsymbol{x}_{r,t}^i) \Big\rangle\Big],
\end{aligned}
\tag{15}
$$

where the first equation in equation 15 holds by $\Delta_r = \frac{m}{n}\bar{\Delta}_r = -\frac{m}{n} \cdot \frac{\eta_l}{N}\sum_{i=1}^{N}\sum_{t=0}^{\mathcal{I}-1}\boldsymbol{g}_{r,t}^i$. The last equation above holds by the unbiasedness of stochastic gradient, then we have

$$
-\mathbb{E}\left[\left\langle \frac{\nabla f(\boldsymbol{x}_r)}{\sqrt{\widehat{\boldsymbol{v}}_r}+\epsilon}, \frac{1}{N}\sum_{i=1}^{N}\nabla f_i(\boldsymbol{x}_{r,t}^i)\right\rangle\right]
$$

$$
= -\frac{1}{2}\mathbb{E}\left[\left\langle \frac{\nabla f(\boldsymbol{x}_r)}{\sqrt{\widehat{\boldsymbol{v}}_r}+\epsilon}, \frac{1}{N}\sum_{i=1}^{N}\nabla f_i(\boldsymbol{x}_{r,t}^i)\right\rangle\right]
$$

$$
- \frac{1}{2}\mathbb{E}\left[\left\langle \frac{\nabla f(\boldsymbol{x}_r)}{\sqrt{\widehat{\boldsymbol{v}}_r}+\epsilon}, \frac{1}{N}\sum_{i=1}^{N}\nabla f_i(\boldsymbol{x}_{r,t}^i) \pm \frac{1}{K}\sum_{k=1}^{K}\nabla \bar{f}_k(\bar{\boldsymbol{x}}_{r,t}^k)\right\rangle\right]
$$

$$
= -\frac{1}{2}\mathbb{E}\left[\left\langle \frac{\nabla f(\boldsymbol{x}_r)}{\sqrt[4]{\widehat{\boldsymbol{v}}_r}+\epsilon}, \frac{1}{\sqrt[4]{\widehat{\boldsymbol{v}}_r}+\epsilon}\frac{1}{N}\sum_{i=1}^{N}\nabla f_i(\boldsymbol{x}_{r,t}^i)\right\rangle\right]
$$

$$
- \frac{1}{2}\mathbb{E}\left[\left\langle \frac{\nabla f(\boldsymbol{x}_r)}{\sqrt[4]{\widehat{\boldsymbol{v}}_r}+\epsilon}, \frac{1}{\sqrt[4]{\widehat{\boldsymbol{v}}_r}+\epsilon}\left(\frac{1}{N}\sum_{i=1}^{N}\nabla f_i(\boldsymbol{x}_{r,t}^i) \pm \frac{1}{K}\sum_{k=1}^{K}\nabla \bar{f}_k(\bar{\boldsymbol{x}}_{r,t}^k)\right)\right\rangle\right]. \tag{16}
$$

Since we have inequalities, $\langle \boldsymbol{a}, \boldsymbol{b}\rangle = \|\boldsymbol{a}\|^2 + \|\boldsymbol{b}\|^2 - \|\boldsymbol{a}-\boldsymbol{b}\|^2$ and $\langle \boldsymbol{a}, \boldsymbol{b}\rangle \leq \frac{1}{2}\|\boldsymbol{a}\|^2 + \frac{1}{2}\|\boldsymbol{b}\|^2$, then we have

$$
-\mathbb{E}\left[\left\langle \frac{\nabla f(\boldsymbol{x}_r)}{\sqrt{\widehat{\boldsymbol{v}}_r}+\epsilon}, \frac{1}{N}\sum_{i=1}^{N}\nabla f_i(\boldsymbol{x}_{r,t}^i)\right\rangle\right]
$$

$$
\leq -\frac{1}{4}\mathbb{E}\left[\left\|\frac{\nabla f(\boldsymbol{x}_r)}{\sqrt[4]{\widehat{\boldsymbol{v}}_r}+\epsilon}\right\|^2 + \left\|\frac{1}{\sqrt[4]{\widehat{\boldsymbol{v}}_r}+\epsilon}\frac{1}{N}\sum_{i=1}^{N}\nabla f_i(\boldsymbol{x}_{r,t}^i)\right\|^2\right.
$$

$$
- \left\|\frac{1}{\sqrt[4]{\widehat{\boldsymbol{v}}_r}+\epsilon}\left(\nabla f(\boldsymbol{x}_r) - \frac{1}{N}\sum_{i=1}^{N}\nabla f_i(\boldsymbol{x}_{r,t}^i)\right)\right\|^2\right] - \frac{1}{4}\mathbb{E}\left[\left\|\frac{\nabla f(\boldsymbol{x}_r)}{\sqrt[4]{\widehat{\boldsymbol{v}}_r}+\epsilon}\right\|^2\right.
$$

$$
+ \left\|\frac{1}{\sqrt[4]{\widehat{\boldsymbol{v}}_r}+\epsilon}\frac{1}{K}\sum_{k=1}^{K}\nabla \bar{f}_k(\bar{\boldsymbol{x}}_{r,t}^k)\right\|^2 - \left\|\frac{1}{\sqrt[4]{\widehat{\boldsymbol{v}}_r}+\epsilon}\left(\nabla f(\boldsymbol{x}_r) - \frac{1}{K}\sum_{k=1}^{K}\nabla \bar{f}_k(\bar{\boldsymbol{x}}_{r,t}^k)\right)\right\|^2\right]
$$

$$
+ \frac{1}{4}\mathbb{E}\left[\left\|\frac{\nabla f(\boldsymbol{x}_r)}{\sqrt[4]{\widehat{\boldsymbol{v}}_r}+\epsilon}\right\|^2\right] + \frac{1}{4}\mathbb{E}\left[\left\|\frac{1}{\sqrt[4]{\widehat{\boldsymbol{v}}_r}+\epsilon}\left(\frac{1}{N}\sum_{i=1}^{N}\nabla f_i(\boldsymbol{x}_{r,t}^i) - \frac{1}{K}\sum_{k=1}^{K}\nabla \bar{f}_k(\bar{\boldsymbol{x}}_{r,t}^k)\right)\right\|^2\right]
$$

$$
\leq -\frac{1}{4C_0}\mathbb{E}[\|\nabla f(\boldsymbol{x}_r)\|^2] - \frac{1}{4C_0}\mathbb{E}\left[\left\|\frac{1}{N}\sum_{i=1}^{N}\nabla f_i(\boldsymbol{x}_{r,t}^i)\right\|^2\right] - \frac{1}{4C_0}\mathbb{E}\left[\left\|\frac{1}{K}\sum_{k=1}^{K}\nabla \bar{f}_k(\bar{\boldsymbol{x}}_{r,t}^k)\right\|^2\right]
$$

$$
+ \frac{1}{4\sqrt{\epsilon}}\mathbb{E}\left[\left\|\nabla f(\boldsymbol{x}_r) - \frac{1}{N}\sum_{i=1}^{N}\nabla f_i(\boldsymbol{x}_{r,t}^i)\right\|^2\right] + \frac{1}{4\sqrt{\epsilon}}\mathbb{E}\left[\left\|\nabla f(\boldsymbol{x}_r) - \frac{1}{K}\sum_{k=1}^{K}\nabla \bar{f}_k(\bar{\boldsymbol{x}}_{r,t}^k)\right\|^2\right]
$$

$$
+ \frac{1}{4\sqrt{\epsilon}}\mathbb{E}\left[\left\|\frac{1}{N}\sum_{i=1}^{N}\nabla f_i(\boldsymbol{x}_{r,t}^i) - \frac{1}{K}\sum_{k=1}^{K}\nabla \bar{f}_k(\bar{\boldsymbol{x}}_{r,t}^k)\right\|^2\right], \tag{17}
$$

where the second inequality holds by the property of variance $\widehat{\boldsymbol{v}}_r$: $\|\boldsymbol{x}\|^2 C_0^{-1} \leq \|\boldsymbol{x}\|^2(\widehat{\boldsymbol{v}}_r + \epsilon)^{-1/2} \leq \|\boldsymbol{x}\|^2\epsilon^{1/2}$ with $C_0 = \sqrt{\eta_l^2\mathcal{I}^2G^2 + \epsilon}$. After applying the property of $L$-smoothness, the last three terms above are highly related to bound the inter-cluster consensus error $\|\boldsymbol{x}_r - \bar{\boldsymbol{x}}_{r,t}^k\|$ and intra-cluster consensus error $\|\bar{\boldsymbol{x}}_{r,t}^k - \boldsymbol{x}_{r,t}^i\|$.

Thus by Assumption 4.1, we have the following result for bounding $I$

$$
I = \eta \mathbb{E}\left[\left\langle \nabla f(\boldsymbol{x}_r), \frac{\Delta_r}{\sqrt{\hat{\boldsymbol{v}}_r + \epsilon}} \right\rangle\right]
$$

$$
\leq \frac{\eta\eta_l}{4C_0}\frac{m}{n}\sum_{t=0}^{\mathcal{I}-1}\left[-\mathbb{E}[\|\nabla f(\boldsymbol{x}_r)\|^2] - \mathbb{E}\left[\left\|\frac{1}{N}\sum_{i=1}^{N}\nabla f_i(\boldsymbol{x}_{r,t}^i)\right\|^2 - \left\|\frac{1}{K}\sum_{k=1}^{K}\nabla \bar{f}_k(\bar{\boldsymbol{x}}_{r,t}^k)\right\|^2\right]\right]
$$

$$
+ \frac{\eta\eta_l}{4\sqrt{\epsilon}}\frac{m}{n}\sum_{t=0}^{\mathcal{I}-1}\left\{\mathbb{E}\left[\left\|\nabla f(\boldsymbol{x}_r) - \frac{1}{N}\sum_{i=1}^{N}\nabla f_i(\boldsymbol{x}_{r,t}^i)\right\|^2\right] + \mathbb{E}\left[\left\|\nabla f(\boldsymbol{x}_r) - \frac{1}{K}\sum_{k=1}^{K}\nabla \bar{f}_k(\bar{\boldsymbol{x}}_{r,t}^k)\right\|^2\right]\right.
$$

$$
\left. + \mathbb{E}\left[\left\|\frac{1}{N}\sum_{i=1}^{N}\nabla f_i(\boldsymbol{x}_{r,t}^i) - \frac{1}{K}\sum_{k=1}^{K}\nabla \bar{f}_k(\bar{\boldsymbol{x}}_{r,t}^k)\right\|^2\right]\right\}
$$

$$
\leq \frac{\eta\eta_l}{4C_0}\frac{m}{n}\sum_{t=0}^{\mathcal{I}-1}\left[-\mathbb{E}[\|\nabla f(\boldsymbol{x}_r)\|^2] - \mathbb{E}\left[\left\|\frac{1}{N}\sum_{i=1}^{N}\nabla f_i(\boldsymbol{x}_{r,t}^i)\right\|^2 - \left\|\frac{1}{K}\sum_{k=1}^{K}\nabla \bar{f}_k(\bar{\boldsymbol{x}}_{r,t}^k)\right\|^2\right]\right]
$$

$$
+ \frac{\eta\eta_l}{4\sqrt{\epsilon}}\frac{m}{n}\sum_{t=0}^{\mathcal{I}-1}\left\{\mathbb{E}\left[\left\|\nabla f(\boldsymbol{x}_r) \pm \frac{1}{K}\sum_{k=1}^{K}\nabla \bar{f}_k(\bar{\boldsymbol{x}}_{r,t}^k) - \frac{1}{N}\sum_{i=1}^{N}\nabla f_i(\boldsymbol{x}_{r,t}^i)\right\|^2\right]\right.
$$

$$
\left. + \mathbb{E}\left[\left\|\nabla f(\boldsymbol{x}_r) - \frac{1}{K}\sum_{k=1}^{K}\nabla \bar{f}_k(\bar{\boldsymbol{x}}_{r,t}^k)\right\|^2\right] + \mathbb{E}\left[\left\|\frac{1}{N}\sum_{i=1}^{N}\nabla f_i(\boldsymbol{x}_{r,t}^i) - \frac{1}{K}\sum_{k=1}^{K}\nabla \bar{f}_k(\bar{\boldsymbol{x}}_{r,t}^k)\right\|^2\right]\right\}
$$

$$
\leq \frac{\eta\eta_l}{4C_0}\frac{m}{n}\sum_{t=0}^{\mathcal{I}-1}\left[-\mathbb{E}[\|\nabla f(\boldsymbol{x}_r)\|^2] - \mathbb{E}\left[\left\|\frac{1}{N}\sum_{i=1}^{N}\nabla f_i(\boldsymbol{x}_{r,t}^i)\right\|^2 - \left\|\frac{1}{K}\sum_{k=1}^{K}\nabla \bar{f}_k(\bar{\boldsymbol{x}}_{r,t}^k)\right\|^2\right]\right]
$$

$$
+ \frac{\eta\eta_l}{4\sqrt{\epsilon}}\frac{m}{n}\sum_{t=0}^{\mathcal{I}-1}\left\{\frac{2L^2}{K}\sum_{k=1}^{K}\mathbb{E}[\|\boldsymbol{x}_r - \bar{\boldsymbol{x}}_{r,t}^k\|^2] + \frac{2L^2}{N}\sum_{k=1}^{K}\sum_{i\in\mathcal{V}_k}\mathbb{E}[\|\bar{\boldsymbol{x}}_{r,t}^k - \boldsymbol{x}_{r,t}^i\|^2]\right.
$$

$$
\left. + \frac{L^2}{K}\sum_{k=1}^{K}\mathbb{E}[\|\boldsymbol{x}_r - \bar{\boldsymbol{x}}_{r,t}^k\|^2] + \frac{L^2}{N}\sum_{k=1}^{K}\sum_{i\in\mathcal{V}_k}\mathbb{E}[\|\bar{\boldsymbol{x}}_{r,t}^k - \boldsymbol{x}_{r,t}^i\|^2]\right\}
$$

$$
\leq \frac{\eta\eta_l}{4C_0}\frac{m}{n}\sum_{t=0}^{\mathcal{I}-1}\left[-\mathbb{E}[\|\nabla f(\boldsymbol{x}_r)\|^2] - \mathbb{E}\left[\left\|\frac{1}{N}\sum_{i=1}^{N}\nabla f_i(\boldsymbol{x}_{r,t}^i)\right\|^2 - \left\|\frac{1}{K}\sum_{k=1}^{K}\nabla \bar{f}_k(\bar{\boldsymbol{x}}_{r,t}^k)\right\|^2\right]\right]
$$

$$
+ \frac{\eta\eta_l}{4\sqrt{\epsilon}}\frac{m}{n}\sum_{t=0}^{\mathcal{I}-1}\left\{\frac{3L^2}{K}\sum_{k=1}^{K}\mathbb{E}[\|\boldsymbol{x}_r - \bar{\boldsymbol{x}}_{r,t}^k\|^2] + \frac{3L^2}{N}\sum_{k=1}^{K}\sum_{i\in\mathcal{V}_k}\mathbb{E}[\|\bar{\boldsymbol{x}}_{r,t}^k - \boldsymbol{x}_{r,t}^i\|^2]\right\}
$$

$$
\leq \frac{\eta\eta_l}{4C_0}\frac{m}{n}\sum_{t=0}^{\mathcal{I}-1}\left[-\mathbb{E}[\|\nabla f(\boldsymbol{x}_r)\|^2] - \mathbb{E}\left[\left\|\frac{1}{N}\sum_{i=1}^{N}\nabla f_i(\boldsymbol{x}_{r,t}^i)\right\|^2 - \left\|\frac{1}{K}\sum_{k=1}^{K}\nabla \bar{f}_k(\bar{\boldsymbol{x}}_{r,t}^k)\right\|^2\right]\right]
$$

$$
+ \frac{\eta\eta_l}{4\sqrt{\epsilon}}\frac{m}{n}\left[3L^2\mathcal{I}^2C_1\eta_l^2(\mathcal{I} + \rho_{\max}^2 H_{\mathcal{I},\rho})(\alpha^2\mathbb{E}[\|\nabla f(\boldsymbol{x}_r)\|^2 + \sigma_c^2) + 3L^2\mathcal{I}^2C_1\rho_{\max}^2 H_{\mathcal{I},\rho}\eta_l^2\bar{\sigma}_l^2\right.
$$

$$
\left. + 3L^2\mathcal{I}^2C_1\eta_l^2\sigma^2\rho_{\max}^2 + 3L^2C_1\left(\mathcal{I}^2 + H_{\mathcal{I},\rho}^2\cdot\rho_{\max}^2\right)\eta_l^2\left(\frac{\sigma^2}{n} + \frac{m(n-m)}{n(n-1)}\mathcal{I}\bar{\sigma}_l^2\right)\right], \tag{18}
$$

where the last inequality holds by Lemma F.1 and F.2.

### E.2 Bounding $II$

Bounding $II$ mainly follows by the update rule and definition of virtual sequence $z_r$,

$$
\begin{aligned}
II &= -\eta\mathbb{E}\left[\left\langle \nabla f(z_r), \frac{\beta_1}{1-\beta_1}(\widehat{V}_{r-1}^{-1/2} - \widehat{V}_r^{-1/2})m_{r-1}\right\rangle\right] \\
&= -\eta\mathbb{E}\left[\left\langle \nabla f(z_r) - \nabla f(x_r) + \nabla f(x_r), \frac{\beta_1}{1-\beta_1}(\widehat{V}_{r-1}^{-1/2} - \widehat{V}_r^{-1/2})m_{r-1}\right\rangle\right] \\
&\leq \eta\mathbb{E}\left[\|\nabla f(x_r)\|\left\|\frac{\beta_1}{1-\beta_1}(\widehat{V}_{r-1}^{-1/2} - \widehat{V}_r^{-1/2})m_{r-1}\right\|\right] \\
&\quad + \eta L\mathbb{E}\left[\|z_r - x_r\|\left\|\frac{\beta_1}{1-\beta_1}(\widehat{V}_{r-1}^{-1/2} - \widehat{V}_r^{-1/2})m_{r-1}\right\|\right] \\
&= \eta\mathbb{E}\left[\|\nabla f(x_r)\|\left\|\frac{\beta_1}{1-\beta_1}(\widehat{V}_{r-1}^{-1/2} - \widehat{V}_r^{-1/2})m_{r-1}\right\|\right] \\
&\quad + \eta^2 L\mathbb{E}\left[\left\|\frac{1}{\sqrt{\widehat{v}_{r-1}+\epsilon}}\frac{\beta_1}{1-\beta_1}m_{r-1}\right\|\left\|\frac{\beta_1}{1-\beta_1}(\widehat{V}_{r-1}^{-1/2} - \widehat{V}_r^{-1/2})m_{r-1}\right\|\right] \\
&\leq \frac{\beta_1}{1-\beta_1}\frac{m}{n}\eta\eta_l\mathcal{I}G^2\mathbb{E}\left[\|\widehat{V}_{r-1}^{-1/2} - \widehat{V}_r^{-1/2}\|_1\right] + \frac{\beta_1^2}{(1-\beta_1)^2}\frac{m^2}{n^2}L\eta^2\eta_l^2\mathcal{I}^2G^2\epsilon^{-1/2}\mathbb{E}\left[\|\widehat{V}_{r-1}^{-1/2} - \widehat{V}_r^{-1/2}\|_1\right], \quad (19)
\end{aligned}
$$

where the first iequality holds by Assumption 4.1, and the last one holds by Assumption 4.3 and Lemma F.8 about bounding $\nabla f(x_r)$ and $m_r$.

### E.3 Bounding $III$

For bounding $III$, use the similar way for bounding $II$,

$$
\begin{aligned}
III &= \frac{\eta^2 L}{2}\mathbb{E}\left[\left\|\widehat{V}_r^{-1/2}\Delta_r + \frac{\beta_1}{1-\beta_1}(\widehat{V}_{r-1}^{-1/2} - \widehat{V}_r^{-1/2})m_{r-1}\right\|^2\right] \\
&\leq \eta^2 L\mathbb{E}\left[\|\widehat{V}_r^{-1/2}\Delta_r\|^2\right] + \eta^2 L\mathbb{E}\left[\left\|\frac{\beta_1}{1-\beta_1}(\widehat{V}_{r-1}^{-1/2} - \widehat{V}_r^{-1/2})m_{r-1}\right\|^2\right] \\
&\leq \frac{\eta^2 L}{\epsilon}\mathbb{E}[\|\Delta_r\|^2] + \frac{\beta_1^2}{(1-\beta_1)^2}\frac{m^2}{n^2}\eta^2\eta_l^2 L\mathcal{I}^2G^2\mathbb{E}\left[\|\widehat{V}_{r-1}^{-1/2} - \widehat{V}_r^{-1/2}\|^2\right], \quad (20)
\end{aligned}
$$

where the first inequality holds by Cauchy-Schwarz inequality, and the second one follows by Assumption 4.3 and Lemma F.8 about bounding $\nabla f(x_r)$ and $m_r$.

### E.4 Bounding $IV$

Similarly, we bound the last term in equation 14,

$$
\begin{aligned}
IV &= \mathbb{E}\left[\left\langle \nabla f(z_r) - \nabla f(x_r), \eta\widehat{V}_r^{-1/2}\Delta_r\right\rangle\right] \\
&\leq \mathbb{E}\left[\|\nabla f(z_r) - \nabla f(x_r)\|\left\|\eta\widehat{V}_r^{-1/2}\Delta_r\right\|\right] \\
&\leq L\mathbb{E}\left[\|z_r - x_r\|\left\|\eta\widehat{V}_r^{-1/2}\Delta_r\right\|\right] \\
&\leq \frac{\eta^2 L}{2}\mathbb{E}\left[\left\|\frac{\beta_1}{1-\beta_1}\widehat{V}_r^{-1/2}m_r\right\|^2\right] + \frac{\eta^2 L}{2}\mathbb{E}\left[\|\widehat{V}_r^{-1/2}\Delta_r\|^2\right] \\
&\leq \frac{\eta^2 L}{2\epsilon}\frac{\beta_1^2}{(1-\beta_1)^2}\mathbb{E}[\|m_r\|^2] + \frac{\eta^2 L}{2\epsilon}\mathbb{E}[\|\Delta_r\|^2], \quad (21)
\end{aligned}
$$

where the first inequality holds due to Young's inequality, and the second one follows from Assumption 4.1 and the definition of virtual sequence $\boldsymbol{z}_r$. By Lemma F.7, we have

$$\sum_{r=1}^{R} \mathbb{E}[\|\boldsymbol{m}_r\|^2] \leq \sum_{r=1}^{R} \mathbb{E}[\|\Delta_r\|^2]. \tag{22}$$

Therefore, the summation of $IV$ term is bounded by

$$\sum_{r=1}^{R} IV \leq \left( \frac{\eta^2 L}{2\epsilon} \frac{\beta_1^2}{(1-\beta_1)^2} + \frac{\eta^2 L}{2\epsilon} \right) \sum_{r=1}^{R} \mathbb{E}[\|\Delta_r\|^2]. \tag{23}$$

Merging $I$ to $IV$ together, we obtain the following result for bounding equation 14,

$$\mathbb{E}[f(\boldsymbol{z}_{r+1})] - f(\boldsymbol{z}_1)$$
$$\leq \frac{\eta\eta_l}{4C_0} \frac{m}{n} \left\{ -\mathcal{I}\mathbb{E}[\|\nabla f(\boldsymbol{x}_r)\|^2] - \sum_{t=0}^{\mathcal{I}-1} \mathbb{E}\left[\left\|\frac{1}{N}\sum_{i=1}^{N} \nabla f_i(\boldsymbol{x}_{r,t}^i)\right\|^2\right] - \sum_{t=0}^{\mathcal{I}-1} \mathbb{E}\left[\left\|\frac{1}{K}\sum_{k=1}^{K} \nabla \bar{f}_k(\bar{\boldsymbol{x}}_{r,t}^k)\right\|^2\right] \right\}$$
$$+ \frac{\eta\eta_l}{4\sqrt{\epsilon}} \frac{m}{n} \left[ 3L^2\mathcal{I}^2 C_1 \eta_l^2 (\mathcal{I} + \rho_{\max}^2 H_{\mathcal{I},\rho})(\alpha^2 \mathbb{E}[\|\nabla f(\boldsymbol{x}_r)\|^2 + \sigma_c^2) + 3L^2\mathcal{I}^2 C_1 \rho_{\max}^2 H_{\mathcal{I},\rho} \eta_l^2 \bar{\sigma}_l^2 \right.$$
$$\left. + 3L^2\mathcal{I}^2 C_1 \eta_l^2 \sigma^2 \rho_{\max}^2 + 3L^2 C_1 (\mathcal{I}^2 + H_{\mathcal{I},\rho}^2 \cdot \rho_{\max}^2) \eta_l^2 \left( \frac{\sigma^2}{n} + \frac{m(n-m)}{n(n-1)} \mathcal{I} \bar{\sigma}_l^2 \right) \right]$$
$$+ \left( \eta \frac{\beta_1}{1-\beta_1} \frac{m}{n} \eta_l \mathcal{I} G^2 + \eta^2 \frac{\beta_1^2}{(1-\beta_1)^2} \frac{m^2}{n^2} L \eta_l^2 \mathcal{I}^2 G^2 \epsilon^{-1/2} \right) \mathbb{E}[\|\widehat{\boldsymbol{V}}_{r-1}^{-1/2} - \widehat{\boldsymbol{V}}_r^{-1/2}\|_1]$$
$$+ \eta^2 L \frac{\beta_1^2}{(1-\beta_1)^2} \frac{m^2}{n^2} \eta_l^2 \mathcal{I}^2 G^2 \mathbb{E}[\|\widehat{\boldsymbol{V}}_{r-1}^{-1/2} - \widehat{\boldsymbol{V}}_r^{-1/2}\|^2]$$
$$+ \left( \frac{\eta^2 L}{2\epsilon} \frac{\beta_1^2}{(1-\beta_1)^2} + \frac{\eta^2 L}{2\epsilon} + \frac{\eta^2 L}{\epsilon} \right) \mathbb{E}[\|\Delta_r\|^2],$$

then substituting the bound of $\|\Delta_r\|^2$ in Lemma F.5, and by applying Lemma F.4, then we have

$$\frac{n}{m} \sum_{r=1}^{R} \left[ \mathbb{E}[f(\boldsymbol{z}_{r+1})] - f(\boldsymbol{z}_1) \right]$$
$$\leq -\frac{\eta\eta_l \mathcal{I}}{4C_0} \sum_{r=1}^{R} \mathbb{E}[\|\nabla f(\boldsymbol{x}_r)\|^2] - \frac{\eta\eta_l}{4C_0} \sum_{r=1}^{R} \sum_{t=0}^{\mathcal{I}-1} \mathbb{E}\left[\left\|\frac{1}{N}\sum_{i=1}^{N} \nabla f_i(\boldsymbol{x}_{r,t}^i)\right\|^2\right]$$
$$- \frac{\eta\eta_l}{4C_0} \sum_{r=1}^{R} \sum_{t=0}^{\mathcal{I}-1} \mathbb{E}\left[\left\|\frac{1}{K}\sum_{k=1}^{K} \nabla \bar{f}_k(\bar{\boldsymbol{x}}_{r,t}^k)\right\|^2\right]$$
$$+ \frac{\eta\eta_l}{4\sqrt{\epsilon}} \sum_{r=1}^{R} \left[ 3L^2\mathcal{I}^2 C_1 \eta_l^2 (\mathcal{I} + \rho_{\max}^2 H_{\mathcal{I},\rho})(\alpha^2 \mathbb{E}[\|\nabla f(\boldsymbol{x}_r)\|^2] + \sigma_c^2) + 3L^2\mathcal{I}^2 C_1 \rho_{\max}^2 H_{\mathcal{I},\rho} \eta_l^2 \bar{\sigma}_l^2 \right.$$
$$\left. + 3L^2\mathcal{I}^2 C_1 \eta_l^2 \sigma^2 \rho_{\max}^2 + 3L^2 C_1 (\mathcal{I}^2 + H_{\mathcal{I},\rho}^2 \cdot \rho_{\max}^2) \eta_l^2 \left( \frac{\sigma^2}{n} + \frac{m(n-m)}{n(n-1)} \mathcal{I} \bar{\sigma}_l^2 \right) \right]$$
$$+ \left( \eta \frac{\beta_1}{1-\beta_1} \eta_l \mathcal{I} G^2 + \eta^2 \frac{\beta_1^2}{(1-\beta_1)^2} \frac{m}{n} L \eta_l^2 \mathcal{I}^2 G^2 \epsilon^{-1/2} \right) \sum_{r=1}^{R} \mathbb{E}[\|\widehat{\boldsymbol{V}}_{r-1}^{-1/2} - \widehat{\boldsymbol{V}}_r^{-1/2}\|_1]$$
$$+ \eta^2 L \frac{\beta_1^2}{(1-\beta_1)^2} \frac{m}{n} \eta_l^2 \mathcal{I}^2 G^2 \sum_{r=1}^{R} \mathbb{E}[\|\widehat{\boldsymbol{V}}_{r-1}^{-1/2} - \widehat{\boldsymbol{V}}_r^{-1/2}\|^2]$$

$$
+ \left( \frac{\eta^2 L}{2\epsilon} \frac{\beta_1^2}{(1-\beta_1)^2} + \frac{\eta^2 L}{2\epsilon} + \frac{\eta^2 L}{\epsilon} \right) \sum_{r=1}^{R} \left\{ \frac{2\eta_l^2 \mathcal{I}}{N} \sigma^2 + \frac{64\eta_l^2 \mathcal{I}(n-m)}{n(n-1)} [\bar{\sigma}_l^2 + \sigma_c^2] \right.
$$

$$
+ \left( \frac{2n(n-m)}{m^2(n-1)} + \frac{64\eta_l^2 \mathcal{I} L^2 (n-m)}{n(n-1)} + 4\eta_l^2 (\mathcal{I}+1)^2 L^2 \right) \Big[ \mathcal{I} C_1 \eta_l^2 H_{\mathcal{I},\rho} \rho_{\max}^2 (\alpha^2 \mathbb{E}[\|\nabla f(\boldsymbol{x}_r)\|^2] + \sigma_c^2)
$$

$$
+ \mathcal{I} C_1 \eta_l^2 H_{\mathcal{I},\rho} \rho_{\max}^2 \bar{\sigma}_l^2 + \mathcal{I} C_1 \eta_l^2 \rho_{\max}^2 \sigma^2 + \mathcal{I} C_1 \eta_l^2 \frac{H_{\mathcal{I},\rho}^2}{\mathcal{I}^2} \cdot \rho_{\max}^2 \left( \frac{\sigma^2}{n} + \frac{m(n-m)}{n(n-1)} \mathcal{I} \bar{\sigma}_l^2 \right) \Big]
$$

$$
+ \frac{64\eta_l^2 L^2 (n-m)}{n(n-1)} \Big[ \mathcal{I}^3 C_1 \eta_l^2 (\alpha^2 \mathbb{E}[\|\nabla f(\boldsymbol{x}_r)\|^2] + \sigma_c^2) + \mathcal{I}^2 C_1 \eta_l^2 \left( \frac{\sigma^2}{n} + \frac{m(n-m)}{n(n-1)} \mathcal{I} \bar{\sigma}_l^2 \right) \Big]
$$

$$
+ \left. \frac{16\eta_l^2 (m-1)}{(n-1)} \sum_{t=0}^{\mathcal{I}-1} \mathbb{E}\left[ \left\| \frac{1}{N} \sum_{i=1}^{N} \nabla f_i(\boldsymbol{x}_{r,t}^i) \right\|^2 \right] + \frac{8\eta_l^2 \mathcal{I} m}{n} \sum_{t=0}^{\mathcal{I}-1} \mathbb{E}\left[ \left\| \frac{1}{K} \sum_{k=1}^{K} \nabla \bar{f}_k(\bar{\boldsymbol{x}}_{r,t}^k) \right\|^2 \right] \right\}. \tag{24}
$$

we need the certain constraint on local learning rate

$$
C_{\beta,\eta} \eta \left[ \frac{16\eta_l^2}{N^2} \frac{(m-1)}{(n-1)} + \frac{2(\mathcal{I}+2)\eta_l^2 m}{N^2 n} \right] \leq \frac{\eta \eta_l}{4C_0}
$$

$$
\Rightarrow \eta_l \leq \frac{1}{4C_0 C_{\beta,\eta}} \left[ \frac{4}{N^2} \frac{(m-1)}{(n-1)} + \frac{2(\mathcal{I}+2)m}{N^2 n} \right]^{-1}, \tag{25}
$$

where $C_{\beta,\eta} = \left( \frac{\eta L}{\epsilon} \frac{\beta_1^2}{(1-\beta_1)^2} + \frac{3\eta L}{\epsilon} \right) = \mathcal{O}(\max\{\eta, 1\})$, we further need the requirement of $\eta_l$, which is same as the requirement in full participation settings

$$
\frac{\eta \eta_l}{4\sqrt{\epsilon}} 3L^2 \mathcal{I}^2 C_1 \eta_l^2 (\mathcal{I} + \rho_{\max}^2 H_{\mathcal{I},\rho}) \alpha^2 + \left( \frac{2n(n-m)}{m^2(n-1)} + \frac{32\eta_l^2 L^2 (n-m)}{n(n-1)} \right) C_{\beta,\eta} \eta \mathcal{I}^2 C_1 \eta_l^2 H_{\mathcal{I},\rho} \rho_{\max}^2 \alpha^2
$$

$$
+ \frac{32\eta_l^2 L^2 (n-m)}{n(n-1)} C_{\beta,\eta} \eta \mathcal{I}^3 C_1 \eta_l^2 \alpha^2 \leq \frac{\eta \eta_l \mathcal{I}}{8C_0},
$$

$$
\Rightarrow \frac{1}{4\sqrt{\epsilon}} 3L^2 \mathcal{I} C_1 \eta_l^2 (\mathcal{I} + \rho_{\max}^2 H_{\mathcal{I},\rho}) \alpha^2 + \left( \frac{2n(n-m)}{m^2(n-1)} + \frac{32\eta_l^2 L^2 (n-m)}{n(n-1)} \right) C_{\beta,\eta} \mathcal{I} C_1 \eta_l H_{\mathcal{I},\rho} \rho_{\max}^2 \alpha^2
$$

$$
+ \frac{32\eta_l^2 L^2 (n-m)}{n(n-1)} C_{\beta,\eta} \mathcal{I}^2 C_1 \eta_l \alpha^2 \leq \frac{1}{8C_0}, \tag{26}
$$

$$
\Rightarrow \eta_l \leq \frac{\sqrt[4]{\epsilon}}{\sqrt{18 L^2 C_0 C_1 \mathcal{I}(\mathcal{I} + \rho_{\max}^2 H_{\mathcal{I},\rho}) \alpha^2}}, \tag{27}
$$

thus we have

$$
\frac{\eta \eta_l \mathcal{I}}{8C_0 R} \sum_{r=1}^{R} \mathbb{E}[\|\nabla f(\boldsymbol{x}_r)\|^2]
$$

$$
\leq \frac{n}{Rm} \big[ \mathbb{E}[f(\boldsymbol{z}_{r+1})] - f(\boldsymbol{z}_1) \big] + \frac{\eta \eta_l}{4R\sqrt{\epsilon}} \sum_{r=1}^{R} \Big[ 3L^2 \mathcal{I}^2 C_1 \eta_l^2 (\mathcal{I} + \rho_{\max}^2 H_{\mathcal{I},\rho}) \sigma_c^2 + 3L^2 \mathcal{I}^2 C_1 \rho_{\max}^2 H_{\mathcal{I},\rho} \eta_l^2 \bar{\sigma}_l^2
$$

$$
+ 3L^2 \mathcal{I}^2 C_1 \eta_l^2 \sigma^2 \rho_{\max}^2 + 3L^2 C_1 \big( \mathcal{I}^2 + H_{\mathcal{I},\rho}^2 \cdot \rho_{\max}^2 \big) \eta_l^2 \left( \frac{\sigma^2}{n} + \frac{m(n-m)}{n(n-1)} \mathcal{I} \bar{\sigma}_l^2 \right) \Big] + \frac{\beta_1}{1-\beta_1} \eta \eta_l \mathcal{I} G^2 \frac{d}{R\sqrt{\epsilon}}
$$

$$
+ 2\frac{\beta_1^2}{(1-\beta_1)^2} \frac{m}{n} \eta^2 \eta_l^2 L \mathcal{I}^2 G^2 \frac{d}{\epsilon} + \left( \frac{\eta^2 L}{2\epsilon} \frac{\beta_1^2}{(1-\beta_1)^2} + \frac{\eta^2 L}{2\epsilon} + \frac{\eta^2 L}{\epsilon} \right) \left\{ \frac{2\eta_l^2 \mathcal{I}}{N} \sigma^2 \right.
$$

$$
+ \left( \frac{2n(n-m)}{m^2(n-1)} + \frac{64\eta_l^2 \mathcal{I} L^2 (n-m)}{n(n-1)} + 4\eta_l^2 (\mathcal{I}+1)^2 L^2 \right) \Big[ \mathcal{I} C_1 \eta_l^2 H_{\mathcal{I},\rho} \rho_{\max}^2 \sigma_c^2
$$

$$
+ \mathcal{I} C_1 \eta_l^2 H_{\mathcal{I},\rho} \rho_{\max}^2 \bar{\sigma}_l^2 + \mathcal{I} C_1 \eta_l^2 \rho_{\max}^2 \sigma^2 + \mathcal{I} C_1 \eta_l^2 \frac{H_{\mathcal{I},\rho}^2}{\mathcal{I}^2} \cdot \rho_{\max}^2 \left( \frac{\sigma^2}{n} + \frac{m(n-m)}{n(n-1)} \mathcal{I} \bar{\sigma}_l^2 \right) \Big]
$$

$$
+ 64\eta_l^2 L^2 \frac{(n-m)}{n(n-1)} \Big[ \mathcal{I}^3 C_1 \eta_l^2 \sigma_c^2 + \mathcal{I}^2 C_1 \eta_l^2 \left( \frac{\sigma^2}{n} + \frac{m(n-m)}{n(n-1)} \mathcal{I} \bar{\sigma}_l^2 \right) \Big] + \frac{64\eta_l^2 \mathcal{I}(n-m)}{n(n-1)} [\bar{\sigma}_l^2 + \sigma_c^2] \right\} \tag{28}
$$

since there is $H_{\mathcal{I},\rho} = \min\{\frac{1}{1-\rho_{\max}}, \mathcal{I}\} \leq \mathcal{I}$ and $\rho_{\max} \leq 1$,

$$
\frac{1}{8C_0 R} \sum_{r=1}^{R} \mathbb{E}[\|\nabla f(\boldsymbol{x}_r)\|^2]
$$

$$
\leq \frac{n}{\eta\eta_l R \mathcal{I} m} \Big[ \mathbb{E}[f(\boldsymbol{z}_{r+1})] - f(\boldsymbol{z}_1) \Big] + \frac{1}{4\sqrt{\epsilon}} \Big[ C \cdot L^2 \eta_l^2 \mathcal{I}(\mathcal{I} + H_{\mathcal{I},\rho})\sigma_c^2
$$

$$
+ C \cdot L^2 \eta_l^2 \mathcal{I}\Big( \rho_{\max}^2 H_{\mathcal{I},\rho}\bar{\sigma}_l^2 + \frac{m(n-m)}{n(n-1)}\mathcal{I}\bar{\sigma}_l^2 + \sigma^2 \rho_{\max}^2 \Big(1 + \frac{1}{n}\Big)\Big) \Big] + C_\beta G^2 \frac{d}{R\sqrt{\epsilon}} + 2C_\beta^2 \frac{m}{n} L\eta\eta_l \mathcal{I} G^2 \frac{d}{\epsilon}
$$

$$
+ \Big( \frac{\eta L}{2\epsilon}\frac{\beta_1^2}{(1-\beta_1)^2} + \frac{\eta L}{2\epsilon} + \frac{\eta L}{\epsilon} \Big) \Big\{ \frac{2\eta_l}{N}\sigma^2 + \Big( \frac{2n(n-m)}{m^2(n-1)} + \frac{64\eta_l^2 \mathcal{I} L^2 (n-m)}{n(n-1)} + 4\eta_l^2 (\mathcal{I}+1)^2 L^2 \Big)
$$

$$
\cdot \Big[ C_1 \eta_l H_{\mathcal{I},\rho}\rho_{\max}^2 \sigma_c^2 + C_1 \eta_l H_{\mathcal{I},\rho}\rho_{\max}^2 \bar{\sigma}_l^2 + C_1 \eta_l \rho_{\max}^2 \sigma^2 + C_1 \eta_l \frac{H_{\mathcal{I},\rho}^2}{\mathcal{I}^2} \cdot \rho_{\max}^2 \Big( \frac{\sigma^2}{n} + \frac{m(n-m)}{n(n-1)}\mathcal{I}\bar{\sigma}_l^2 \Big) \Big]
$$

$$
+ \frac{64(n-m)}{n(n-1)} \Big[ C_1 L^2 \eta_l^3 \mathcal{I}^2 \sigma_c^2 + C_1 L^2 \eta_l^3 \mathcal{I}\Big( \frac{\sigma^2}{n} + \frac{m(n-m)}{n(n-1)}\mathcal{I}\bar{\sigma}_l^2 \Big) + \eta_l \bar{\sigma}_l^2 + \eta_l \sigma_c^2 \Big] \Big\}, \tag{29}
$$

where $C$ is a constant irrelevant to parameters and $\rho_{\max} = \max_{k\in[K]}\rho_k$, $H_{\mathcal{I},\rho} = \min\left\{\frac{1}{1-\rho_{\max}}, \mathcal{I}\right\}$, $C_\beta = \frac{\beta_1}{1-\beta_1}$ and $\bar{\sigma}_L^2 = \frac{1}{K}\sum_{k=1}^{K}\sigma_k^2$. This concludes the proof.

If we further apply the constraint of

$$
\frac{n(n-m)}{m^2(n-1)} H_{\mathcal{I},\rho}\rho_{\max}^2 \leq \frac{1}{n}, \tag{30}
$$

where the condition Eq. 30 implies that the spectral gap $\rho_{\max}$ satisfies

$$
\frac{\rho_{\max}^2}{1-\rho_{\max}} \leq \frac{m^2}{n^2}, \tag{31}
$$

with the condition of Eq. 31, then we assume there is $\rho_{\max} \leq \frac{m}{m+n}$, which satisfies

$$
\frac{n(n-m)}{m^2(n-1)} H_{\mathcal{I},\rho}\rho_{\max}^2 = \frac{n(n-m)}{m^2(n-1)}\frac{\rho_{\max}^2}{1-\rho_{\max}} \leq \frac{n(n-m)}{m^2(n-1)}\frac{m^2}{(m+n)n} \leq \frac{1}{n}. \tag{32}
$$

Also by choosing a constant $\widetilde{C}$, we have

$$\frac{1}{R}\sum_{r=1}^{R}\mathbb{E}[\|\nabla f(\boldsymbol{x}_r)\|^2]$$

$$\leq 8C_0\bigg\{\underbrace{\frac{n[\mathbb{E}[f(\boldsymbol{z}_{r+1})]-f(\boldsymbol{z}_0)]}{\eta\eta_l R\mathcal{I}m}+\frac{1}{R}\bigg(\frac{C_\beta G^2 d}{\sqrt{\epsilon}}+\frac{2C_\beta^2\eta\eta_l\mathcal{I}LG^2 dm}{\epsilon n}\bigg)}_{T_1}$$

$$+\underbrace{\frac{CL^2\eta_l^2}{4\sqrt{\epsilon}}\bigg[\mathcal{I}(\mathcal{I}+H_{\mathcal{I},\rho})\sigma_c^2+\bigg(\mathcal{I}\rho_{\max}^2 H_{\mathcal{I},\rho}\bar{\sigma}_l^2+\frac{n+1}{n}\mathcal{I}\sigma^2\rho_{\max}^2+\frac{m(n-m)}{n(n-1)}\mathcal{I}^2\bar{\sigma}_l^2\bigg)\bigg]}_{T_2}$$

$$+\underbrace{\frac{\eta_l}{N}\bigg(\frac{\eta L}{\epsilon}\frac{\beta_1^2}{(1-\beta_1)^2}+\frac{3\eta L}{\epsilon}\bigg)\sigma^2}_{T_3}+\underbrace{\frac{C_3\eta_l}{n}\bigg(\frac{\eta L}{\epsilon}\frac{\beta_1^2}{(1-\beta_1)^2}+\frac{3\eta L}{\epsilon}\bigg)[\bar{\sigma}_l^2+\sigma_c^2]}_{T_4}$$

$$+\underbrace{\frac{C_2\eta_l}{n}\bigg(\frac{\eta L}{\epsilon}\frac{\beta_1^2}{(1-\beta_1)^2}+\frac{3\eta L}{\epsilon}\bigg)\bigg[\sigma_c^2+\bar{\sigma}_l^2+\sigma^2+H_{\mathcal{I},\rho}\frac{\sigma^2}{\mathcal{I}^2 n}+\frac{m(n-m)}{n(n-1)}\frac{H_{\mathcal{I},\rho}}{\mathcal{I}}\bar{\sigma}_l^2\bigg]}_{T_5}$$

$$+\underbrace{\frac{C_3L^2\eta_l^3\mathcal{I}}{n}\bigg(\frac{\eta L}{\epsilon}\frac{\beta_1^2}{(1-\beta_1)^2}+\frac{3\eta L}{\epsilon}\bigg)}_{T_6}$$

$$\cdot\underbrace{\bigg[(\mathcal{I}+H_{\mathcal{I},\rho}\rho_{\max}^2)\sigma_c^2+H_{\mathcal{I},\rho}\rho_{\max}^2\bar{\sigma}_l^2+\frac{n+1}{n}\rho_{\max}^2\sigma^2+\frac{\sigma^2}{n}+\frac{m(n-m)}{n(n-1)}\mathcal{I}\bar{\sigma}_l^2\bigg]}_{T_7}$$

$$+\underbrace{\frac{C_4(\mathcal{I}+1)^2\eta_l^3 m}{n}\bigg(\frac{\eta L}{\epsilon}\frac{\beta_1^2}{(1-\beta_1)^2}+\frac{3\eta L}{\epsilon}\bigg)\bigg[\sigma_c^2+\bar{\sigma}_l^2+\sigma^2+H_{\mathcal{I},\rho}\frac{\sigma^2}{\mathcal{I}^2 n}+\frac{m(n-m)}{n(n-1)}\frac{H_{\mathcal{I},\rho}}{\mathcal{I}}\bar{\sigma}_l^2\bigg]}_{T_8} \tag{33}$$

By adopting learning rates $\eta=\Theta(n\sqrt{\frac{\mathcal{I}}{m}})$ and $\eta_l=\Theta(\frac{1}{\sqrt{R\mathcal{I}}})$, then we have

$$T_1=\frac{n[\mathbb{E}[f(\boldsymbol{z}_{r+1})]-f(\boldsymbol{z}_0)]}{\eta\eta_l R\mathcal{I}m}+\frac{1}{R}\bigg(\frac{C_\beta G^2 d}{\sqrt{\epsilon}}+\frac{2C_\beta^2\eta\eta_l\mathcal{I}LG^2 dm}{\epsilon n}\bigg)$$

$$=\frac{\mathbb{E}[f(\boldsymbol{z}_{r+1})]-f(\boldsymbol{z}_0)}{\sqrt{R\mathcal{I}m}}+\frac{1}{R}\bigg(\frac{C_\beta G^2 d}{\sqrt{\epsilon}}+\frac{2C_\beta^2 LG^2 d\sqrt{\mathcal{I}m}}{\epsilon\sqrt{R}}\bigg)$$

$$=\mathcal{O}\bigg(\frac{\mathbb{E}[f(\boldsymbol{z}_{r+1})]-f(\boldsymbol{z}_0)}{\sqrt{R\mathcal{I}m}}+\frac{G^2}{R}\bigg)+\widetilde{\mathcal{O}}\bigg(\frac{1}{R^{3/2}}\bigg).$$

$$T_2=\frac{CL^2\eta_l^2}{4\sqrt{\epsilon}}\bigg[\mathcal{I}(\mathcal{I}+H_{\mathcal{I},\rho})\sigma_c^2+\bigg(\mathcal{I}\rho_{\max}^2 H_{\mathcal{I},\rho}\bar{\sigma}_l^2+\frac{n+1}{n}\mathcal{I}\sigma^2\rho_{\max}^2+\frac{m(n-m)}{n(n-1)}\mathcal{I}^2\bar{\sigma}_l^2\bigg)\bigg]$$

$$=\frac{CL^2}{4R\mathcal{I}^2\sqrt{\epsilon}}\bigg[\mathcal{I}(\mathcal{I}+H_{\mathcal{I},\rho})\sigma_c^2+\bigg(\mathcal{I}\rho_{\max}^2 H_{\mathcal{I},\rho}\bar{\sigma}_l^2+\frac{n+1}{n}\mathcal{I}\sigma^2\rho_{\max}^2+\frac{m(n-m)}{n(n-1)}\mathcal{I}^2\bar{\sigma}_l^2\bigg)\bigg]$$

$$=\frac{CL^2}{4R\mathcal{I}\sqrt{\epsilon}}\bigg[(\mathcal{I}+H_{\mathcal{I},\rho})\sigma_c^2+\bigg(\rho_{\max}^2 H_{\mathcal{I},\rho}\bar{\sigma}_l^2+\frac{n+1}{n}\sigma^2\rho_{\max}^2+\frac{m(n-m)}{n(n-1)}\mathcal{I}\bar{\sigma}_l^2\bigg)\bigg]$$

$$\leq\frac{CL^2}{4R\mathcal{I}\sqrt{\epsilon}}\bigg[2\mathcal{I}\sigma_c^2+\bigg(\rho_{\max}^2\mathcal{I}\bar{\sigma}_l^2+\frac{n+1}{n}\sigma^2\rho_{\max}^2+\frac{m}{n}\mathcal{I}\bar{\sigma}_l^2\bigg)\bigg]$$

$$=\mathcal{O}\bigg(\frac{1}{R}\bigg[L^2[\sigma_c^2+(1+\rho_{\max}^2)\bar{\sigma}_l^2]+\frac{\rho_{\max}^2\sigma^2}{\mathcal{I}}\bigg]\bigg).$$

where the inequality holds by the definition of $H_{\mathcal{I},\rho} = \min\left\{\frac{1}{1-\rho_{\max}}, \mathcal{I}\right\}$, and the big-$\mathcal{O}$ holds because $m \leq n$.

$$
\begin{aligned}
T_3 &= \frac{\eta_l}{N}\left(\frac{\eta L}{\epsilon}\frac{\beta_1^2}{(1-\beta_1)^2} + \frac{3\eta L}{\epsilon}\right)\sigma^2 = \frac{1}{N}\left(\frac{n}{\sqrt{R\mathcal{I}m}}\frac{L}{\epsilon}\frac{\beta_1^2}{(1-\beta_1)^2} + \frac{n}{\sqrt{R\mathcal{I}m}}\frac{3L}{\epsilon}\right)\sigma^2 \\
&= \mathcal{O}\left(\frac{nL}{N\sqrt{R\mathcal{I}m}}\sigma^2\right) = \mathcal{O}\left(\frac{L}{K\sqrt{R\mathcal{I}m}}\sigma^2\right).
\end{aligned}
\tag{34}
$$

The simplification of $T_4$ is very similar to that of $T_3$,

$$
T_4 = \frac{C_3\eta_l}{n}\left(\frac{\eta L}{\epsilon}\frac{\beta_1^2}{(1-\beta_1)^2} + \frac{3\eta L}{\epsilon}\right)[\bar{\sigma}_l^2 + \sigma_c^2] = \mathcal{O}\left(\frac{L}{\sqrt{R\mathcal{I}m}}[\bar{\sigma}_l^2 + \sigma_c^2]\right).
$$

The simplification of the first half of $T_5$ is very similar to that of $T_3$ and $T_4$,

$$
\begin{aligned}
T_5 &= \frac{C_2\eta_l}{n}\left(\frac{\eta L}{\epsilon}\frac{\beta_1^2}{(1-\beta_1)^2} + \frac{3\eta L}{\epsilon}\right)\left[\sigma_c^2 + \bar{\sigma}_l^2 + \sigma^2 + H_{\mathcal{I},\rho}\frac{\sigma^2}{\mathcal{I}^2 n} + \frac{m(n-m)}{n(n-1)}\frac{H_{\mathcal{I},\rho}}{\mathcal{I}}\bar{\sigma}_l^2\right] \\
&= \frac{C_2}{n}\left(\frac{n}{\sqrt{R\mathcal{I}m}}\frac{L}{\epsilon}\frac{\beta_1^2}{(1-\beta_1)^2} + \frac{n}{\sqrt{R\mathcal{I}m}}\frac{3L}{\epsilon}\right)\cdot\left[\sigma_c^2 + \bar{\sigma}_l^2 + \sigma^2 + H_{\mathcal{I},\rho}\frac{\sigma^2}{\mathcal{I}^2 n} + \frac{m(n-m)}{n(n-1)}\frac{H_{\mathcal{I},\rho}}{\mathcal{I}}\bar{\sigma}_l^2\right] \\
&\leq C_2\left(\frac{1}{\sqrt{R\mathcal{I}m}}\frac{L}{\epsilon}\frac{\beta_1^2}{(1-\beta_1)^2} + \frac{1}{\sqrt{R\mathcal{I}m}}\frac{3L}{\epsilon}\right)\cdot\left[\sigma_c^2 + \bar{\sigma}_l^2 + \sigma^2 + \frac{\sigma^2}{\mathcal{I}n} + \bar{\sigma}_l^2\right] \\
&= \mathcal{O}\left(\frac{L}{\sqrt{R\mathcal{I}m}}[\sigma_c^2 + \bar{\sigma}_l^2 + \sigma^2]\right),
\end{aligned}
$$

The simplification of $T_6$ is very similar to that of $T_3$ and $T_4$ as well, so that

$$
\begin{aligned}
T_6 &= \frac{C_3 L^2 \eta_l^3 \mathcal{I}}{n}\left(\frac{\eta L}{\epsilon}\frac{\beta_1^2}{(1-\beta_1)^2} + \frac{3\eta L}{\epsilon}\right) \\
&= \mathcal{O}\left(\frac{1}{R\mathcal{I}}\cdot\frac{L}{\sqrt{R\mathcal{I}m}}[\bar{\sigma}_l^2 + \sigma_c^2]L^2\right) = \tilde{\mathcal{O}}\left(\frac{1}{R^{3/2}}\right).
\end{aligned}
$$

We bound $T_6 \cdot T_7$ together. since we previously have the bound for $T_6$, then

$$
\begin{aligned}
T_6 \cdot T_7 &= T_6 \cdot\left[(\mathcal{I} + H_{\mathcal{I},\rho}\rho_{\max}^2)\sigma_c^2 + H_{\mathcal{I},\rho}\rho_{\max}^2\bar{\sigma}_l^2 + \frac{n+1}{n}\rho_{\max}^2\sigma^2 + \frac{\sigma^2}{n} + \frac{m(n-m)}{n(n-1)}\mathcal{I}\bar{\sigma}_l^2\right] \\
&\leq T_6 \cdot\left[2\mathcal{I}\sigma_c^2 + \mathcal{I}\bar{\sigma}_l^2 + \frac{n+1}{n}\sigma^2 + \frac{\sigma^2}{n} + \mathcal{I}\bar{\sigma}_l^2\right] \\
&= \tilde{\mathcal{O}}\left(\frac{1}{R^{3/2}}\right).
\end{aligned}
$$

The simplification of the first half of $T_8$ is very similar to that of $T_3$ and $T_4$,

$$
\begin{aligned}
T_8 &= \frac{C_4(\mathcal{I}+1)^2\eta_l^3 m}{n}\left(\frac{\eta L}{\epsilon}\frac{\beta_1^2}{(1-\beta_1)^2} + \frac{3\eta L}{\epsilon}\right)\left[\sigma_c^2 + \bar{\sigma}_l^2 + \sigma^2 + H_{\mathcal{I},\rho}\frac{\sigma^2}{\mathcal{I}^2 n} + \frac{m(n-m)}{n(n-1)}\frac{H_{\mathcal{I},\rho}}{\mathcal{I}}\bar{\sigma}_l^2\right] \\
&= \frac{C_4(\mathcal{I}+1)^2\eta_l^2 m}{n}\left(\frac{\eta\eta_l L}{\epsilon}\frac{\beta_1^2}{(1-\beta_1)^2} + \frac{3\eta\eta_l L}{\epsilon}\right)\left[\sigma_c^2 + \bar{\sigma}_l^2 + \sigma^2 + H_{\mathcal{I},\rho}\frac{\sigma^2}{\mathcal{I}^2 n} + \frac{m(n-m)}{n(n-1)}\frac{H_{\mathcal{I},\rho}}{\mathcal{I}}\bar{\sigma}_l^2\right] \\
&= \frac{C_4(\mathcal{I}+1)^2 m}{R\mathcal{I}^2 n}\left(\frac{n}{\sqrt{R\mathcal{I}m}}\frac{L}{\epsilon}\frac{\beta_1^2}{(1-\beta_1)^2} + \frac{n}{\sqrt{R\mathcal{I}m}}\frac{3L}{\epsilon}\right)\cdot\left[\sigma_c^2 + \bar{\sigma}_l^2 + \sigma^2 + H_{\mathcal{I},\rho}\frac{\sigma^2}{\mathcal{I}^2 n} + \frac{m(n-m)}{n(n-1)}\frac{H_{\mathcal{I},\rho}}{\mathcal{I}}\bar{\sigma}_l^2\right] \\
&\leq \frac{C_4(\mathcal{I}+1)^2 m}{R\mathcal{I}^2 n}\left(\frac{n}{\sqrt{R\mathcal{I}m}}\frac{L}{\epsilon}\frac{\beta_1^2}{(1-\beta_1)^2} + \frac{n}{\sqrt{R\mathcal{I}m}}\frac{3L}{\epsilon}\right)\cdot\left[\sigma_c^2 + \bar{\sigma}_l^2 + \sigma^2 + \frac{\sigma^2}{\mathcal{I}n} + \bar{\sigma}_l^2\right] \\
&= \tilde{\mathcal{O}}\left(\frac{1}{R^{3/2}}\right).
\end{aligned}
$$

Therefore, merging terms from $T_1$ to $T_8$, equation 33 should be

$$\frac{1}{R}\sum_{r=1}^{R}\mathbb{E}[\|\nabla f(\boldsymbol{x}_r)\|^2] = T_1 + T_2 + T_3 + T_4 + T_5 + T_6 \cdot T_7 + T_8$$

$$= \underbrace{\mathcal{O}\left(\frac{\mathbb{E}[f(\boldsymbol{z}_{r+1})] - f(\boldsymbol{z}_0)}{\sqrt{R\mathcal{I}m}} + \frac{G^2}{R}\right)}_{T_1} + \underbrace{\mathcal{O}\left(\frac{1}{R}\left[L^2[\sigma_c^2 + (1+\rho_{\max}^2)\bar{\sigma}_l^2] + \frac{\rho_{\max}^2\sigma^2}{\mathcal{I}}\right]\right)}_{T_2}$$

$$+ \underbrace{\mathcal{O}\left(\frac{L}{K\sqrt{R\mathcal{I}m}}\sigma^2\right)}_{T_3} + \underbrace{\mathcal{O}\left(\frac{L}{\sqrt{R\mathcal{I}m}}[\bar{\sigma}_l^2 + \sigma_c^2]\right)}_{T_4} + \underbrace{\mathcal{O}\left(\frac{L}{\sqrt{R\mathcal{I}m}}[\sigma_c^2 + \bar{\sigma}_l^2 + \sigma^2]\right)}_{T_5} + \underbrace{\tilde{\mathcal{O}}\left(\frac{1}{R^{3/2}}\right)}_{T_1, T_6 \cdot T_7, T_8}, \qquad (35)$$

note that $N = Kn$, re-organizing them, we have

$$\frac{1}{R}\sum_{r=1}^{R}\mathbb{E}[\|\nabla f(\boldsymbol{x}_r)\|^2] = \mathcal{O}\left(\frac{1}{\sqrt{R\mathcal{I}m}}\left[[f_0 - f_*] + \left([\sigma^2 + \bar{\sigma}_l^2 + \sigma_c^2] + \frac{\sigma^2}{K}\right)L\right]\right)$$

$$+ \mathcal{O}\left(\frac{1}{R}\left[G^2 + L^2[\sigma_c^2 + (1+\rho_{\max}^2)\bar{\sigma}_l^2] + \frac{\rho_{\max}^2\sigma^2}{\mathcal{I}}\right]\right) + \tilde{\mathcal{O}}\left(\frac{1}{R^{3/2}}\right). \qquad (36)$$

Based on the theoretical analysis above, we obtain the general convergence bound for (C)AFGA. Here we conclude the proof for Theorem D.3. □

*Proof of Theorem 4.6.* Note that the proof for AFGA is a special case of the proof for CAFGA. When there is only one cluster containing all clients $[N]$ in the CAFGA framework, it reduces to AFGA. Therefore, for the convergence analysis of AFGA, we replace the Assumption D.1 with Assumption 4.4, and replace Assumption D.2 with Assumption 4.5. This results in $m = M, n = N, K = 1, \rho_{\max} = \rho, \sigma_c = 0$ and $\sigma_g = \sigma_k = \bar{\sigma}_l$, and then we have

$$\frac{1}{R}\sum_{r=1}^{R}\mathbb{E}[\|\nabla f(\boldsymbol{x}_r)\|^2]$$

$$= \mathcal{O}\left(\frac{1}{\sqrt{R\mathcal{I}M}}\left[[f_0 - f_*] + L[\sigma^2 + \sigma_g^2]\right]\right) + \mathcal{O}\left(\frac{1}{R}\left[G^2 + L^2(1+\rho^2)\sigma_g^2 + \frac{\rho^2\sigma^2}{\mathcal{I}}\right]\right) + \tilde{\mathcal{O}}\left(\frac{1}{R^{3/2}}\right). \qquad (37)$$

hence we conclude the proof for Theorem 4.6. □

# F Supporting Lemmas

**Lemma F.1** (Inter-cluster consensus error). *For local learning rate which satisfying the condition $\eta_l \leq \frac{1}{8\mathcal{I}L}$, denote $C_{\mathcal{I}} = 1 + \frac{3}{2} \cdot \frac{1}{4\mathcal{I}-1}$, recall the definition for $\bar{\boldsymbol{x}}$ in Eq. 9, the inter-cluster model difference after $s$ local steps satisfies*

$$\frac{1}{K}\sum_{k=1}^{K}\mathbb{E}[\|\bar{\boldsymbol{x}}_{r,t+1}^k - \boldsymbol{x}_r\|^2] \leq C_{\mathcal{I}}\frac{1}{K}\sum_{k=1}^{K}\mathbb{E}[\|\bar{\boldsymbol{x}}_{r,t}^k - \boldsymbol{x}_r\|^2] + 8\mathcal{I}\eta_l^2(\alpha^2\mathbb{E}[\|\nabla f(\boldsymbol{x}_r)\|^2] + \sigma_c^2) + \eta_l^2\frac{\sigma^2}{n}. \qquad (38)$$

*Proof.* Note that the following proof is similar to Lemma 3 in (Reddi et al., 2020). By definition and auxiliary sequences, we have

$$\mathbb{E}[\|\bar{\boldsymbol{x}}_{r,t+1}^k - \boldsymbol{x}_r\|^2] = \mathbb{E}[\|\bar{\boldsymbol{x}}_{r,t}^k - \boldsymbol{x}_r - \eta_l \bar{\boldsymbol{g}}_{r,t}^k\|^2]$$

$$= \mathbb{E}\left[\left\|\bar{\boldsymbol{x}}_{r,t}^k - \boldsymbol{x}_r - \eta_l \left(\bar{\boldsymbol{g}}_{r,t}^k \mp \frac{1}{n}\sum_{i\in\mathcal{S}_{r,t}^k}\nabla f_i(\boldsymbol{x}_{r,t}^i) \mp \frac{m}{n}\nabla\bar{f}_k(\bar{\boldsymbol{x}}_{r,t}^k) \mp \frac{m}{n}\nabla\bar{f}_k(\boldsymbol{x}_r)\right)\right\|^2\right]$$

$$\leq (1+\gamma)\mathbb{E}[\|\bar{\boldsymbol{x}}_{r,t}^k - \boldsymbol{x}_r\|^2] + \eta_l^2\mathbb{E}\left[\left\|\bar{\boldsymbol{g}}_{r,t}^k - \frac{1}{n}\sum_{i\in\mathcal{S}_{r,t}^k}\nabla f_i(\boldsymbol{x}_{r,t}^i)\right\|^2\right] + 4(1+\gamma^{-1})\eta_l^2\mathbb{E}\left[\left\|\frac{1}{n}\sum_{i\in\mathcal{S}_{r,t}^k}[\nabla f_i(\boldsymbol{x}_{r,t}^i) - \nabla f_i(\bar{\boldsymbol{x}}_{r,t}^k)]\right\|^2\right]$$

$$+ 4(1+\gamma^{-1})\eta_l^2\mathbb{E}\left[\left\|\frac{1}{n}\sum_{i\in\mathcal{S}_{r,t}^k}\nabla f_i(\bar{\boldsymbol{x}}_{r,t}^k) - \frac{m}{n}\nabla\bar{f}_k(\bar{\boldsymbol{x}}_{r,t}^k)\right\|^2\right] + 4(1+\gamma^{-1})\eta_l^2\mathbb{E}[\|\nabla\bar{f}_k(\bar{\boldsymbol{x}}_{r,t}^k) - \nabla\bar{f}_k(\boldsymbol{x}_r)\|^2]$$

$$+ 4(1+\gamma^{-1})\eta_l^2\mathbb{E}[\|\nabla\bar{f}_k(\boldsymbol{x}_r)\|^2]$$

$$\leq (1+\gamma)\mathbb{E}[\|\bar{\boldsymbol{x}}_{r,t}^k - \boldsymbol{x}_r\|^2] + \eta_l^2\frac{\sigma^2}{n} + 4(1+\gamma^{-1})\eta_l^2 L^2\mathbb{E}[\|\bar{\boldsymbol{x}}_{r,t}^k - \boldsymbol{x}_r\|^2] + 4(1+\gamma^{-1})\eta_l^2 L^2\frac{1}{n}\sum_{i=1}^n\mathbb{E}[\|\boldsymbol{x}_{r,t}^i - \bar{\boldsymbol{x}}_{r,t}^k\|^2]$$

$$+ 4(1+\gamma^{-1})\eta_l^2\mathbb{E}[\|\nabla\bar{f}_k(\boldsymbol{x}_r)\|^2] + \frac{m(n-m)}{n(n-1)}4(1+\gamma^{-1})\eta_l^2\sigma_k^2$$

$$\leq [(1+\gamma) + 4(1+\gamma^{-1})\eta_l^2 L^2]\cdot\mathbb{E}[\|\bar{\boldsymbol{x}}_{r,t}^k - \boldsymbol{x}_r\|^2] + 4(1+\gamma^{-1})\eta_l^2 L^2\frac{1}{n}\sum_{i=1}^n\mathbb{E}[\|\boldsymbol{x}_{r,t}^i - \bar{\boldsymbol{x}}_{r,t}^k\|^2]$$

$$+ \eta_l^2\frac{\sigma^2}{n} + 4(1+\gamma^{-1})\eta_l^2\mathbb{E}[\|\nabla\bar{f}_k(\boldsymbol{x}_r)\|^2] + \frac{m(n-m)}{n(n-1)}4(1+\gamma^{-1})\eta_l^2\sigma_k^2, \tag{39}$$

where the first equality holds by Eq. 10. The first inequality holds due to $\boldsymbol{g}_{r,t}^i$ is an unbiased estimator of $\nabla f_i(\boldsymbol{x}_{r,t}^i)$ and Young's inequality. The second inequality holds by Assumption 4.1 and 4.2, also the property of sampling in the cluster (see details in Lemma F.6), i.e.,

$$\mathbb{E}\left[\left\|\frac{1}{n}\sum_{i\in\mathcal{S}_{r,t}^k}\nabla f_i(\bar{\boldsymbol{x}}_{r,t}^k) - \frac{m}{n}\nabla\bar{f}_k(\bar{\boldsymbol{x}}_{r,t}^k)\right\|^2\right] = \mathbb{E}\left[\left\|\frac{1}{n}\sum_{i\in\mathcal{V}^k}\mathbb{I}\{i\in\mathcal{S}_{r,t}^k\}[\nabla f_i(\bar{\boldsymbol{x}}_{r,t}^k) - \nabla\bar{f}_k(\bar{\boldsymbol{x}}_{r,t}^k)]\right\|^2\right]$$

$$= \frac{m(n-m)}{n(n-1)}\frac{1}{n^2}\sum_{i\in\mathcal{V}^k}\mathbb{E}[\|\nabla f_i(\bar{\boldsymbol{x}}_{r,t}^k) - \nabla\bar{f}_k(\bar{\boldsymbol{x}}_{r,t}^k)\|^2] + \frac{m(m-1)}{n(n-1)}\mathbb{E}\left[\left\|\frac{1}{n}\sum_{i\in\mathcal{V}^k}\nabla f_i(\bar{\boldsymbol{x}}_{r,t}^k) - \nabla\bar{f}_k(\bar{\boldsymbol{x}}_{r,t}^k)\right\|^2\right]$$

$$\leq \frac{m(n-m)}{n(n-1)}\sigma_k^2.$$

Averaging Eq. 39 over $k = 1, ..., K$ clusters, we have

$$\frac{1}{K}\sum_{k=1}^K\mathbb{E}[\|\bar{\boldsymbol{x}}_{r,t+1}^k - \boldsymbol{x}_r\|^2]$$

$$\leq [(1+\gamma) + 4(1+\gamma^{-1})\eta_l^2 L^2]\frac{1}{K}\sum_{k=1}^K\mathbb{E}[\|\bar{\boldsymbol{x}}_{r,t}^k - \boldsymbol{x}_r\|^2] + 4(1+\gamma^{-1})\eta_l^2 L^2\frac{1}{N}\sum_{k=1}^K\sum_{i=1}^n\mathbb{E}[\|\boldsymbol{x}_{r,t}^i - \bar{\boldsymbol{x}}_{r,t}^k\|^2]$$

$$+ 4(1+\gamma^{-1})\eta_l^2\frac{1}{K}\sum_{k=1}^K\mathbb{E}[\|\nabla\bar{f}_k(\boldsymbol{x}_r)\|^2] + \eta_l^2\frac{\sigma^2}{n} + \frac{m(n-m)}{n(n-1)}4(1+\gamma^{-1})\eta_l^2\bar{\sigma}_l^2$$

$$\leq [(1+\gamma) + 4(1+\gamma^{-1})\eta_l^2 L^2]\frac{1}{K}\sum_{k=1}^K\mathbb{E}[\|\bar{\boldsymbol{x}}_{r,t}^k - \boldsymbol{x}_r\|^2] + 4(1+\gamma^{-1})\eta_l^2 L^2\frac{1}{N}\sum_{k=1}^K\sum_{i=1}^n\mathbb{E}[\|\boldsymbol{x}_{r,t}^i - \bar{\boldsymbol{x}}_{r,t}^k\|^2]$$

$$+ 4(1+\gamma^{-1})\eta_l^2(\alpha^2\mathbb{E}[\|\nabla f(\boldsymbol{x}_r)\|^2] + \sigma_c^2) + \eta_l^2\frac{\sigma^2}{n} + \frac{m(n-m)}{n(n-1)}4(1+\gamma^{-1})\eta_l^2\bar{\sigma}_l^2, \tag{40}$$

where the second inequality holds by Assumption 4.4, and incorporates the previously defined term $\bar{\sigma}_l^2 = \frac{1}{K}\sum_{k=1}^{K}\sigma_k^2$. Choosing $\gamma = \frac{1}{4\mathcal{I}-1}$ with the condition of $\eta_l \leq \frac{1}{8\mathcal{I}L}$, we have

$$
\frac{1}{K}\sum_{k=1}^{K}\mathbb{E}[\|\bar{\boldsymbol{x}}_{r,t+1}^k - \boldsymbol{x}_r\|^2]
$$

$$
\leq \left(1 + \frac{1}{4\mathcal{I}-1} + \frac{1}{2(4\mathcal{I}-1)}\right)\frac{1}{K}\sum_{k=1}^{K}\mathbb{E}[\|\bar{\boldsymbol{x}}_{r,t}^k - \boldsymbol{x}_r\|^2] + 16\mathcal{I}\eta_l^2 L^2 \frac{1}{N}\sum_{k=1}^{K}\sum_{i\in\mathcal{V}_k}\mathbb{E}[\|\boldsymbol{x}_{r,t}^i - \bar{\boldsymbol{x}}_{r,t}^k\|^2]
$$

$$
+ 16\mathcal{I}\eta_l^2(\alpha^2\mathbb{E}[\|\nabla f(\boldsymbol{x}_r)\|^2] + \sigma_c^2) + \eta_l^2\frac{\sigma^2}{n} + \frac{m(n-m)}{n(n-1)}16\mathcal{I}\eta_l^2\bar{\sigma}_l^2
$$

$$
= C_{\mathcal{I}}\frac{1}{K}\sum_{k=1}^{K}\mathbb{E}[\|\bar{\boldsymbol{x}}_{r,t}^k - \boldsymbol{x}_r\|^2] + 16\mathcal{I}\eta_l^2 L^2 \frac{1}{N}\sum_{k=1}^{K}\sum_{i\in\mathcal{V}_k}\mathbb{E}[\|\boldsymbol{x}_{r,t}^i - \bar{\boldsymbol{x}}_{r,t}^k\|^2] + 16\mathcal{I}\eta_l^2(\alpha^2\mathbb{E}[\|\nabla f(\boldsymbol{x}_r)\|^2] + \sigma_c^2)
$$

$$
+ \eta_l^2\frac{\sigma^2}{n} + \frac{m(n-m)}{n(n-1)}16\mathcal{I}\eta_l^2\bar{\sigma}_l^2, \tag{41}
$$

where $C_{\mathcal{I}} = 1 + \frac{3}{2}\cdot\frac{1}{4\mathcal{I}-1}$. This concludes the proof. $\qquad\square$

### F.1 Lemma for intra-cluster consensus error

**Lemma F.2.** *The intra-cluster consensus error $\sum_{i=1}^{n}\|\bar{\boldsymbol{x}}_{r,t}^k - \boldsymbol{x}_{r,t}^i\|^2$, also known as $\|X_{r,t}^{k,\perp}\|_F^2$, has the following upper bound,*

$$
\frac{1}{N}\sum_{k=1}^{K}\mathbb{E}[\|X_{r,t+1}^{k,\perp}\|_F^2]
$$

$$
\leq \left(\max_{k\in[K]}\rho_k^2(1+\zeta_k^{-1}) + \eta_l^2\cdot 4L^2\max_{k\in[K]}\{\rho_k^2(1+\zeta_k)\}\right)\frac{1}{N}\sum_{k=1}^{K}\mathbb{E}[\|X_{r,t}^{k,\perp}\|^2]
$$

$$
+ \eta_l^2\max_{k\in[K]}\{\rho_k^2(1+\zeta_k)\}\cdot 4L^2\mathbb{E}[\|\bar{\boldsymbol{x}}_{r,t}^k - \boldsymbol{x}_r\|^2] + \eta_l^2\max_{k\in[K]}\{\rho_k^2(1+\zeta_k)\}\cdot 4(\alpha^2\mathbb{E}[\|\nabla f(\boldsymbol{x}_r)\|^2] + \sigma_c^2)
$$

$$
+ \eta_l^2\frac{1}{K}\sum_{k=1}^{K}\rho_k^2(1+\zeta_k)4\sigma_k^2 + \eta_l^2\sigma^2\rho_{\max}^2, \tag{42}
$$

*where $\zeta_k$ is some constant related to the Young's inequality, and it could be uniformly chosen for all $k = 1,...,K$.*

*Proof.* By definition we have $X_{r,t}^k = (\boldsymbol{x}_{r,t}^1,...,\boldsymbol{x}_{r,t}^n)'$ and $X_{r,t}^{k,\perp} = X_{r,t}^k(I_n - J)$, where $J = \frac{1}{n}\boldsymbol{1}_n\cdot\boldsymbol{1}_n'$. Thus we have

$$
\sum_{i=1}^{n}\|\bar{\boldsymbol{x}}_{r,t}^k - \boldsymbol{x}_{r,t}^i\|^2 = \|(\boldsymbol{x}_{r,t}^1,...,\boldsymbol{x}_{r,t}^n)(I_n - J)\cdot I_n\cdot(I_n - J)(\boldsymbol{x}_{r,t}^1,...,\boldsymbol{x}_{r,t}^n)'\|_F
$$

$$
= \|X_{r,t}^k{}'(I_n - J)\cdot(I_n - J)X_{r,t}^k\|_F
$$

$$
= \|X_{r,t}^{k,\perp}{}'\cdot X_{r,t}^{k,\perp}\|_F
$$

$$
= \|X_{r,t}^{k,\perp}\|_F^2, \tag{43}
$$

Recall the update rule of AFGA and CAFGA, there is $X_{r,t+1}^{k,\perp} = (W_k - J)(X_{r,t}^{k,\perp} - \eta_l G_{r,t}^k)$, then we have

$$
\begin{aligned}
\mathbb{E}[\|X_{r,t+1}^{k,\perp}\|_F^2] &= \mathbb{E}(\mathbb{E}(\|(W_k - J)(X_{r,t}^{k,\perp} - \eta_l G_{r,t}^k)\|^2 | \mathcal{F}_{r,t-1})) \\
&= \mathbb{E}(\mathbb{E}(\|(W_k - J)(X_{r,t}^{k,\perp} - \eta_l \nabla F(X_{r,t}^k) + \eta_l \nabla F(X_{r,t}^k) - \eta_l G_{r,t}^k)\|^2 | \mathcal{F}_{r,t-1})) \\
&= \mathbb{E}(\mathbb{E}(\|(W_k - J)(X_{r,t}^{k,\perp} - \eta_l \nabla F(X_{r,t}^k))\|^2 | \mathcal{F}_{r,t-1})) \\
&\quad + \eta_l^2 \mathbb{E}(\mathbb{E}(\|(W_k - J)(\nabla F(X_{r,t}^k) - G_{r,t}^k)\|^2 | \mathcal{F}_{r,t-1})) \\
&\leq \mathbb{E}[\|(W_k - J)(X_{r,t}^{k,\perp} - \eta_l \nabla F(X_{r,t}^k))\|^2] + \eta_l^2 \rho_k^2 n \sigma^2 \\
&\leq \rho_k^2(1 + \zeta_k^{-1}) \cdot \mathbb{E}[\|X_{r,t}^{k,\perp}\|^2] + \rho_k^2(1 + \zeta_k)\eta_l^2 \mathbb{E}[\|\nabla F(X_{r,t}^k)\|^2] + \eta_l^2 \rho_k^2 n \sigma^2,
\end{aligned}
\tag{44}
$$

where the $\nabla F_k(X^k) \in \mathbb{R}^{n \times d}$ is associated to cluster $k$ by stacking $\nabla f_i(\boldsymbol{x}^i)$ for $i \in \mathcal{V}_k$ row-wise. The third equality is due to the unbiasedness of stochastic gradient. The first inequality holds by Assumption 4.2 and $\|\nabla F(X_{r,t}^k) - G_{r,t}^k\|_F = \sum_{i=1}^n \|\nabla f_i(\boldsymbol{x}_{r,t}^i) - \boldsymbol{g}_{r,t}^i\|^2$. For the Frobenius norm, there is $\|AB\|_F \leq \|A\|_2 \|B\|_F$. The second inequality holds by Young's inequality with some parameter $\zeta_k > 0$ and $\|AB\|_F \leq \|A\|_2 \|B\|_F$ as well. For $\nabla F_k(X_{r,t}^k)$, by definition, we have

$$
\begin{aligned}
\|\nabla F_k(X_{r,t}^k)\|_F^2 &= \sum_{i \in \mathcal{V}_k} \|\nabla f_i(\boldsymbol{x}_{r,t}^i)\|^2 \\
&= \sum_{i \in \mathcal{V}_k} \|\nabla f_i(\boldsymbol{x}_{r,t}^i) - \nabla f_i(\bar{\boldsymbol{x}}_{r,t}^k) + \nabla f_i(\bar{\boldsymbol{x}}_{r,t}^k) - \nabla \bar{f}_k(\bar{\boldsymbol{x}}_{r,t}^k) + \nabla \bar{f}_k(\bar{\boldsymbol{x}}_{r,t}^k) - \nabla \bar{f}_k(\boldsymbol{x}_r) + \nabla \bar{f}_k(\boldsymbol{x}_r)\|^2 \\
&\leq \sum_{i \in \mathcal{V}_k} \Big[ 4\|\nabla f_i(\boldsymbol{x}_{r,t}^i) - \nabla f_i(\bar{\boldsymbol{x}}_{r,t}^k)\|^2 + 4\|\nabla f_i(\bar{\boldsymbol{x}}_{r,t}^k) - \nabla \bar{f}_k(\bar{\boldsymbol{x}}_{r,t}^k)\|^2 + 4\|\nabla \bar{f}_k(\bar{\boldsymbol{x}}_{r,t}^k) - \nabla \bar{f}_k(\boldsymbol{x}_r)\|^2 \\
&\quad + 4\|\nabla \bar{f}_k(\boldsymbol{x}_r)\|^2 \Big] \\
&\leq \sum_{i \in \mathcal{V}_k} \Big[ 4\|\nabla f_i(\bar{\boldsymbol{x}}_{r,t}^k) - \nabla \bar{f}_k(\bar{\boldsymbol{x}}_{r,t}^k)\|^2 + 4L^2\|\boldsymbol{x}_{r,t}^i - \bar{\boldsymbol{x}}_{r,t}^k\|^2 + 4L^2\|\bar{\boldsymbol{x}}_{r,t}^k - \boldsymbol{x}_r\|^2 + 4\|\nabla \bar{f}_k(\boldsymbol{x}_r)\|^2 \Big] \\
&\leq 4L^2\|X_{r,t}^{k,\perp}\|^2 + 4L^2 n\|\bar{\boldsymbol{x}}_{r,t}^k - \boldsymbol{x}_r\|^2 + 4n\|\nabla \bar{f}_k(\boldsymbol{x}_r)\|^2 + 4n\sigma_k^2,
\end{aligned}
\tag{45}
$$

where the first inequality holds by Cauchy inequality, the second inequality holds by Assumption 4.1, and the last inequality holds by Assumption 4.4. Averaging Eq. 45 over $k = 1, ..., K$, we have the following

iteration

$$\frac{1}{N} \sum_{k=1}^{K} \mathbb{E}[\|X_{r,t+1}^{k,\perp}\|^2]$$

$$\leq \frac{1}{N} \sum_{k=1}^{K} \rho_k^2 (1 + \zeta_k^{-1}) \cdot \mathbb{E}[\|X_{r,t}^{k,\perp}\|^2] + \frac{1}{N} \sum_{k=1}^{K} \rho_k^2 (1 + \zeta_k) \eta_l^2 \mathbb{E}[\|\nabla F(X_{r,t}^k))\|^2] + \eta_l^2 \sigma^2 \frac{1}{K} \sum_{k=1}^{K} \rho_k^2$$

$$\leq \frac{1}{N} \sum_{k=1}^{K} \rho_k^2 (1 + \zeta_k^{-1}) \cdot \mathbb{E}[\|X_{r,t}^{k,\perp}\|^2] + \eta_l^2 \frac{1}{N} \sum_{k=1}^{K} \rho_k^2 (1 + \zeta_k) \cdot 4L^2 \mathbb{E}[\|X_{r,t}^{k,\perp}\|^2]$$

$$+ \eta_l^2 \frac{1}{K} \sum_{k=1}^{K} \rho_k^2 (1 + \zeta_k) \cdot 4L^2 \mathbb{E}[\|\bar{\boldsymbol{x}}_{r,t}^k - \boldsymbol{x}_r\|^2] + \eta_l^2 \frac{1}{K} \sum_{k=1}^{K} \rho_k^2 (1 + \zeta_k) \cdot 4\mathbb{E}[\|\nabla \bar{f}_k(\boldsymbol{x}_r)\|^2]$$

$$+ \eta_l^2 \frac{1}{K} \sum_{k=1}^{K} \rho_k^2 (1 + \zeta_k) \cdot 4\sigma_k^2 + \eta_l^2 \sigma^2 \rho_{\max}^2$$

$$\leq \left( \max_{k \in [K]} \rho_k^2 (1 + \zeta_k^{-1}) + \eta_l^2 \cdot 4L^2 \max_{k \in [K]} \{\rho_k^2 (1 + \zeta_k)\} \right) \frac{1}{N} \sum_{k=1}^{K} \mathbb{E}[\|X_{r,t}^{k,\perp}\|^2]$$

$$+ \eta_l^2 \max_{k \in [K]} \{\rho_k^2 (1 + \zeta_k)\} \cdot 4L^2 \mathbb{E}[\|\bar{\boldsymbol{x}}_{r,t}^k - \boldsymbol{x}_r\|^2] + \eta_l^2 \max_{k \in [K]} \{\rho_k^2 (1 + \zeta_k)\} \cdot 4(\alpha^2 \mathbb{E}[\|\nabla f(\boldsymbol{x}_r)\|^2] + \sigma_c^2)$$

$$+ \eta_l^2 \frac{1}{K} \sum_{k=1}^{K} \rho_k^2 (1 + \zeta_k) 4\sigma_k^2 + \eta_l^2 \sigma^2 \rho_{\max}^2. \tag{46}$$

This concludes the proof. $\qquad \square$

## F.2 Lemma for summation of intra-cluster and inter-cluster consensus errors

**Lemma F.3.** *If the local learning rate satisfies the condition:* $\eta_l \leq \frac{1}{8\mathcal{I}L}$, *the for all local round* $s = 0, ..., \mathcal{I}-1$, *there is*

$$\frac{1}{N} \sum_{k=1}^{K} \mathbb{E}[\|X_{r,t}^{k,\perp}\|^2] + \frac{1}{K} \sum_{k=1}^{K} \mathbb{E}[\|\bar{\boldsymbol{x}}_{r,t}^k - \boldsymbol{x}_r\|^2]$$

$$\leq (t+1)C_1 \eta_l^2 (\mathcal{I} + \rho_{\max}^2 H_{\mathcal{I},\rho})(\alpha^2 \mathbb{E}[\|\nabla f(\boldsymbol{x}_r)\|^2] + \sigma_c^2) + (t+1)C_1 \rho_{\max}^2 H_{\mathcal{I},\rho} \eta_l^2 \bar{\sigma}_l^2$$

$$+ (t+1)C_1 \eta_l^2 \sigma^2 \rho_{\max}^2 + (t+1)C_1 \left( 1 + \frac{H_{\mathcal{I},\rho}^2}{\mathcal{I}^2} \cdot \rho_{\max}^2 \right) \eta_l^2 \left( \frac{\sigma^2}{n} + \frac{m(n-m)}{n(n-1)} \mathcal{I} \bar{\sigma}_l^2 \right), \tag{47}$$

*where* $C_1$ *is a constant independent to parameters.*

*Proof.* Denote an auxiliary vector

$$M_{r,t} = \left( \frac{1}{N} \sum_{k=1}^{K} \mathbb{E}[\|X_{r,t}^{k,\perp}\|^2], \quad \frac{1}{K} \sum_{k=1}^{K} \mathbb{E}[\|\bar{\boldsymbol{x}}_{r,t}^k - \boldsymbol{x}_r\|^2] \right)^T. \tag{48}$$

From Lemma F.1 and F.2, we have the following inequality which is defined element-wise for $s = 0, ..., \mathcal{I}-1$

$$M_{r,t+1} \leq G \cdot M_{r,t} + B_{r,t}, \tag{49}$$

where

$$G = \begin{pmatrix} \max_{k \in [K]} \rho_k^2 (1 + \zeta_k^{-1}) + \eta_l^2 \rho_L \cdot 4L^2 & \eta_l^2 \rho_L \cdot 4L^2 \\ 16\mathcal{I}\eta_l^2 L^2 & C_{\mathcal{I}} \end{pmatrix} \tag{50}$$

$$B_{r,t} = \begin{pmatrix} 4\rho_L \eta_l^2 (\alpha^2 \mathbb{E}[\|\nabla f(\boldsymbol{x}_r)\|^2] + \sigma_c^2) + 4\rho_L \eta_l^2 \bar{\sigma}_l^2 + \eta_l^2 \sigma^2 \rho_{\max}^2 \\ 16\mathcal{I}\eta_l^2 (\alpha^2 \mathbb{E}[\|\nabla f(\boldsymbol{x}_r)\|^2] + \sigma_c^2) + \eta_l^2 \frac{\sigma^2}{n} + \frac{16m(n-m)}{n(n-1)} \mathcal{I}\eta_l^2 \sigma_k^2 \end{pmatrix} = \begin{pmatrix} b^{(1)} \\ b^{(2)} \end{pmatrix}. \tag{51}$$

Consider the eigen-decomposition of matrix $G$,

$$G = \frac{1}{16\mathcal{I}\eta_l^2 L^2(\lambda_1 - \lambda_2)}\begin{pmatrix}\lambda_1 - C_{\mathcal{I}} & \lambda_2 - C_{\mathcal{I}} \\ 16\mathcal{I}\eta_l^2 L^2 & 16\mathcal{I}\eta_l^2 L^2\end{pmatrix} \cdot \begin{pmatrix}\lambda_1 & 0 \\ 0 & \lambda_2\end{pmatrix} \cdot \begin{pmatrix}16\mathcal{I}\eta_l^2 L^2 & -\lambda_2 + C_{\mathcal{I}} \\ -16\mathcal{I}\eta_l^2 L^2 & \lambda_1 - C_{\mathcal{I}}\end{pmatrix}, \tag{52}$$

where we assume $\lambda_1 \leq \lambda_2$, thus we have

$$G^j B = \frac{1}{16\mathcal{I}\eta_l^2 L^2(\lambda_1 - \lambda_2)}\begin{pmatrix}\lambda_1 - C_{\mathcal{I}} & \lambda_2 - C_{\mathcal{I}} \\ 16\mathcal{I}\eta_l^2 L^2 & 16\mathcal{I}\eta_l^2 L^2\end{pmatrix}\begin{pmatrix}\lambda_1 & 0 \\ 0 & \lambda_2\end{pmatrix}\begin{pmatrix}16\mathcal{I}\eta_l^2 L^2 & -\lambda_2 + C_{\mathcal{I}} \\ -16\mathcal{I}\eta_l^2 L^2 & \lambda_1 - C_{\mathcal{I}}\end{pmatrix}\begin{pmatrix}b^{(1)} \\ b^{(2)}\end{pmatrix}$$

$$= \frac{1}{16\mathcal{I}\eta_l^2 L^2(\lambda_1 - \lambda_2)}\begin{pmatrix}(\lambda_1 - C_{\mathcal{I}})(\lambda_1^j 16\mathcal{I}\eta_l^2 L^2 b^{(1)} + \lambda_1^j(-\lambda_2 + C_{\mathcal{I}})b^{(2)}) + (\lambda_2 - C_{\mathcal{I}})(-\lambda_2^j 16\mathcal{I}\eta_l^2 L^2 b^{(1)} + \lambda_2^j(\lambda_1 - C_{\mathcal{I}})b^{(2)}) \\ 16\mathcal{I}\eta_l^2 L^2(\lambda_1^j 16\mathcal{I}\eta_l^2 L^2 b^{(1)} + \lambda_1^j(-\lambda_2 + C_{\mathcal{I}})b^{(2)}) + 16\mathcal{I}\eta_l^2 L^2(-\lambda_2^j 16\mathcal{I}\eta_l^2 L^2 b^{(1)} + \lambda_2^j(\lambda_1 - C_{\mathcal{I}})b^{(2)})\end{pmatrix}.$$

$$\tag{53}$$

Therefore the sum of two elements has the following result

$$(1,1)G^j B_{r,t-j} = \lambda_1^j b^{(1)} + \lambda_2^j b^{(2)} + \frac{\lambda_2^j - \lambda_1^j}{\lambda_2 - \lambda_1}\left(16\mathcal{I}\eta_l^2 L^2 b^{(1)} + 4\eta_l^2 \rho_L L^2 b^{(2)}\right)$$

$$\leq \lambda_2^j(b^{(1)} + b^{(2)}) + \frac{\lambda_2^j - \lambda_1^j}{\lambda_2 - \lambda_1}\left(16\mathcal{I}\eta_l^2 L^2 b^{(1)} + 4\eta_l^2 \rho_L L^2 b^{(2)}\right). \tag{54}$$

Therefore, we have the following result

$$\sum_{j=0}^{t}(1,1)G^j B_{r,t-j} \leq \sum_{j=0}^{t}\left(\lambda_2^j(b_{r,t-j}^{(1)} + b_{r,t-j}^{(2)}) + \frac{\lambda_2^j - \lambda_1^j}{\lambda_2 - \lambda_1}\left(16\mathcal{I}\eta_l^2 L^2 b^{(1)} + 4\eta_l^2 \rho_L L^2 b^{(2)}\right)\right). \tag{55}$$

Since $\lambda_2 \geq C_{\mathcal{I}} > 1$, we have

$$\frac{\lambda_2^j - \lambda_1^j}{\lambda_2 - \lambda_1} = \lambda_2^{l-1}\sum_{t=0}^{l-1}\left(\frac{\lambda_1}{\lambda_2}\right)^s \leq \lambda_2^{j-1}\min\left\{\frac{\lambda_2}{\lambda_2 - \lambda_1}, l\right\} \leq \lambda_2^j \min\left\{\frac{1}{\lambda_2 - \lambda_1}, l\right\}, \tag{56}$$

thus we have

$$\sum_{j=0}^{t}(1,1)G^j B_{r,t-j} \leq \sum_{j=0}^{t}\lambda_2^j(b_{r,t-j}^{(1)} + b_{r,t-j}^{(2)}) + \sum_{l=0}^{s}\left(\lambda_2^j \min\left\{\frac{1}{\lambda_2 - \lambda_1}, l\right\}\left(16\mathcal{I}\eta_l^2 L^2 b^{(1)} + 4\eta_l^2 \rho_L L^2 b^{(2)}\right)\right). \tag{57}$$

By the definition of $\rho_L = \max_{k\in[k]}\rho_k^2(1 + \zeta_k)$ and by the Gershgorin's theorem, since $\eta_l > 0$, we have the upper bound for $\lambda_2$,

$$\lambda_2 \leq \max\left\{\max_{k\in[K]}\rho_k^2(1 + \zeta_k^{-1}) + \eta_l^2 \rho_L \cdot 8L^2, C_{\mathcal{I}} + 16\mathcal{I}\eta_l^2 L^2\right\}$$

$$< \max\left\{\max_{k\in[K]}\rho_k^2(1 + \zeta_k^{-1}) + \frac{\rho_L}{(4\mathcal{I} - 1)2\mathcal{I}}, 1 + \frac{2}{4\mathcal{I} - 1}\right\}, \tag{58}$$

where the last inequality holds by the bound of $\eta_l^2 \leq \frac{1}{64\mathcal{I}^2 L^2} < \frac{1}{16\mathcal{I}(4\mathcal{I}-1)L^2}$. Define a distance constant $H_{\mathcal{I},\rho} = \min\left\{\mathcal{I}, \frac{1}{1-\rho_{\max}}\right\}$. Next we consider two cases: *small or dense* communication network with $\rho_{\max} \leq 1 - \frac{1}{\mathcal{I}}$ and *large and sparse* communication network with $\rho_{\max} > 1 - \frac{1}{\mathcal{I}}$.

**Case 1:** For $\rho_{\max} \leq 1 - \frac{1}{\mathcal{I}}$, i.e., $\frac{1}{1-\rho_{\max}} \leq \mathcal{I}$, thus we have $H_{\mathcal{I},\rho} = \frac{1}{1-\rho_{\max}}$. Let $\zeta_k = \frac{\rho_k}{1-\rho_k}$, then we have

$$\max_{k\in[k]}\rho_k^2(1 + \zeta_k^{-1}) = \rho_{\max}, \quad \rho_L = \max_{k\in[k]}\left\{\frac{\rho_k^2}{1-\rho_k}\right\} = \frac{\rho_{\max}^2}{1-\rho_{\max}} = \rho_{\max}^2 H_{\mathcal{I},\rho}, \tag{59}$$

where the middle part of the second equality holds by the monotonically increasing of $\frac{x^2}{1-x}$. Then the bound for $\lambda_2$ is formalized as

$$
\begin{aligned}
\lambda_2 &\le \max\left\{\rho_{\max} + \frac{\rho_{\max}^2}{(1-\rho_{\max})2\mathcal{I}(4\mathcal{I}-1)}, 1 + \frac{3}{2(4\mathcal{I}-1)}\right\} \\
&\le \max\left\{1 - \frac{1}{\mathcal{I}} + \frac{(1-\frac{1}{\mathcal{I}})^2}{2(\mathcal{I}-1)}, 1 + \frac{3}{2(4\mathcal{I}-1)}\right\} \\
&< 1 + \frac{3}{4\mathcal{I}-1},
\end{aligned}
\tag{60}
$$

where the second inequality holds by $\rho_{\max} \le 1 - \frac{1}{\mathcal{I}}$. Then by $s \le \mathcal{I}$ and $\lambda_2 \ge 1$ (just by the definition of matrix $G$ can get this result), we can obtain the following bound

$$
\sum_{j=0}^{t} \lambda_2^j b_{t-j}^1 \le \left(\left(1 + \frac{2}{4\mathcal{I}-1}\right)^{\mathcal{I}}\right) \cdot \sum_{j=0}^{t} b_j^{(1)} \le 3 \cdot \sum_{j=0}^{t} b_j^{(1)}.
\tag{61}
$$

We also have

$$
\rho_{\max} + \eta_l^2 \rho_L 4L^2 \le \rho_{\max} + \frac{\rho_{\max}}{(1-\rho_{\max})(4\mathcal{I}-1)4\mathcal{I}} \le 1 - \frac{1}{\mathcal{I}} + \frac{(1-\frac{1}{\mathcal{I}})^2}{4(4\mathcal{I}-1)} \le C_{\mathcal{I}},
\tag{62}
$$

where the second inequality holds by the upper bound for $\rho_{\max}$. By the definition of matrix $G$, we bound the difference of $\lambda_2 - \lambda_1$,

$$
\begin{aligned}
\lambda_2 - \lambda_1 &= C_{\mathcal{I}} - \rho_{\max} - \eta_l^2 \rho_L 4L^2 \\
&\ge C_{\mathcal{I}} - \left(\rho_{\max} + \frac{\rho_{\max}}{(1-\rho_{\max})(4\mathcal{I}-1)4\mathcal{I}}\right) \\
&\ge C_{\mathcal{I}} - \left(\rho_{\max} + \rho_{\max} \cdot \frac{1-\frac{1}{\mathcal{I}}}{4(4\mathcal{I}-1)}\right) \\
&\ge 1 + \frac{1}{4\mathcal{I}-1} - \left(\rho_{\max} + \rho_{\max} \cdot \frac{1}{4\mathcal{I}-1}\right) \\
&= (1 - \rho_{\max})\left(1 + \frac{1}{4\mathcal{I}-1}\right) \\
&\ge 1 - \rho_{\max}.
\end{aligned}
\tag{63}
$$

where the first and second inequality hold by the defined notations. Then we have

$$
\begin{aligned}
\sum_{j=0}^{t}(1,1)G^j B_{r,t-j} &\le \sum_{j=0}^{t} \lambda_2^j (b_{r,t-j}^{(1)} + b_{r,t-j}^{(2)}) + \sum_{j=0}^{t}\left(\lambda_2^j \min\left\{\frac{1}{\lambda_2-\lambda_1}, j\right\}\left(16\mathcal{I}\eta_l^2 L^2 b_{r,t-j}^{(1)} + 4\eta_l^2 \rho_L L^2 b_{r,t-j}^{(2)}\right)\right) \\
&\le \sum_{j=0}^{t} 3(b_{r,j}^{(1)} + b_{r,j}^{(2)}) + \sum_{j=0}^{t} 3\eta_l^2\left(16\mathcal{I}L^2 b_{r,j}^{(1)} + 4\rho_L L^2 b_{r,j}^{(2)}\right)\left(\min\left\{\frac{1}{\lambda_2-\lambda_1}, \mathcal{I}\right\}\right) \\
&\le \sum_{j=0}^{t} 3(b_{r,j}^{(1)} + b_{r,j}^{(2)}) + \sum_{j=0}^{t} 3\eta_l^2\left(16\mathcal{I}L^2 b_{r,j}^{(1)} + 4H_{\mathcal{I},\rho}^2 \rho_{\max}^2 L^2 b_{r,j}^{(2)}\right) \\
&\le \sum_{j=0}^{t} 3(b_{r,j}^{(1)} + b_{r,j}^{(2)}) + \sum_{j=0}^{t} \frac{H_{\mathcal{I},\rho}}{16\mathcal{I}(4\mathcal{I}-1)} \cdot \left(48\mathcal{I}b_{r,j}^{(1)} + 12H_{\mathcal{I},\rho}\rho_{\max}^2 b_{r,j}^{(2)}\right) \\
&\le \sum_{j=0}^{t} 4(b_{r,j}^{(1)} + b_{r,j}^{(2)}) + \sum_{j=0}^{t} \frac{H_{\mathcal{I},\rho}^2}{\mathcal{I}^2} \cdot \rho_{\max}^2 b_{r,j}^{(2)}.
\end{aligned}
\tag{64}
$$

Then by the definition of $b^{(1)}$ and $b^{(2)}$, we have

$$\sum_{j=0}^{t}(1,1)G^j B_{r,t-j}$$

$$\leq \sum_{j=0}^{t} 4\bigg( (4\rho_L\eta_l^2 + 16\mathcal{I}\eta_l^2)(\alpha^2\mathbb{E}[\|\nabla f(\boldsymbol{x}_r)\|^2]+\sigma_c^2) + 4\rho_L\eta_l^2\bar\sigma_l^2 + \eta_l^2\sigma^2\rho_{\max}^2 + \eta_l^2\frac{\sigma^2}{n} + 16\frac{m(n-m)}{n(n-1)}\mathcal{I}\eta_l^2\bar\sigma_l^2 \bigg)$$

$$+ \sum_{j=0}^{t}\frac{H_{\mathcal{I},\rho}^2}{\mathcal{I}^2}\cdot\rho_{\max}^2\bigg(16\mathcal{I}\eta_l^2(\alpha^2\mathbb{E}[\|\nabla f(\boldsymbol{x}_r)\|^2]+\sigma_c^2) + \eta_l^2\frac{\sigma^2}{n} + 16\frac{m(n-m)}{n(n-1)}\mathcal{I}\eta_l^2\bar\sigma_l^2 \bigg)$$

$$= (t+1)\bigg[\bigg( 4(4\rho_L\eta_l^2+16\mathcal{I}\eta_l^2) + \frac{H_{\mathcal{I},\rho}^2}{\mathcal{I}^2}\cdot\rho_{\max}^2 16\mathcal{I}\eta_l^2 \bigg)(\alpha^2\mathbb{E}[\|\nabla f(\boldsymbol{x}_r)\|^2]+\sigma_c^2) + 16\rho_L\eta_l^2\bar\sigma_l^2$$

$$+ 16\eta_l^2\sigma^2\rho_{\max}^2 + \bigg(4\eta_l^2 + \frac{H_{\mathcal{I},\rho}^2}{\mathcal{I}^2}\cdot\rho_{\max}^2\eta_l^2\bigg)\bigg(\frac{\sigma^2}{n} + 16\frac{m(n-m)}{n(n-1)}\mathcal{I}\bar\sigma_l^2\bigg)\bigg]$$

$$\leq (t+1)\bigg[\bigg(16\rho_{\max}^2 H_{\mathcal{I},\rho}\eta_l^2 + 64\mathcal{I}\eta_l^2 + 16\eta_l^2 H_{\mathcal{I},\rho}\cdot\rho_{\max}^2\bigg)(\alpha^2\mathbb{E}[\|\nabla f(\boldsymbol{x}_r)\|^2]+\sigma_c^2) + 16\rho_{\max}^2 H_{\mathcal{I},\rho}\eta_l^2\bar\sigma_l^2$$

$$+ 16\eta_l^2\sigma^2\rho_{\max}^2 + \bigg(4\eta_l^2 + \frac{H_{\mathcal{I},\rho}^2}{\mathcal{I}^2}\cdot\rho_{\max}^2\eta_l^2\bigg)\bigg(\frac{\sigma^2}{n} + 16\frac{m(n-m)}{n(n-1)}\mathcal{I}\bar\sigma_l^2\bigg)\bigg]$$

$$\leq (t+1)C_1\eta_l^2(\mathcal{I}+\rho_{\max}^2 H_{\mathcal{I},\rho})(\alpha^2\mathbb{E}[\|\nabla f(\boldsymbol{x}_r)\|^2]+\sigma_c^2) + (t+1)C_1\rho_{\max}^2 H_{\mathcal{I},\rho}\eta_l^2\bar\sigma_l^2$$

$$+ (t+1)C_1\eta_l^2\sigma^2\rho_{\max}^2 + (t+1)C_1\bigg(1+\frac{H_{\mathcal{I},\rho}^2}{\mathcal{I}^2}\cdot\rho_{\max}^2\bigg)\eta_l^2\bigg(\frac{\sigma^2}{n}+\frac{m(n-m)}{n(n-1)}\mathcal{I}\bar\sigma_l^2\bigg), \tag{65}$$

where $C_1$ is some universal constant. The inequality holds by $\rho_L = \rho_{\max}^2 H_{\mathcal{I},\rho}$ and $H_{\mathcal{I},\rho} \leq \mathcal{I}$.
**Case 2:** In this case we have $\rho_{\max} > 1 - \frac{1}{\mathcal{I}}$, which means $H_{\mathcal{I},\rho} = \mathcal{I}$. Let $\zeta_k = (4\mathcal{I}-1)$, thus we have

$$\max_{k\in[K]}\rho_k^2(1+\zeta_k^{-1}) = \rho_{\max}^2(1+(4\mathcal{I}-1)^{-1}), \quad \rho_L = 4\mathcal{I}\rho_{\max}^2 H_{\mathcal{I},\rho}. \tag{66}$$

The upper bound for $\lambda_2$ has the form of

$$\lambda_2 \leq \max\bigg\{\max_{k\in[K]}\rho_k^2(1+\zeta_k^{-1}) + \eta_l^2\rho_L\cdot 8L^2, C_{\mathcal{I}}\bigg\}$$

$$\leq \max\bigg\{\rho_{\max}^2(1+(4\mathcal{I}-1)^{-1}) + \frac{2\rho_{\max}^2}{4\mathcal{I}-1}, 1+\frac{3}{2(4\mathcal{I}-1)}\bigg\}$$

$$\leq 1 + \frac{3}{4\mathcal{I}-1}. \tag{67}$$

By the fact of $\min\big\{\frac{1}{\lambda_2-\lambda_1}, l\big\} \leq \mathcal{I} = H_{\mathcal{I},\rho}$, we have

$$\sum_{j=0}^{t}(1,1)G^j B_{r,t-j} \leq \sum_{j=0}^{t}\lambda_2^j(b_{r,t-j}^{(1)}+b_{r,t-j}^{(2)}) + \sum_{j=0}^{t}\bigg(\lambda_2^j\min\bigg\{\frac{1}{\lambda_2-\lambda_1}, l\bigg\}\eta_l^2\cdot 4\rho_L L^2 b_{r,t-j}^{(2)}\bigg)$$

$$\leq \sum_{j=0}^{t}3(b_{r,j}^{(1)}+b_{r,j}^{(2)}) + \sum_{j=0}^{t}\eta_l^2\cdot 16\rho_{\max}H_{\mathcal{I},\rho}L^2 b_l^{(2)}\cdot 3H_{\mathcal{I},\rho}$$

$$\leq \sum_{j=0}^{t}3(b_{r,j}^{(1)}+b_{r,j}^{(2)}) + \sum_{j=0}^{t}16\rho_{\max}H_{\mathcal{I},\rho}b_l^{(2)}\cdot\frac{3}{16}\frac{H_{\mathcal{I},\rho}}{\mathcal{I}^2}$$

$$= \sum_{j=0}^{t}3(b_{r,j}^{(1)}+b_{r,j}^{(2)}) + \sum_{j=0}^{t}3\rho_{\max}b_l^{(2)}\cdot\frac{H_{\mathcal{I},\rho}^2}{\mathcal{I}^2}, \tag{68}$$

where the above inequalities hold by the fact that $\rho_L = 4\mathcal{I}\rho_{\max}^2 = 4H_{\mathcal{I},\rho}\rho_{\max}^2$ and the constraint on step size $\eta_l$. Thus we can get a similar upper bound as Eq. 65 in Case 1. This concludes the proof. $\qquad\square$

**Lemma F.4.** *With the similar condition in Lemma F.3, we have the corresponding bound for the intra-cluster consensus error* $\|X_{r,t}^{k,\perp}\|_F^2$,

$$
\frac{1}{N}\sum_{k=1}^{K}\mathbb{E}[\|X_{r,t}^{k,\perp}\|^2] \le (t+1)C_1\eta_l^2 H_{\mathcal{I},\rho}\rho_{\max}^2(\alpha^2\mathbb{E}[\|\nabla f(\boldsymbol{x}_r)\|^2]+\sigma_c^2) + (t+1)C_1\eta_l^2 H_{\mathcal{I},\rho}\rho_{\max}^2\bar{\sigma}_l^2
$$

$$
+ (t+1)C_1\eta_l^2\rho_{\max}^2\sigma^2 + (t+1)C_1\frac{H_{\mathcal{I},\rho}^2}{\mathcal{I}^2}\eta_l^2\rho_{\max}^2\frac{\sigma^2}{n} + (t+1)C_1\frac{m(n-m)}{n(n-1)}\frac{H_{\mathcal{I},\rho}^2}{\mathcal{I}}\eta_l^2\rho_{\max}^2\bar{\sigma}_l^2.
\tag{69}
$$

*Proof.* With the same definition of the auxiliary vector $M_{r,t}$ and the matrix $G$ and $B_{r,t}$ in the proof of Lemma F.3, there is

$$
M_{r,t} = \left(\frac{1}{N}\sum_{k=1}^{K}\mathbb{E}[\|X_{r,t}^{k,\perp}\|^2], \quad \frac{1}{K}\sum_{k=1}^{K}\mathbb{E}[\|\bar{\boldsymbol{x}}_{r,t}^k - \boldsymbol{x}_r\|^2]\right)^{\top},
$$

$$
M_{r,t} = G^{t+1}M_{r,0} + \sum_{j=0}^{t}G^j B_{r,t-j} = \sum_{j=0}^{t}G^j B_{r,t-j},
\tag{70}
$$

hence we have

$$
\frac{1}{N}\sum_{k=1}^{K}\mathbb{E}[\|X_{r,t}^{k,\perp}\|^2] = (1,0)\cdot M_{r,t} = (1,0)\cdot\sum_{j=0}^{t}G^j B_{r,t-j}
$$

$$
= \sum_{j=0}^{t}\left[\lambda_1^j b_{r,j}^{(1)} + \frac{\lambda_2^j - \lambda_1^j}{\lambda_2 - \lambda_1}4\eta_l^2\rho_L L^2 b_{r,j}^{(2)}\right]
$$

$$
\le \sum_{j=0}^{t}\left[\lambda_2^j b_{r,j}^{(1)} + \frac{\lambda_2^j - \lambda_1^j}{\lambda_2 - \lambda_1}4\eta_l^2\rho_L L^2 b_{r,j}^{(2)}\right]
$$

$$
\le \sum_{j=0}^{t}\left[\lambda_2^j b_{r,j}^{(1)} + \frac{\lambda_2^j - \lambda_1^j}{\lambda_2 - \lambda_1}4\eta_l^2\rho_L L^2 b_{r,j}^{(2)}\right],
\tag{71}
$$

with the similar proof techniques as in Lemma F.3, there is

$$
\frac{1}{N}\sum_{k=1}^{K}\mathbb{E}[\|X_{r,t}^{k,\perp}\|^2] \le \sum_{j=0}^{t}\left[3b_{r,j}^{(1)} + \frac{1}{\mathcal{I}^2}H_{\mathcal{I},\rho}^2\rho_{\max}^2 b_{r,j}^{(2)}\right]
$$

$$
\le \sum_{j=0}^{t}\left[12\rho_L\eta_l^2(\alpha^2\mathbb{E}[\|\nabla f(\boldsymbol{x}_r)\|^2]+\sigma_c^2) + 12\rho_L\eta_l^2\bar{\sigma}_l^2 + 3\eta_l^2\sigma^2\rho_{\max}^2\right.
$$

$$
\left.+ \frac{H_{\mathcal{I},\rho}^2}{\mathcal{I}^2}\rho_{\max}^2\left(16\mathcal{I}\eta_l^2(\alpha^2\mathbb{E}[\|\nabla f(\boldsymbol{x}_r)\|^2]+\sigma_c^2) + 16\frac{m(n-m)}{n(n-1)}\mathcal{I}\eta_l^2\bar{\sigma}_l^2 + \eta_l^2\frac{\sigma^2}{n}\right)\right]
$$

$$
\le (t+1)C_1\eta_l^2 H_{\mathcal{I},\rho}\rho_{\max}^2(\alpha^2\mathbb{E}[\|\nabla f(\boldsymbol{x}_r)\|^2]+\sigma_c^2) + (t+1)C_1\eta_l^2 H_{\mathcal{I},\rho}\rho_{\max}^2\bar{\sigma}_l^2
$$

$$
+ (t+1)C_1\eta_l^2\rho_{\max}^2\sigma^2 + (t+1)C_1\eta_l^2\frac{H_{\mathcal{I},\rho}^2}{\mathcal{I}^2}\rho_{\max}^2\frac{\sigma^2}{n} + (t+1)C_1\frac{m(n-m)}{n(n-1)}\eta_l^2\frac{H_{\mathcal{I},\rho}^2}{\mathcal{I}}\rho_{\max}^2\bar{\sigma}_l^2.
\tag{72}
$$

This concludes the proof. $\qquad\square$

### F.3  Lemmas for model difference $\Delta_r$

There is a corresponding Lemma about model difference $\Delta_r$ for the partial participation settings.

**Lemma F.5.** *The global model difference $\Delta_r = \sum_{k=1}^{K} \sum_{i \in cS_r} \Delta_r^i$ in partial participation settings satisfies*

$$\mathbb{E}[\|\Delta_r\|^2]$$
$$\leq \frac{2\eta_l^2 \mathcal{I}}{N} \sigma^2 + 2\eta_l^2 (\mathcal{I}-1) \sum_{t=0}^{\mathcal{I}-1} \mathbb{E}\left[\left\|\frac{1}{N}\sum_{i=1}^{N} \nabla f_i(\boldsymbol{x}_{r,t}^i)\right\|^2\right] + 4\eta_l^2 \mathbb{E}\left[\left\|\frac{1}{K}\sum_{k=1}^{K} \nabla \bar{f}_k(\bar{\boldsymbol{x}}_{r,\mathcal{I}-1}^k)\right\|^2\right]$$
$$+ 8\left(\frac{n-m}{m(n-1)} + \eta_l^2 L^2\right)\left(\frac{1}{N}\sum_{k=1}^{K} \mathbb{E}[\|X_{r,\mathcal{I}-1}^{k,\perp}\|^2]\right) + \frac{2\eta_l^2 \sigma^2}{N}\left(\frac{n-m}{m} \cdot \rho_{\max}^2\right). \tag{73}$$

*Proof.* Recall the definition of $\bar{\boldsymbol{x}}_{r,t}$, there is $\bar{\boldsymbol{x}}_{r,t} = \frac{1}{N}\sum_{i=1}^{N} \boldsymbol{x}_{r,t}^i$ (here we don't need to consider of client sampling) and the intra-cluster average $\bar{\boldsymbol{x}}_{r,t+1}^k = \bar{\boldsymbol{x}}_{r,t}^k - \eta_l \bar{\boldsymbol{x}}_{r,t+1}^k$, where $\bar{\boldsymbol{x}}_{r,t+1}^k = \frac{1}{n}\sum_{i \in \mathcal{V}_k} \boldsymbol{g}_{r,t}^i$.

For the model difference $\Delta_r$, we have

$$\mathbb{E}[\|\Delta_r\|^2] = \mathbb{E}\left[\left\|\frac{1}{K}\sum_{k=1}^{K}\frac{1}{m}\sum_{i \in \mathcal{S}_r^k} \boldsymbol{x}_{r,\mathcal{I}}^i - \boldsymbol{x}_r\right\|^2\right]$$

$$= \mathbb{E}\left[\left\|\frac{1}{K}\sum_{k=1}^{K}\frac{1}{m}\sum_{i \in \mathcal{S}_r^k} \boldsymbol{x}_{r,\mathcal{I}}^i \mp \bar{\boldsymbol{x}}_{r,\mathcal{I}} \mp \bar{\boldsymbol{x}}_{r,\mathcal{I}-1} \mp \cdots \mp \bar{\boldsymbol{x}}_{r,1} - \boldsymbol{x}_r\right\|^2\right]$$

$$\leq 2\mathbb{E}\left[\left\|\frac{1}{Np}\sum_{k=1}^{K}\sum_{i \in \mathcal{S}_r^k} \boldsymbol{x}_{r,\mathcal{I}}^i - \bar{\boldsymbol{x}}_{r,\mathcal{I}}\right\|^2\right] + 2\mathbb{E}\left[\left\|\bar{\boldsymbol{x}}_{r,\mathcal{I}} \mp \bar{\boldsymbol{x}}_{r,\mathcal{I}-1} \mp \cdots \mp \bar{\boldsymbol{x}}_{r,1} - \boldsymbol{x}_r\right\|^2\right], \tag{74}$$

where the inequality holds by Cauchy-Schwarz inequality. For the first term in Eq. 74, by the probability of the sampling strategy (see details in Lemma F.6), we have

$$\mathbb{E}\left[\left\|\frac{1}{Np}\sum_{k=1}^{K}\sum_{i \in \mathcal{S}_r^k} \boldsymbol{x}_{r,\mathcal{I}}^i - \bar{\boldsymbol{x}}_{r,\mathcal{I}}\right\|^2\right] = \frac{1}{(Np)^2}\mathbb{E}\left[\left\|\sum_{k=1}^{K}\sum_{i \in \mathcal{S}_r^k} \boldsymbol{x}_{r,\mathcal{I}}^i - \bar{\boldsymbol{x}}_{r,\mathcal{I}}\right\|^2\right]$$

$$\leq \frac{1}{(Np)^2}\mathbb{E}\left[\frac{m(m-1)}{n(n-1)}\left\|\sum_{k=1}^{K}\sum_{i \in \mathcal{V}_k} \boldsymbol{x}_{r,\mathcal{I}}^i - \bar{\boldsymbol{x}}_{r,\mathcal{I}}\right\|^2 + \frac{Km(n-m)}{n(n-1)}\sum_{k=1}^{K}\sum_{i \in \mathcal{V}_k} \|\boldsymbol{x}_{r,\mathcal{I}}^i - \bar{\boldsymbol{x}}_{r,\mathcal{I}}\|^2\right]$$

$$= \frac{K}{(Np)^2}\frac{m(n-m)}{n(n-1)}\sum_{k=1}^{K}\sum_{i \in \mathcal{V}_k} \mathbb{E}[\|\boldsymbol{x}_{r,\mathcal{I}}^i - \bar{\boldsymbol{x}}_{r,\mathcal{I}}\|^2]. \tag{75}$$

For the second part in Eq. 74, we have

$$
\mathbb{E}\left[\left\|\bar{\boldsymbol{x}}_{r,\mathcal{I}} \pm \cdots \pm \bar{\boldsymbol{x}}_{r,1} - \boldsymbol{x}_r\right\|^2\right]
$$

$$
= \mathbb{E}\left[\left\|\eta_l \sum_{t=0}^{\mathcal{I}-1} \bar{\boldsymbol{g}}_{r,t}\right\|^2\right]
$$

$$
= \mathbb{E}\left[\left\|\frac{\eta_l}{N} \sum_{t=0}^{\mathcal{I}-1} \sum_{k=1}^{K} \sum_{i \in \mathcal{S}_r^k} \boldsymbol{g}_{r,t}^i\right\|^2\right]
$$

$$
= \mathbb{E}\left[\left\|\frac{\eta_l}{N} \sum_{t=0}^{\mathcal{I}-1} \sum_{k=1}^{K} \sum_{i \in \mathcal{S}_r^k} (\boldsymbol{g}_{r,t}^i \pm \nabla f_i(\boldsymbol{x}_{r,t}^i))\right\|^2\right]
$$

$$
= \mathbb{E}\left[\left\|\frac{\eta_l}{N} \sum_{t=0}^{\mathcal{I}-1} \sum_{k=1}^{K} \sum_{i \in \mathcal{S}_r^k} (\boldsymbol{g}_{r,t}^i - \nabla f_i(\boldsymbol{x}_{r,t}^i))\right\|^2\right] + \mathbb{E}\left[\left\|\frac{\eta_l}{N} \sum_{t=0}^{\mathcal{I}-1} \sum_{k=1}^{K} \sum_{i \in \mathcal{S}_r^k} \nabla f_i(\boldsymbol{x}_{r,t}^i)\right\|^2\right]
$$

$$
\leq \frac{\eta_l^2 (\mathcal{I}-1) M}{N^2} \sigma^2 + \mathbb{E}\left[\left\|\frac{\eta_l}{N} \sum_{t=0}^{\mathcal{I}-1} \sum_{k=1}^{K} \sum_{i \in \mathcal{S}_r^k} \nabla f_i(\boldsymbol{x}_{r,t}^i)\right\|^2\right], \tag{76}
$$

where

$$
\mathbb{E}\left[\left\|\frac{\eta_l}{N} \sum_{t=0}^{\mathcal{I}-1} \sum_{k=1}^{K} \sum_{i \in \mathcal{S}_r^k} \nabla f_i(\boldsymbol{x}_{r,t}^i)\right\|^2\right]
$$

$$
= \mathbb{E}\left[\left\|\sum_{t=0}^{\mathcal{I}-1} \left(\frac{\eta_l}{N} \sum_{k=1}^{K} \sum_{i \in \mathcal{S}_r^k} \nabla f_i(\boldsymbol{x}_{r,t}^i) \mp \frac{\eta_l}{N} \sum_{k=1}^{K} \sum_{i \in \mathcal{S}_r^k} \nabla f_i(\bar{\boldsymbol{x}}_{r,t}^k) \mp \frac{\eta_l M}{N^2} \sum_{i=1}^{N} \nabla f_i(\bar{\boldsymbol{x}}_{r,t}^k)\right)\right\|^2\right]
$$

$$
\leq 2\mathbb{E}\left[\left\|\sum_{t=0}^{\mathcal{I}-1} \left(\frac{\eta_l}{N} \sum_{k=1}^{K} \sum_{i \in \mathcal{S}_r^k} \nabla f_i(\bar{\boldsymbol{x}}_{r,t}^k) - \frac{\eta_l M}{N^2} \sum_{i=1}^{N} \nabla f_i(\bar{\boldsymbol{x}}_{r,t}^k)\right)\right\|^2\right]
$$

$$
+ 4\mathbb{E}\left[\left\|\sum_{t=0}^{\mathcal{I}-1} \frac{\eta_l m}{N} \sum_{k=1}^{K} \nabla \bar{f}_k(\bar{\boldsymbol{x}}_{r,t}^k)\right\|^2\right] + 4\mathbb{E}\left[\left\|\sum_{t=0}^{\mathcal{I}-1} \left(\frac{\eta_l}{N} \sum_{k=1}^{K} \sum_{i \in \mathcal{S}_r^k} [\nabla f_i(\boldsymbol{x}_{r,t}^i) - \nabla f_i(\bar{\boldsymbol{x}}_{r,t}^k)]\right)\right\|^2\right]
$$

$$
\leq 2\sum_{t=0}^{\mathcal{I}-1} \mathbb{E}\left[\left\|\frac{\eta_l}{N} \sum_{k=1}^{K} \sum_{i \in \mathcal{S}_r^k} \nabla f_i(\bar{\boldsymbol{x}}_{r,t}^k) - \frac{\eta_l M}{N^2} \sum_{i=1}^{N} \nabla f_i(\bar{\boldsymbol{x}}_{r,t}^k)\right\|^2\right]
$$

$$
+ 4(\mathcal{I}-1) \sum_{t=0}^{\mathcal{I}-1} \mathbb{E}\left[\left\|\frac{\eta_l m}{N} \sum_{k=1}^{K} \nabla \bar{f}_k(\bar{\boldsymbol{x}}_{r,t}^k)\right\|^2\right] + 4(\mathcal{I}-1) \sum_{t=0}^{\mathcal{I}-1} L^2 \frac{\eta_l^2 M}{N^2} \sum_{k=1}^{K} \sum_{i \in \mathcal{S}_r^k} \mathbb{E}[\|\boldsymbol{x}_{r,t}^i - \bar{\boldsymbol{x}}_{r,t}^k\|^2], \tag{77}
$$

therefore,

$$
\mathbb{E}\left[\left\|\frac{\eta_l}{N}\sum_{t=0}^{\mathcal{I}-1}\sum_{k=1}^{K}\sum_{i\in\mathcal{S}_r^k}\nabla f_i(\boldsymbol{x}_{r,t}^i)\right\|^2\right]
$$

$$
\leq 4\sum_{t=0}^{\mathcal{I}-1}\mathbb{E}\left[\left\|\frac{\eta_l}{N}\sum_{k=1}^{K}\sum_{i\in\mathcal{S}_r^k}\nabla f_i(\bar{\boldsymbol{x}}_{r,t}^k)\right\|^2\right] + 4\sum_{t=0}^{\mathcal{I}-1}\mathbb{E}\left[\left\|\frac{\eta_l M}{N^2}\sum_{i=1}^{N}\nabla f_i(\bar{\boldsymbol{x}}_{r,t}^k)\right\|^2\right]
$$

$$
+ 4(\mathcal{I}-1)\sum_{t=0}^{\mathcal{I}-1}\mathbb{E}\left[\left\|\frac{\eta_l m}{N}\sum_{k=1}^{K}\nabla\bar{f}_k(\bar{\boldsymbol{x}}_{r,t}^k)\right\|^2\right] + 4(\mathcal{I}-1)L^2\frac{\eta_l^2 M}{N^2}\sum_{t=0}^{\mathcal{I}-1}\sum_{k=1}^{K}\sum_{i\in\mathcal{S}_r^k}\mathbb{E}[\|\boldsymbol{x}_{r,t}^i - \bar{\boldsymbol{x}}_{r,t}^k\|^2]
$$

$$
= 4\sum_{t=0}^{\mathcal{I}-1}\mathbb{E}\left[\left\|\frac{\eta_l}{N}\sum_{k=1}^{K}\sum_{i\in\mathcal{S}_r^k}\nabla f_i(\bar{\boldsymbol{x}}_{r,t}^k)\right\|^2\right] + 4\mathcal{I}\sum_{t=0}^{\mathcal{I}-1}\mathbb{E}\left[\left\|\frac{\eta_l M}{N^2}\sum_{i=1}^{N}\nabla f_i(\bar{\boldsymbol{x}}_{r,t}^k)\right\|^2\right]
$$

$$
+ 4(\mathcal{I}-1)L^2\frac{\eta_l^2 M}{N^2}\sum_{t=0}^{\mathcal{I}-1}\sum_{k=1}^{K}\sum_{i\in\mathcal{S}_r^k}\mathbb{E}[\|\boldsymbol{x}_{r,t}^i - \bar{\boldsymbol{x}}_{r,t}^k\|^2]
$$

$$
\leq 8\sum_{t=0}^{\mathcal{I}-1}\mathbb{E}\left[\left\|\frac{\eta_l}{N}\sum_{k=1}^{K}\sum_{i\in\mathcal{S}_r^k}\nabla f_i(\boldsymbol{x}_{r,t}^i)\right\|^2\right] + 4\mathcal{I}\sum_{t=0}^{\mathcal{I}-1}\mathbb{E}\left[\left\|\frac{\eta_l M}{N^2}\sum_{i=1}^{N}\nabla f_i(\bar{\boldsymbol{x}}_{r,t}^k)\right\|^2\right]
$$

$$
+ 4(\mathcal{I}+1)L^2\frac{\eta_l^2 M}{N^2}\sum_{t=0}^{\mathcal{I}-1}\sum_{k=1}^{K}\sum_{i\in\mathcal{S}_r^k}\mathbb{E}[\|\boldsymbol{x}_{r,t}^i - \bar{\boldsymbol{x}}_{r,t}^k\|^2], \tag{78}
$$

where the last inequality follows

$$
4\sum_{t=0}^{\mathcal{I}-1}\mathbb{E}\left[\left\|\frac{\eta_l}{N}\sum_{k=1}^{K}\sum_{i\in\mathcal{S}_r^k}\nabla f_i(\bar{\boldsymbol{x}}_{r,t}^k)\right\|^2\right]
$$

$$
= 4\sum_{t=0}^{\mathcal{I}-1}\mathbb{E}\left[\left\|\frac{\eta_l}{N}\sum_{k=1}^{K}\sum_{i\in\mathcal{S}_r^k}[\nabla f_i(\bar{\boldsymbol{x}}_{r,t}^k)\mp\nabla f_i(\boldsymbol{x}_{r,t}^i)]\right\|^2\right]
$$

$$
\leq 8\sum_{t=0}^{\mathcal{I}-1}\mathbb{E}\left[\left\|\frac{\eta_l}{N}\sum_{k=1}^{K}\sum_{i\in\mathcal{S}_r^k}\nabla f_i(\boldsymbol{x}_{r,t}^i)\right\|^2\right] + 8\sum_{t=0}^{\mathcal{I}-1}\mathbb{E}\left[\left\|\frac{\eta_l}{N}\sum_{k=1}^{K}\sum_{i\in\mathcal{S}_r^k}[\nabla f_i(\bar{\boldsymbol{x}}_{r,t}^k)-\nabla f_i(\boldsymbol{x}_{r,t}^i)]\right\|^2\right]
$$

$$
\leq 8\sum_{t=0}^{\mathcal{I}-1}\mathbb{E}\left[\left\|\frac{\eta_l}{N}\sum_{k=1}^{K}\sum_{i\in\mathcal{S}_r^k}\nabla f_i(\boldsymbol{x}_{r,t}^i)\right\|^2\right] + 8\frac{\eta_l^2 M}{N^2}L^2\sum_{t=0}^{\mathcal{I}-1}\sum_{k=1}^{K}\sum_{i\in\mathcal{S}_r^k}\mathbb{E}[\|\boldsymbol{x}_{r,t}^i - \bar{\boldsymbol{x}}_{r,t}^k\|^2], \tag{79}
$$

and we use the characteristic of conditional expectation, where we use the characteristic of conditional expectation, i.e., $\mathbb{E}[\mathbb{E}_{\mathcal{S}_s}[\frac{\eta_l}{N}\sum_{k=1}^{K}\sum_{i\in\mathcal{S}_r^k}\nabla f_i(\bar{\boldsymbol{x}}_{r,t}^k)]] = \mathbb{E}[\frac{\eta_l M}{N^2}\sum_{k=1}^{K}\sum_{i\in\mathcal{V}_k}\nabla f_i(\bar{\boldsymbol{x}}_{r,t}^k)]$, Update the expectation term, that is $\mathbb{E}[\mathbb{E}_{\mathcal{S}_s}[\frac{\eta_l}{N}\sum_{k=1}^{K}\sum_{i\in\mathcal{S}_r^k}\nabla f_i(\bar{\boldsymbol{x}}_{r,t}^k) - \frac{\eta_l M}{N^2}\sum_{i=1}^{N}\nabla f_i(\bar{\boldsymbol{x}}_{r,t}^k)]] = 0$ and $\forall r \neq s, i \in \mathcal{S}_r^k$ is independent

with $i \in \mathcal{S}_r^k$. Then we have

$$8 \sum_{t=0}^{\mathcal{I}-1} \mathbb{E}\left[\left\|\frac{\eta_l}{N} \sum_{k=1}^{K} \sum_{i \in \mathcal{S}_r^k} \nabla f_i(\boldsymbol{x}_{r,t}^i)\right\|^2\right]$$

$$= \frac{8\eta_l^2}{N^2} \sum_{t=0}^{\mathcal{I}-1} \mathbb{E}\left[\left\|\sum_{k=1}^{K} \sum_{i \in \mathcal{V}_k} \mathbb{P}\{i \in \mathcal{S}_r^k\} \nabla f_i(\boldsymbol{x}_{r,t}^i)\right\|^2\right]$$

$$\leq \frac{8\eta_l^2 K}{N^2} \frac{m(n-m)}{n(n-1)} \sum_{t=0}^{\mathcal{I}-1} \sum_{k=1}^{K} \sum_{i \in \mathcal{V}_k} \mathbb{E}[\|\nabla f_i(\boldsymbol{x}_{r,t}^i)\|^2]$$

$$+ \frac{8\eta_l^2}{N^2} \frac{m(m-1)}{n(n-1)} \sum_{t=0}^{\mathcal{I}-1} \mathbb{E}\left[\left\|\sum_{k=1}^{K} \sum_{i \in \mathcal{V}_k} \nabla f_i(\boldsymbol{x}_{r,t}^i)\right\|^2\right]$$

$$= \frac{8\eta_l^2 K}{N^2} \frac{m(n-m)}{n(n-1)} \sum_{t=0}^{\mathcal{I}-1} \sum_{i=1}^{N} \mathbb{E}[\|\nabla f_i(\boldsymbol{x}_{r,t}^i)\|^2] + \frac{8\eta_l^2}{N^2} \frac{m(m-1)}{n(n-1)} \sum_{t=0}^{\mathcal{I}-1} \mathbb{E}\left[\left\|\sum_{i=1}^{N} \nabla f_i(\boldsymbol{x}_{r,t}^i)\right\|^2\right], \qquad (80)$$

where the inequality holds by Lemma F.6.

$$\sum_{t=0}^{\mathcal{I}-1} \sum_{i=1}^{N} \mathbb{E}[\|\nabla f_i(\boldsymbol{x}_{r,t}^i)\|^2]$$

$$= \sum_{t=0}^{\mathcal{I}-1} \sum_{i=1}^{N} \mathbb{E}[\|\nabla f_i(\boldsymbol{x}_{r,t}^i) \mp \nabla f_i(\bar{\boldsymbol{x}}_{r,t}^k) \mp \bar{f}_k(\bar{\boldsymbol{x}}_{r,t}^k) \mp \bar{f}_k(\boldsymbol{x}_r)\|^2]$$

$$\leq \sum_{t=0}^{\mathcal{I}-1} \sum_{k=1}^{K} \mathbb{E}[4L^2\|X_{r,t}^{k,\perp}\|^2 + 4L^2 n\|\bar{\boldsymbol{x}}_{r,t}^k - \boldsymbol{x}_r\|^2 + 4n\|\nabla \bar{f}_k(\boldsymbol{x}_r)\|^2 + 4n\sigma_k^2]$$

$$\leq \sum_{t=0}^{\mathcal{I}-1} \sum_{k=1}^{K} \left[4L^2 \mathbb{E}[\|X_{r,t}^{k,\perp}\|^2] + 4L^2 n\mathbb{E}[\|\bar{\boldsymbol{x}}_{r,t}^k - \boldsymbol{x}_r\|^2] + 4n\alpha^2 \mathbb{E}[\|\nabla f(\boldsymbol{x}_r)\|^2] + 4n\sigma_k^2 + 4n\sigma_c^2\right]. \qquad (81)$$

Then combining the previous terms, we have

$$
\mathbb{E}[\|\Delta_r\|^2]
$$

$$
\leq 2\mathbb{E}\left[\left\|\frac{1}{Np}\sum_{k=1}^{K}\sum_{i\in\mathcal{S}_r^k}\boldsymbol{x}_{r,\mathcal{I}}^i - \bar{\boldsymbol{x}}_{r,\mathcal{I}}\right\|^2\right] + 2\mathbb{E}\left[\left\|\bar{\boldsymbol{x}}_{r,\mathcal{I}} \mp \bar{\boldsymbol{x}}_{r,\mathcal{I}-1} \mp \cdots \mp \bar{\boldsymbol{x}}_{r,1} - \boldsymbol{x}_r\right\|^2\right]
$$

$$
\leq \frac{2K}{(Np)^2}\frac{m(n-m)}{n(n-1)}\sum_{k=1}^{K}\sum_{i\in\mathcal{V}_k}\mathbb{E}[\|\boldsymbol{x}_{r,\mathcal{I}}^i - \bar{\boldsymbol{x}}_{r,\mathcal{I}}^k\|^2] + \frac{2\eta_l^2\mathcal{I}M}{N^2}\sigma^2
$$

$$
+ 2\mathbb{E}\left[\left\|\frac{\eta_l}{N}\sum_{t=0}^{\mathcal{I}-1}\sum_{k=1}^{K}\sum_{i\in\mathcal{S}_r^k}\nabla f_i(\boldsymbol{x}_{r,t}^i)\right\|^2\right]
$$

$$
\leq \frac{2K}{(Np)^2}\frac{m(n-m)}{n(n-1)}\sum_{k=1}^{K}\sum_{i\in\mathcal{V}_k}\mathbb{E}[\|\boldsymbol{x}_{r,\mathcal{I}}^i - \bar{\boldsymbol{x}}_{r,\mathcal{I}}^k\|^2] + \frac{2\eta_l^2\mathcal{I}M}{N^2}\sigma^2
$$

$$
+ 16\sum_{t=0}^{\mathcal{I}-1}\mathbb{E}\left[\left\|\frac{\eta_l}{N}\sum_{k=1}^{K}\sum_{i\in\mathcal{S}_r^k}\nabla f_i(\boldsymbol{x}_{r,t}^i)\right\|^2\right] + 8\mathcal{I}\sum_{t=0}^{\mathcal{I}-1}\mathbb{E}\left[\left\|\frac{\eta_l m}{N}\sum_{k=1}^{K}\nabla\bar{f}_k(\bar{\boldsymbol{x}}_{r,t}^k)\right\|^2\right]
$$

$$
+ 8(\mathcal{I}+1)\frac{\eta_l^2 ML^2}{N^2}\sum_{t=0}^{\mathcal{I}-1}\sum_{k=1}^{K}\sum_{i\in\mathcal{S}_r}\mathbb{E}[\|\boldsymbol{x}_{r,t}^i - \bar{\boldsymbol{x}}_{r,t}^k\|^2]
$$

$$
\leq \frac{2K}{(Np)^2}\frac{m(n-m)}{n(n-1)}\sum_{k=1}^{K}\sum_{i\in\mathcal{V}_k}\mathbb{E}[\|\boldsymbol{x}_{r,\mathcal{I}}^i - \bar{\boldsymbol{x}}_{r,\mathcal{I}}^k\|^2] + \frac{2\eta_l^2\mathcal{I}M}{N^2}\sigma^2
$$

$$
+ \frac{16\eta_l^2 K}{N^2}\frac{m(n-m)}{n(n-1)}\sum_{t=0}^{\mathcal{I}-1}\sum_{i=1}^{N}\mathbb{E}[\|\nabla f_i(\boldsymbol{x}_{r,t}^i)\|^2] + \frac{16\eta_l^2}{N^2}\frac{m(m-1)}{n(n-1)}\sum_{t=0}^{\mathcal{I}-1}\mathbb{E}\left[\left\|\sum_{i=1}^{N}\nabla f_i(\boldsymbol{x}_{r,t}^i)\right\|^2\right]
$$

$$
+ 8\mathcal{I}\sum_{t=0}^{\mathcal{I}-1}\mathbb{E}\left[\left\|\frac{\eta_l m}{N}\sum_{k=1}^{K}\nabla\bar{f}_k(\bar{\boldsymbol{x}}_{r,t}^k)\right\|^2\right] + 8(\mathcal{I}+1)\frac{\eta_l^2 ML^2}{N^2}\sum_{t=0}^{\mathcal{I}-1}\sum_{k=1}^{K}\sum_{i\in\mathcal{S}_r}\mathbb{E}[\|\boldsymbol{x}_{r,t}^i - \bar{\boldsymbol{x}}_{r,t}^k\|^2]
$$

$$
\leq \frac{2K}{(Np)^2}\frac{m(n-m)}{n(n-1)}\sum_{k=1}^{K}\mathbb{E}[\|X_{r,\mathcal{I}}^{k,\perp}\|^2] + \frac{2\eta_l^2\mathcal{I}M}{N^2}\sigma^2
$$

$$
+ \frac{16\eta_l^2 K}{N^2}\frac{m(n-m)}{n(n-1)}\sum_{t=0}^{\mathcal{I}-1}\sum_{k=1}^{K}\left[4L^2\mathbb{E}[\|X_{r,t}^{k,\perp}\|^2] + 4L^2 n\mathbb{E}[\|\bar{\boldsymbol{x}}_{r,t}^k - \boldsymbol{x}_r\|^2]\right.
$$

$$
+ 4n\alpha^2\mathbb{E}[\|\nabla f(\boldsymbol{x}_r)\|^2] + 4n\sigma_k^2 + 4n\sigma_c^2\bigg] + \frac{16\eta_l^2}{N^2}\frac{m(m-1)}{n(n-1)}\sum_{t=0}^{\mathcal{I}-1}\mathbb{E}\left[\left\|\sum_{i=1}^{N}\nabla f_i(\boldsymbol{x}_{r,t}^i)\right\|^2\right]
$$

$$
+ 8\mathcal{I}\sum_{t=0}^{\mathcal{I}-1}\mathbb{E}\left[\left\|\frac{\eta_l m}{N}\sum_{k=1}^{K}\nabla\bar{f}_k(\bar{\boldsymbol{x}}_{r,t}^k)\right\|^2\right] + 8(\mathcal{I}+1)\frac{\eta_l^2 ML^2}{N^2}\sum_{t=0}^{\mathcal{I}-1}\sum_{k=1}^{K}\sum_{i\in\mathcal{S}_r}\mathbb{E}[\|\boldsymbol{x}_{r,t}^i - \bar{\boldsymbol{x}}_{r,t}^k\|^2], \tag{82}
$$

where we have

$$
\frac{16\eta_l^2 K}{N^2}\frac{m(n-m)}{n(n-1)}\sum_{t=0}^{\mathcal{I}-1}\sum_{k=1}^{K}\left[4L^2\mathbb{E}[\|X_{r,t}^{k,\perp}\|^2]+4L^2 n\mathbb{E}[\|\bar{\boldsymbol{x}}_{r,t}^{k}-\boldsymbol{x}_r\|^2]\right]
$$

$$
=\frac{16\eta_l^2 K}{N^2}\frac{m(n-m)}{n(n-1)}\sum_{t=0}^{\mathcal{I}-1}\sum_{k=1}^{K}\left[4L^2\mathbb{E}[\|X_{r,t}^{k,\perp}\|^2]+4L^2 n\mathbb{E}[\|\bar{\boldsymbol{x}}_{r,t}^{k}-\boldsymbol{x}_r\|^2]\right]
$$

$$
=\frac{64\eta_l^2 L^2}{N}\frac{m(n-m)}{n^2(n-1)}\sum_{t=0}^{\mathcal{I}-1}\sum_{k=1}^{K}\mathbb{E}[\|X_{r,t}^{k,\perp}\|^2]+\frac{64\eta_l^2 L^2}{K}\frac{m(n-m)}{n^2(n-1)}\sum_{t=0}^{\mathcal{I}-1}\sum_{k=1}^{K}\mathbb{E}[\|\bar{\boldsymbol{x}}_{r,t}^{k}-\boldsymbol{x}_r\|^2]
$$

$$
=64\eta_l^2 L^2\frac{m(n-m)}{n^2(n-1)}\sum_{t=0}^{\mathcal{I}-1}\left[\frac{1}{N}\sum_{k=1}^{K}\mathbb{E}[\|X_{r,t}^{k,\perp}\|^2]+\frac{1}{K}\sum_{k=1}^{K}\mathbb{E}[\|\bar{\boldsymbol{x}}_{r,t}^{k}-\boldsymbol{x}_r\|^2]\right]
$$

$$
\leq 64\eta_l^2 L^2\frac{m(n-m)}{n^2(n-1)}\left[\mathcal{I}^2 C_1\eta_l^2(\mathcal{I}+\rho_{\max}^2 H_{\mathcal{I},\rho})(\alpha^2\mathbb{E}[\|\nabla f(\boldsymbol{x}_r)\|^2]+\sigma_c^2)+\mathcal{I}^2 C_1\rho_{\max}^2 H_{\mathcal{I},\rho}\eta_l^2\bar{\sigma}_l^2\right.
$$

$$
\left.+\mathcal{I}^2 C_1\eta_l^2\sigma^2\rho_{\max}^2+\mathcal{I}^2 C_1\left(1+\frac{H_{\mathcal{I},\rho}^2}{\mathcal{I}^2}\cdot\rho_{\max}^2\right)\eta_l^2\left(\frac{\sigma^2}{n}+\frac{m(n-m)}{n(n-1)}\mathcal{I}\bar{\sigma}_l^2\right)\right], \tag{83}
$$

we have

$$
\frac{2K}{(Np)^2}\frac{m(n-m)}{n(n-1)}\sum_{k=1}^{K}\mathbb{E}[\|X_{r,\mathcal{I}}^{k,\perp}\|^2]
$$

$$
=\frac{2}{Kn}\frac{(n-m)}{m(n-1)}\sum_{k=1}^{K}\mathbb{E}[\|X_{r,\mathcal{I}}^{k,\perp}\|^2]
$$

$$
\leq\frac{2(n-m)}{m(n-1)}\left[\mathcal{I}C_1\eta_l^2 H_{\mathcal{I},\rho}\rho_{\max}^2(\alpha^2\mathbb{E}[\|\nabla f(\boldsymbol{x}_r)\|^2]+\sigma_c^2)+\mathcal{I}C_1\eta_l^2 H_{\mathcal{I},\rho}\rho_{\max}^2\bar{\sigma}_l^2\right.
$$

$$
\left.+\mathcal{I}C_1\eta_l^2\rho_{\max}^2\sigma^2+\mathcal{I}C_1\eta_l^2\frac{H_{\mathcal{I},\rho}^2}{\mathcal{I}^2}\cdot\rho_{\max}^2\left(\frac{\sigma^2}{n}+\frac{m(n-m)}{n(n-1)}\mathcal{I}\bar{\sigma}_l^2\right)\right], \tag{84}
$$

and

$$
8(\mathcal{I}+1)\frac{\eta_l^2 ML^2}{N^2}\sum_{t=0}^{\mathcal{I}-1}\sum_{k=1}^{K}\sum_{i\in\mathcal{S}_r}\mathbb{E}[\|\boldsymbol{x}_{r,t}^{i}-\bar{\boldsymbol{x}}_{r,t}^{k}\|^2]
$$

$$
\leq 8(\mathcal{I}+1)\frac{\eta_l^2 ML^2}{N^2}\sum_{t=0}^{\mathcal{I}-1}\sum_{k=1}^{K}\sum_{i\in\mathcal{V}_k}\mathbb{E}[\|\boldsymbol{x}_{r,t}^{i}-\bar{\boldsymbol{x}}_{r,t}^{k}\|^2]
$$

$$
=8(\mathcal{I}+1)\frac{\eta_l^2 mL^2}{n}\sum_{t=0}^{\mathcal{I}-1}\frac{1}{N}\sum_{k=1}^{K}\mathbb{E}[\|X_{r,t}^{k,\perp}\|^2]
$$

$$
=8(\mathcal{I}+1)\frac{\eta_l^2 mL^2}{n}\sum_{t=0}^{\mathcal{I}-1}\left[(t+1)C_1\eta_l^2 H_{\mathcal{I},\rho}\rho_{\max}^2(\alpha^2\mathbb{E}[\|\nabla f(\boldsymbol{x}_r)\|^2]+\sigma_c^2)+(t+1)C_1\eta_l^2 H_{\mathcal{I},\rho}\rho_{\max}^2\bar{\sigma}_l^2\right.
$$

$$
\left.+(t+1)C_1\eta_l^2\rho_{\max}^2\sigma^2+(t+1)C_1\eta_l^2\frac{H_{\mathcal{I},\rho}^2}{\mathcal{I}^2}\cdot\rho_{\max}^2\frac{\sigma^2}{n}\right]
$$

$$
=4(\mathcal{I}+1)^2\frac{\eta_l^2 mL^2}{n}\left[\mathcal{I}C_1\eta_l^2 H_{\mathcal{I},\rho}\rho_{\max}^2(\alpha^2\mathbb{E}[\|\nabla f(\boldsymbol{x}_r)\|^2]+\sigma_c^2)+\mathcal{I}C_1\eta_l^2 H_{\mathcal{I},\rho}\rho_{\max}^2\bar{\sigma}_l^2\right.
$$

$$
\left.+\mathcal{I}C_1\eta_l^2\rho_{\max}^2\sigma^2+\mathcal{I}C_1\eta_l^2\frac{H_{\mathcal{I},\rho}^2}{\mathcal{I}^2}\cdot\rho_{\max}^2\left(\frac{\sigma^2}{n}+\frac{m(n-m)}{n(n-1)}\mathcal{I}\bar{\sigma}_l^2\right)\right]. \tag{85}
$$

Then by merging pieces together, we have

$$
\mathbb{E}[\|\Delta_r\|^2]
$$
$$
\leq \frac{2\eta_l^2 \mathcal{I} M}{N^2}\sigma^2 + \left(\frac{2(n-m)}{m(n-1)} + \frac{4(\mathcal{I}+1)^2\eta_l^2 m L^2}{n}\right)\left[\mathcal{I}C_1\eta_l^2 H_{\mathcal{I},\rho}\rho_{\max}^2(\alpha^2\mathbb{E}[\|\nabla f(\boldsymbol{x}_r)\|]^2 + \sigma_c^2) + \mathcal{I}C_1\eta_l^2 H_{\mathcal{I},\rho}\rho_{\max}^2\bar{\sigma}_l^2\right.
$$
$$
\left. + \mathcal{I}C_1\eta_l^2\rho_{\max}^2\sigma^2 + \mathcal{I}C_1\eta_l^2\frac{H_{\mathcal{I},\rho}^2}{\mathcal{I}^2}\rho_{\max}^2\left(\frac{\sigma^2}{n} + \frac{m(n-m)}{n(n-1)}\mathcal{I}\bar{\sigma}_l^2\right)\right]
$$
$$
+ 64\eta_l^2 L^2\frac{m(n-m)}{n^2(n-1)}\left[\mathcal{I}^2 C_1\eta_l^2(\mathcal{I} + \rho_{\max}^2 H_{\mathcal{I},\rho})(\alpha^2\mathbb{E}[\|\nabla f(\boldsymbol{x}_r)\|]^2 + \sigma_c^2)\right.
$$
$$
\left. + \mathcal{I}^2 C_1\rho_{\max}^2 H_{\mathcal{I},\rho}\eta_l^2\bar{\sigma}_l^2 + \mathcal{I}^2 C_1\eta_l^2\sigma^2\rho_{\max}^2 + \mathcal{I}^2 C_1\eta_l^2\left(1 + \frac{H_{\mathcal{I},\rho}^2}{\mathcal{I}^2}\cdot\rho_{\max}^2\right)\left(\frac{\sigma^2}{n} + \frac{m(n-m)}{n(n-1)}\mathcal{I}\bar{\sigma}_l^2\right)\right]
$$
$$
+ \frac{16\eta_l^2 K}{N^2}\frac{m(n-m)}{n(n-1)}\sum_{t=0}^{\mathcal{I}-1}\sum_{k=1}^{K}[4n\sigma_k^2 + 4n\sigma_c^2]
$$
$$
+ \frac{16\eta_l^2}{N^2}\frac{m(m-1)}{n(n-1)}\sum_{t=0}^{\mathcal{I}-1}\mathbb{E}\left[\left\|\sum_{i=1}^{N}\nabla f_i(\boldsymbol{x}_{r,t}^i)\right\|^2\right] + \frac{8\eta_l^2\mathcal{I} m^2}{n^2}\sum_{t=0}^{\mathcal{I}-1}\mathbb{E}\left[\left\|\frac{1}{K}\sum_{k=1}^{K}\nabla \bar{f}_k(\bar{\boldsymbol{x}}_{r,t}^k)\right\|^2\right]
$$
$$
\leq \frac{2\eta_l^2\mathcal{I} M}{N^2}\sigma^2 + \frac{64\eta_l^2\mathcal{I} m(n-m)}{n^2(n-1)}[\bar{\sigma}_l^2 + \sigma_c^2]
$$
$$
+ \left(\frac{2(n-m)}{m(n-1)} + 64\eta_l^2 L^2\frac{m(n-m)}{n^2(n-1)}\mathcal{I} + \frac{4(\mathcal{I}+1)^2\eta_l^2 m L^2}{n}\right)\left[\mathcal{I}C_1\eta_l^2 H_{\mathcal{I},\rho}\rho_{\max}^2(\alpha^2\mathbb{E}[\|\nabla f(\boldsymbol{x}_r)\|]^2 + \sigma_c^2)\right.
$$
$$
\left. + \mathcal{I}C_1\eta_l^2 H_{\mathcal{I},\rho}\rho_{\max}^2\bar{\sigma}_l^2 + \mathcal{I}C_1\eta_l^2\rho_{\max}^2\sigma^2 + \mathcal{I}C_1\eta_l^2\frac{H_{\mathcal{I},\rho}^2}{\mathcal{I}^2}\rho_{\max}^2\left(\frac{\sigma^2}{n} + \frac{m(n-m)}{n(n-1)}\mathcal{I}\bar{\sigma}_l^2\right)\right]
$$
$$
+ 64\eta_l^2 L^2\frac{m(n-m)}{n^2(n-1)}\left[\mathcal{I}^3 C_1\eta_l^2(\alpha^2\mathbb{E}[\|\nabla f(\boldsymbol{x}_r)\|^2 + \sigma_c^2) + \mathcal{I}^2 C_1\eta_l^2\left(\frac{\sigma^2}{n} + \frac{m(n-m)}{n(n-1)}\mathcal{I}\bar{\sigma}_l^2\right)\right]
$$
$$
+ \frac{16\eta_l^2 m(m-1)}{n(n-1)}\sum_{t=0}^{\mathcal{I}-1}\mathbb{E}\left[\left\|\frac{1}{N}\sum_{i=1}^{N}\nabla f_i(\boldsymbol{x}_{r,t}^i)\right\|^2\right] + \frac{8\eta_l^2\mathcal{I} m^2}{n^2}\sum_{t=0}^{\mathcal{I}-1}\mathbb{E}\left[\left\|\frac{1}{K}\sum_{k=1}^{K}\nabla \bar{f}_k(\bar{\boldsymbol{x}}_{r,t}^k)\right\|^2\right]. \tag{86}
$$

Thus it concludes the proof.

$\square$

## F.4 Additional Supporting Lemmas

**Lemma F.6** (Cluster sampling). *For model weights $\boldsymbol{y}_r^{k,i}, \forall k \in [K], i \in \mathcal{V}_k, r \in [R], t \in [\mathcal{I}]$, there is*

$$
\mathbb{E}\left[\left\|\sum_{k\in[K]}\sum_{i=1}^{n}\mathbb{I}\{i\in\mathcal{S}_r^k\}\boldsymbol{y}_r^{k,i}\right\|^2\right] \leq \mathbb{E}\left[\frac{m(m-1)}{n(n-1)}\left\|\sum_{k=1}^{K}\sum_{i\in\mathcal{V}_k}\boldsymbol{y}_r^{k,i}\right\|^2 + \frac{Km(n-m)}{n(n-1)}\sum_{k=1}^{K}\sum_{i\in\mathcal{V}_k}\|\boldsymbol{y}_r^{k,i}\|^2\right]. \tag{87}
$$

*Proof.*

$$\mathbb{E}\bigg[\bigg\|\sum_{k=1}^{K}\sum_{i\in\mathcal{V}_k}\mathbb{I}\{i\in\mathcal{S}_r^k\}\boldsymbol{y}_r^{k,i}\bigg\|^2\bigg]$$

$$= \mathbb{P}\{i\in\mathcal{S}_r^k\}\mathbb{E}\bigg[\sum_{k=1}^{K}\sum_{i\in\mathcal{V}_k}\|\boldsymbol{y}_r^{k,i}\|^2\bigg] + \mathbb{P}\{i\neq j\in\mathcal{S}_r^k\}\mathbb{E}\bigg[\sum_{k=1}^{K}\sum_{i\neq j\in\mathcal{V}_k}(\boldsymbol{y}_r^{k,i})'(\boldsymbol{y}_r^{k,j})\bigg]$$

$$+ \mathbb{P}\{i\in\mathcal{S}_r^k, j\in\mathcal{S}_l^t|k\neq l\in[K]\}\mathbb{E}\bigg[\sum_{k\neq l}\sum_{i\in\mathcal{V}_k}\sum_{j\in\mathcal{V}_l}(\boldsymbol{y}_r^{i,k})'(\boldsymbol{y}_r^{j,l})\bigg]$$

$$= \mathbb{E}\bigg[\frac{m}{n}\sum_{k=1}^{K}\sum_{i\in\mathcal{V}_k}\|\boldsymbol{y}_r^{k,i}\|^2 + \frac{m(m-1)}{n(n-1)}\sum_{k=1}^{K}\sum_{i\neq j\in\mathcal{V}_k}(\boldsymbol{y}_r^{k,i})'(\boldsymbol{y}_r^{k,j})$$

$$+ \frac{m^2}{n^2}\sum_{k\neq l}\sum_{i\in\mathcal{V}_k}\sum_{j\in\mathcal{V}_l}(\boldsymbol{y}_r^{i,k})'(\boldsymbol{y}_r^{j,l})\bigg]$$

$$= \mathbb{E}\bigg[\frac{m(m-1)}{n(n-1)}\bigg\|\sum_{k=1}^{K}\sum_{i\in\mathcal{V}_k}\boldsymbol{y}_r^{k,i}\bigg\|^2 + \frac{m(n-m)}{n(n-1)}\sum_{k=1}^{K}\sum_{i\in\mathcal{V}_k}\|\boldsymbol{y}_r^{k,i}\|^2$$

$$+ \frac{m(n-m)}{n^2(n-1)}\sum_{k\neq l}\sum_{i\in\mathcal{V}_k}\sum_{j\in\mathcal{V}_l}(\boldsymbol{y}_r^{i,k})'(\boldsymbol{y}_r^{j,l})\bigg]$$

$$\leq \mathbb{E}\bigg[\frac{m(m-1)}{n(n-1)}\bigg\|\sum_{k=1}^{K}\sum_{i\in\mathcal{V}_k}\boldsymbol{y}_r^{k,i}\bigg\|^2 + \frac{m(n-m)}{n(n-1)}\sum_{k=1}^{K}\sum_{i\in\mathcal{V}_k}\|\boldsymbol{y}_r^{k,i}\|^2$$

$$+ \frac{m(n-m)}{n^2(n-1)}\sum_{k\neq l}\sum_{i\in\mathcal{V}_k}\sum_{j\in\mathcal{V}_l}\bigg(\frac{1}{2}\|\boldsymbol{y}_r^{i,k}\|^2 + \frac{1}{2}\|\boldsymbol{y}_r^{j,l}\|^2\bigg)\bigg], \tag{88}$$

where the third equation holds by the probability of random sampling with replacement, i.e., $\mathbb{P}\{i\in\mathcal{S}_r^k\} = \frac{m}{n}, \mathbb{P}\{i\neq j\in\mathcal{S}_r^k\} = \frac{m(m-1)}{n(n-1)}, \mathbb{P}\{i\in\mathcal{S}_r^k, j\in\mathcal{S}_l^t|k\neq l\in[K]\} = \frac{m^2}{n^2}$. The forth equation holds by $\langle\boldsymbol{a},\boldsymbol{b}\rangle = \frac{1}{2}[\|\boldsymbol{a}\|^2 + \|\boldsymbol{b}\|^2 - \|\boldsymbol{a}-\boldsymbol{b}\|^2], \frac{1}{2}\sum_{i\neq j}\|\boldsymbol{a}_i - \boldsymbol{a}_j\|^2 = \sum_{i=1}^{n}n\|\boldsymbol{a}_i\|^2 - \|\sum_{i=1}^{n}\boldsymbol{a}_i\|^2$, and $\|\sum_{k=1}^{K}\sum_{i\in\mathcal{V}_k}\boldsymbol{y}_r^{k,i}\|^2 = \sum_{k=1}^{K}\|\sum_{i\in\mathcal{V}_k}\boldsymbol{y}_r^{k,i}\|^2 + \sum_{k\neq l}\sum_{i\in\mathcal{V}_k}\sum_{j\in\mathcal{V}_l}\langle\boldsymbol{y}_r^{k,i}\boldsymbol{y}_r^{l,j}\rangle$. The last inequality holds by $\boldsymbol{a}'\boldsymbol{b} \leq \frac{1}{2}\|\boldsymbol{a}\|^2 + \frac{1}{2}\|\boldsymbol{b}\|^2$. Re-organize the last item,

$$\frac{m(n-m)}{n^2(n-1)}\sum_{k\neq l}\sum_{i\in\mathcal{V}_k}\sum_{j\in\mathcal{V}_l}\bigg(\frac{1}{2}\|\boldsymbol{y}_r^{i,k}\|^2 + \frac{1}{2}\|\boldsymbol{y}_r^{j,l}\|^2\bigg) = \frac{m(n-m)}{n^2(n-1)}(K-1)n\sum_{k=1}^{K}\sum_{i\in\mathcal{V}_k}\|\boldsymbol{y}_r^{k,i}\|^2, \tag{89}$$

then we have

$$\mathbb{E}\bigg[\bigg\|\sum_{k\in[K]}\sum_{i=1}^{n}\mathbb{I}\{i\in\mathcal{S}_r^k\}\boldsymbol{y}_r^{k,i}\bigg\|^2\bigg]$$

$$\leq \mathbb{E}\bigg[\frac{m(m-1)}{n(n-1)}\bigg\|\sum_{k=1}^{K}\sum_{i\in\mathcal{V}_k}\boldsymbol{y}_r^{k,i}\bigg\|^2 + \frac{m(n-m)}{n(n-1)}\sum_{k=1}^{K}\sum_{i\in\mathcal{V}_k}\|\boldsymbol{y}_r^{k,i}\|^2$$

$$+ \frac{m(n-m)}{n^2(n-1)}(K-1)n\sum_{k=1}^{K}\sum_{i\in\mathcal{V}_k}\|\boldsymbol{y}_r^i\|^2\bigg]$$

$$= \mathbb{E}\bigg[\frac{m(m-1)}{n(n-1)}\bigg\|\sum_{k=1}^{K}\sum_{i\in\mathcal{V}_k}\boldsymbol{y}_r^{k,i}\bigg\|^2 + \frac{Km(n-m)}{n(n-1)}\sum_{k=1}^{K}\sum_{i\in\mathcal{V}_k}\|\boldsymbol{y}_r^{k,i}\|^2\bigg]. \tag{90}$$

This concludes the proof. $\qquad\square$

**Lemma F.7** (Lemma for momentum term in the update rule). *The first order momentum terms $\boldsymbol{m}_r$ in Algorithm 1 hold the following relationship w.r.t. model difference $\Delta_r$:*

$$\sum_{r=1}^{R} \mathbb{E}[\|\boldsymbol{m}_r\|^2] \leq \sum_{r=1}^{R} \mathbb{E}[\|\Delta_r\|^2]. \tag{91}$$

*Proof.* By the updating rule, we have

$$
\begin{aligned}
\mathbb{E}[\|\boldsymbol{m}_r\|^2] &= \mathbb{E}\left[\left\|(1-\beta_1)\sum_{u=1}^{r}\beta_1^{r-u}\Delta_u\right\|^2\right] \\
&\leq (1-\beta_1)^2 \sum_{i=1}^{d} \mathbb{E}\left[\left(\sum_{u=1}^{r}\beta_1^{r-u}\Delta_{u,i}\right)^2\right] \\
&\leq (1-\beta_1)^2 \sum_{i=1}^{d} \mathbb{E}\left[\left(\sum_{u=1}^{r}\beta_1^{r-u}\right)\left(\sum_{u=1}^{r}\beta_1^{r-u}(\Delta_{u,i})^2\right)\right] \\
&\leq (1-\beta_1)\sum_{u=1}^{r}\beta_1^{r-u}\mathbb{E}[\|\Delta_u\|^2],
\end{aligned} \tag{92}
$$

summing over $t = 1, ..., T$ yields

$$
\begin{aligned}
\sum_{r=1}^{R}\mathbb{E}[\|\boldsymbol{m}_r\|^2] &= (1-\beta_1)\sum_{r=1}^{R}\sum_{u=1}^{r}\beta_1^{r-u}\mathbb{E}[\|\Delta_u\|^2] \\
&= (1-\beta_1)\sum_{u=1}^{R}\sum_{r=u}^{R}\beta_1^{r-u}\mathbb{E}[\|\Delta_u\|^2] \\
&\leq (1-\beta_1)\sum_{u=1}^{R}\frac{1}{1-\beta_1}\mathbb{E}[\|\Delta_u\|^2] \\
&= \sum_{u=1}^{R}\mathbb{E}[\|\Delta_u\|^2].
\end{aligned} \tag{93}
$$

This concludes the proof. $\qquad\square$

**Lemma F.8.** *Under Assumptions 4.3, for AFGA and CAFGA, we have $\|\nabla f(\boldsymbol{x})\| \leq G$, $\|\Delta_r\| \leq \frac{m\eta_l \mathcal{I} G}{n}$, $\|\boldsymbol{m}_r\| \leq \frac{m\eta_l \mathcal{I} G}{n}$, $\|\boldsymbol{v}_r\| \leq \frac{m^2\eta_l^2 \mathcal{I}^2 G^2}{n^2}$ and $\|\widehat{\boldsymbol{v}}_r\| \leq \frac{m^2\eta_l^2 \mathcal{I}^2 G^2}{n^2}$.*

*Proof.* Since $f$ has $G$-bounded stochastic gradients, for any $\boldsymbol{x}$ and $\xi$, we have $\|\nabla f(\boldsymbol{x}, \xi)\| \leq G$, we have

$$\|\nabla f(\boldsymbol{x})\| = \|\mathbb{E}_\xi \nabla f(\boldsymbol{x}, \xi)\| \leq \mathbb{E}_\xi \|\nabla f(\boldsymbol{x}, \xi)\| \leq G.$$

For AFGA and CAFGA, the model difference $\bar{\Delta}_r^k$ on cluster $k$ satisfies,

$$\bar{\Delta}_r^k = \bar{\boldsymbol{x}}_{r,\mathcal{I}}^k - \boldsymbol{x}_r = -\eta_l \sum_{t=0}^{\mathcal{I}-1} \bar{\boldsymbol{g}}_{r,t}^k = -\eta_l \sum_{t=0}^{\mathcal{I}-1}\frac{1}{n}\sum_{i\in\mathcal{V}_k}\bar{\boldsymbol{g}}_{r,t}^i = -\eta_l\sum_{t=0}^{\mathcal{I}-1}\frac{1}{n}\sum_{i\in\mathcal{S}_{r,t}^k}\boldsymbol{g}_{r,t}^i,$$

therefore,

$$\|\bar{\Delta}_r^k\| = \left\|\eta_l\sum_{t=0}^{\mathcal{I}-1}\bar{\boldsymbol{g}}_{r,t}^k\right\| = \left\|\eta_l\sum_{t=0}^{\mathcal{I}-1}\frac{1}{n}\sum_{i\in\mathcal{S}_{r,t}^k}\boldsymbol{g}_{r,t}^i\right\| \leq \frac{m\eta_l\mathcal{I}G}{n},$$

for the global model difference $\Delta_r$,

$$\|\Delta_r\| = \left\|\frac{1}{K}\sum_{k\in[K]}\bar{\Delta}_r^k\right\| \le \frac{m\eta_l \mathcal{I}G}{n}.$$

Thus we can obtain the bound for momentum $\boldsymbol{m}_r$ and variance $\boldsymbol{v}_r$,

$$\|\boldsymbol{m}_r\| = \left\|(1-\beta_1)\sum_{s=1}^{r}\beta_1^{r-s}\Delta_s\right\| \le \frac{m\eta_l \mathcal{I}G}{n}, \quad \|\boldsymbol{v}_r\| = \left\|(1-\beta_2)\sum_{s=1}^{r}\beta_2^{r-s}\Delta_s^2\right\| \le \frac{m^2\eta_l^2 \mathcal{I}^2 G^2}{n^2}.$$

By the updating rule of $\widehat{\boldsymbol{v}}_r$, there exists a $j \in [r]$ such that $\widehat{\boldsymbol{v}}_r = \boldsymbol{v}_j$. Then

$$\|\widehat{\boldsymbol{v}}_r\| \le \frac{m^2\eta_l^2 \mathcal{I}^2 G^2}{n^2}. \tag{94}$$

This concludes the proof. $\qquad\square$

**Lemma F.9.** *For the variance difference sequence* $\widehat{\boldsymbol{V}}_{r-1}^{-1/2} - \widehat{\boldsymbol{V}}_r^{-1/2}$, *we have*

$$\sum_{r=1}^{R}\left\|\widehat{\boldsymbol{V}}_{r-1}^{-1/2} - \widehat{\boldsymbol{V}}_r^{-1/2}\right\|_1 \le \frac{d}{\sqrt{\epsilon}}, \quad \sum_{r=1}^{R}\left\|\widehat{\boldsymbol{V}}_{r-1}^{-1/2} - \widehat{\boldsymbol{V}}_r^{-1/2}\right\|^2 \le \frac{d}{\epsilon} \tag{95}$$

*Proof.* The proof of Lemma F.9 is exactly the same as the proof of Lemma C.2 in (Wang et al., 2022b). $\quad\square$

