# OpenReview forum: "On the Data Heterogeneity in Adaptive Federated Learning"
_TMLR — Accepted by TMLR_

### Review · Reviewer_DE59 · 2024-04-26

**Summary Of Contributions:**

This paper introduces AFGA, an enhanced version of the FedAMSGrad algorithm, which incorporates adaptive federated learning techniques. AFGA integrates two novel components: client re-sampling and gossip communications. Client re-sampling involves selecting a new subset of clients for gradient computation during each local iteration, whereas gossip communications enable clients to exchange weights with their connected peers. To mitigate the additional communication overhead introduced by gossip communications, the authors propose a variant, clustered-clients AFGA, where clients are grouped into clusters to restrict re-sampling and communications within each cluster.

**Audience:**

Yes

**Claims And Evidence:**

Yes

**Requested Changes:**

1. The rationale behind client re-sampling is unclear, given the inherent nature of partial participation in FL due to varying client availability. The method assumes control over client selection, which may not reflect practical scenarios where available clients can fluctuate unpredictably.

2. Could you clarify the relationship between $S_{r,t}$ and $S_r$? It's important to understand how these selections differ across iterations and rounds.

3. What are the variance or deviation measures for the mean values presented in Table 1? This information would help assess the statistical significance of the reported results.

4. The term R# (representing the total number of rounds to achieve a specified accuracy level) used in Tables 1 and 2 is not defined until the caption of Table 5. This should be clarified earlier in the document for better reader comprehension.

5. Regarding the clustered-clients AFGA (CAFGA), is the clustering of clients into equal-sized groups based on randomness, or is there a specific metric assessing client similarity that influences cluster composition? Understanding this clustering basis is essential for evaluating the algorithm's practical applicability and performance efficiency.

**Strengths And Weaknesses:**

The strengths of this paper include a clear presentation of the proposed algorithms and comparative analysis with existing federated learning (FL) algorithms, highlighting improvements in accuracy and convergence speed.

However, the paper falls short in its discussion of data heterogeneity impacts. It lacks a thorough analysis of how the proposed algorithm performs under varying degrees of data heterogeneity, which is crucial for validating the effectiveness of any FL algorithm.

---

> ### Author Response · Authors · 2024-05-18
> **Response to Reviewer DE59**
>
> Thank you for your constructive comments and suggestions.
> ### **Q1** Lacks of discussion of data heterogeneity impacts
> We apologize for the inadequate discussion on the impact of data heterogeneity in our paper. We have included additional discussion on how data heterogeneity affects both theoretical analysis and empirical results in the revision.
>
> In fact, we have already provided some detailed explanations in Remark 3.1. In brief, the design of frequent re-sampling and gossip communications makes AFGA effectively reduce the inconsistency between local clients, thus helping reduce the impact of data heterogeneity in adaptive FL.
>
> From the perspective of theoretical analysis, the impact of data heterogeneity is reflected in the variance $\sigma_g$ within the convergence rate. The proposed AFGA eliminated the $O\big(\frac{\sqrt{I}}{\sqrt{R M}} \sigma_g^2\big)$ term in the rate of FedAMSGrad, thus improving the convergence rate in partial participation settings.
>
>
> For the experiments, we have conducted ablation studies on how various degrees of data heterogeneity affect the proposed method in Section 6.3. We observe that a more balanced data distribution attains faster convergence and higher accuracy, data heterogeneity among clients continues to affect both the convergence and generalization performance of our proposed AFGA and CAFGA methods.
>
> ### **Q2** About the rationale behind client re-sampling
> The rationale behind client re-sampling in each local iteration is to enhance sample frequency and mitigate the impact of training on certain local data distributions. It allows more clients to be included and participate in local training, which results in training a global model with a more balanced data distribution than without re-sampling. This helps reduce the impact of data heterogeneity.
>
>
> Note that the client re-sampling strategy does not make the proposed method impractical. The current random sampling design is for the ease of theoretical analysis. In practice, the server can simply sample from available clients in each iteration. This is essentially the same as the traditional FL methods such as FedAvg: for analysis, we randomly sample from all clients but in practice, we just sample from available ones when performing partial participation FL. Our experiments show that AFGA is compatible with low client participation ratios, suggesting that this client re-sampling approach is feasible in practical scenarios. We have added more discussions in the revision.
>
>
> ### **Q3** About the relationship between $S_{r,t}$ and $S_r$
> We are sorry for the confusion. In our paper, $t$ indicates the local iterations and $r$ indicates the global communication round. $S_r$ is initialized at the beginning of each global round $r$, the server broadcasts the latest global model to clients in $S_r$. $S_{r,t}$ is initialized for each local iteration $t$, and clients in $S_{r,t}$ conduct local training with gradient computation. We have already provided some detailed explanations in the part of client re-sampling in section 3.2.
>
> ### **Q4** About the variance or deviation of the reported results
> We have conducted experiments with 3 different random seeds and we have updated Tables 1,2,3, and 5 with the average accuracy and the standard derivation in our revision.
>
> ### **Q5** About \#R
> Thanks for pointing this out, we have clarified the notation of \#R when introducing the implementation overview in the revision.
>
> ### **Q6** About the clustering basis
> Thanks for the suggestion, we have clarified and addressed this in the revision. For CAFGA, our basic intuition is clients can be grouped based on locations and network capabilities, in which gossip communications are less expensive than communicating with the
> whole network. In experiments, clustering is grouping the clients into equal-sized clusters based on randomness.
>
> Moreover, we think that the idea of client re-sampling and gossip communication is orthogonal to the clustered-based FL methods which utilize the client's similarity and clustered them into groups. We will leave this interesting extension of AFGA as future work.

---

### Review · Reviewer_Zydi · 2024-04-29

**Summary Of Contributions:**

In this work, the authors propose a new method for adaptive optimization in federated learning. The method is based on peer-to-peer gossip communications between local clients. With this additional step of local averaging, the authors show that the proposed method outperforms prior baselines for adaptive optimization in FL.

**Audience:**

Yes

**Claims And Evidence:**

No

**Requested Changes:**

- Explanation of 'R#' is missing.
- How is hyperparameter, for example $W$ is tuned?
- Clean up convergence bound, as I mentioned in the weakness part above.

**Strengths And Weaknesses:**

Strengths:
- The work comes with provable convergence guarantees.
- The paper is generally well-written and easy to understand.
- Empirical results seems solid.

Weaknesses:
- Having an entire gossip matrix that is shared across clients is a very strong assumption and does not seem practical.
- From the appendix it seems that setting $\rho=0$ defaults the best performance? Or am I missing anything? Relation between $\rho$ and utility is missing.
- The convergence bound seems worse than the author claimed. Equation 3 is only dominated by $O\left(\frac{1}{\sqrt{R\mathcal{I}M}}\right)$ if $R>\sqrt{R\mathcal{I}M}$. Otherwise it is at least $O\left(\frac{1}{\sqrt{R\mathcal{I}M}}\right)+O\left(\frac{1}{R}\right)$. And this is given $\rho=0$. Also, I don't think omitting parameters such as $M,N,L$ is good in order to understand the bound. I suggest the authors to clean up the bound so that it only contains $R, L, N$ and subsampling rate $q=M/N$ or $M$ itself.
- Explanation of 'R#' is missing. Is that the number of rounds? How is this selected as this number is different for different methods.
- Table 5 adjust the spacing.

---

> ### Author Response · Authors · 2024-05-18
> **Response to Reviewer Zydi**
>
> Thank you for your constructive comments and suggestions.
> ### **Q1** About sharing the entire gossip matrix across clients
> We are sorry for the confusion. In fact, clients do not need to know the entire mixing matrix $W$. We use the matrix form mainly for the ease of theoretical analysis. In practice, client $i$ only needs to know the weights of $W_{ij}$ so that it can communicate with its neighbors to aggregate the local model. We have made this clear in the revision.
>
> ### **Q2** About the relation between $\rho$ and utility
> Your observation is correct. Remark 4.10 concludes the dependency on $\rho$ from the perspective of theoretical convergence. In a nutshell, $\rho$ appears in the $O(\frac{1}{R})$-order terms in Equation (3). If $\rho=0$, the  $O(\frac{1}{R})$-order terms would reduce to $O(\frac{G^2}{R})$-order terms. This indicates that the variance $\sigma^2$ and $\sigma_g^2$ terms could be eliminated in $O(\frac{1}{R})$-order terms, leading to better convergence results. This is also quite intuitive since $\rho=0$ corresponds to the fully connected case where each client exchanges their model with all other clients.
>
>
> ### **Q3** About the dominant term of the convergence bound
> We are sorry for the confusion but we didn't overclaim on the convergence bound. What we claimed in Remark 4.7 is that the convergence rate of $O(\frac{1}{\sqrt{RIM}})$ is achieved **when there are sufficient global communication rounds $R$ with $R\geq IM$**. This is exactly what you mentioned in the review. Note that in terms of convergence result, people generally refer to the case where $R$ is sufficiently large as it reflects the long-term trend.
>
>
> ### **Q4** About cleaning up the convergence bound
> Thanks for the kind suggestion. We have simplified the bound by omitting terms that do not significantly contribute to our theoretical analysis. However, it is essential to retain terms related to $G$, $\sigma$, and $\sigma_g$ within the convergence bound. Omitting these could lead to a misinterpretation of the results. For instance, consider the term $O(\frac{1}{R}[G^2 + L^2 \rho^2 \sigma_g^2 + \frac{\rho^2 \sigma^2}{I}])$. Removing the $G$-related term results in $O(\frac{1}{R}[L^2 \rho^2 \sigma_g^2 + \frac{\rho^2 \sigma^2}{I}])$. If $\rho=0$, $O(\frac{1}{R}[L^2 \rho^2 \sigma_g^2 + \frac{\rho^2 \sigma^2}{I}])$ reduces to zero, suggesting erroneously that the convergence bound reduces an $O(\frac{1}{R})$-order term, which is not consistent with the actual convergence rate.
>
> For a clearer overview of the convergence rate w.r.t. $R, I, M$, we have introduced a new Corollary 4.7 that presents a cleaned-up convergence rate. Additionally, we have revised Remark 4.8 accordingly.
>
> ### **Q5** About \#R and the selection in experiments
> \#R is the number of global communication rounds, we refer to \#R or \#Rounds in tables and figures. We have added an explanation in the revision. In our experiments, we keep this number the same across different baselines. we report 500 rounds for the CIFAR-10 and the Shakespeare datasets and 600 rounds for the CIFAR-100 dataset. For each dataset and setting, the total rounds of training \#R are fixed across all baseline methods to ensure a fair comparison.
>
> ### **Q6** About the formatting for Table 5
> Thanks for pointing this out. We have fixed the formatting issue in the revision.

---

### Review · Reviewer_i2Nk · 2024-05-05

**Summary Of Contributions:**

This paper introduces a novel adaptive federated learning framework, Adaptive Federated learning with Local Gossip Averaging (AFGA), aimed at addressing the data heterogeneity problem prevalent in adaptive federated learning. The proposed framework incorporates a client re-sampling strategy and decentralized peer-to-peer gossip communications to mitigate the dissimilarity in data distribution across clients, thus enhancing model convergence without requiring additional communication or gradient computation costs. The paper also includes a theoretical discussion and emprical evidence.

**Audience:**

Yes

**Broader Impact Concerns:**

none noted.

**Claims And Evidence:**

Yes

**Requested Changes:**

1. The paper reports "the average of the last 5 global rounds to represent final accuracy", please be more explicit about the rationales or the customs behind it.

2. Given the fact that the experimental setting might be too specific (due to the nature of Federated learning), it is a good idea to include error bar (std) in the reported tables.

3. The theoretical results relies on several assumption, in particular, please offer more dicussions on how realistic the Assumption 4.5 is, in theory and experimental (through simulations). Otherwise, it will be hard to evaluate the theoretical contributions. In addition, it seems graph G is never mentioned ahead of Assumption 4.5.

4. minor: Table 5 formatting issues.

**Strengths And Weaknesses:**

Strengths:
+ the paper has a fairly comprehensive discussion existing methods
+ the paper has most of the basic components including the theoretical discussion and ablation studies.

Weakness:
- The theoretical discussion introduces several new assumptions, some assumptions might not be so realistic.
- It's unclear why the authors choose ConvMixer as the main experimental architecture, instead of more population choices (typically used in Federated Learning)

---

> ### Author Response · Authors · 2024-05-18
> **Response to Reviewer i2Nk**
>
> Thank you for your constructive comments and suggestions.
> ### **Q1** About how realistic Assumption 4.5 is
>
> We are sorry for the confusion. Assumption 4.5 is realistic and can be easily verified both theoretically and experimentally.
> * For a doubly stochastic matrix $W$, i.e., $W \in [0,1]^{n\times n}$, $W \mathbf{1} = \mathbf{1}$, $\mathbf{1}^\top W= \mathbf{1}^\top$ and $\text{null}(\mathbf{I} - W) = \text{span}(\mathbf{1})$, it is naturally hold that  $||W - \frac{1}{n}\mathbf{1}\mathbf{1}^\top ||_2 \leq 1$. In Assumption 4.5, we assume there exists $\rho \in [0,1)$ such that the mixing matrix satisfies $||W - \frac{1}{n}\mathbf{1}\mathbf{1}^\top ||_2 \leq \rho$, which is easy to satisfy. In fact, this $\rho$ describes the connectivity of the clients: a smaller spectral gap $\rho$ indicates denser connectivity between clients. Specifically, $\rho = 0$ indicates that all elements in the matrix $W$ are $\frac{1}{n}$, and this means that each client would be connected and communicated with other clients in the network with a mixing weight of $\frac{1}{n}$. $\rho \to 1$ means the matrix $W$ tends to be elements with either 0 or 1, corresponding to a graph that is nearly disconnected. We assume that there exists $\rho \in [0,1)$ to satisfy $||W - \frac{1}{n}\mathbf{1}\mathbf{1}^\top ||_2 \leq \rho$ since our proposed method is contributed by gossip communications between clients.
> * This assumption could also be easily verified in practice. For example, the ring topology adopted in our experiments by default has $\rho=0.995$ for 50 clients, and $\rho=0.873$ for 10 clients. We also moved the ablation study for the spectral gap $\rho$ to Section 6.3, where it shows the results of using different structures and their corresponding $\rho$.
>
>
> We have added more discussion about Assumption 4.5 in the revision, and we have added some theoretical discussion in Appendix E as well. Hope this will resolve your concern about the realisticality of Assumption 4.5.
>
>
> ### **Q2** About choosing ConvMixer as the main experimental architecture
>
> The ConvMixer-256-8 network shares similar ideas to vision transformerto which uses patch embeddings to preserve locality and similarly, and is trained via adaptive gradient methods by default. In this paper, our focus is on data heterogeneity within adaptive federated learning. Therefore, we believe that an architecture originally trained via an adaptive gradient optimization method provides the most accurate representation of how our proposed method enhances performance in the subfield of adaptive FL.
>
> To demonstrate effectiveness across more popular model architectures, we conducted additional experiments with the ResNet-18 model on the CIFAR-10 datasets. These experiments followed a similar setup to our main paper, involving 50 clients with a participation ratio of 0.1. The results presented in the following table demonstrate that the proposed CAFGA outperforms other federated learning baselines, achieving a 0.4% improvement over FedAMSGrad and an enhancement of more than 1% compared to other baselines.
>
> |Method|FedAvg|FedAdam|FedAMSGrad|SCAFFOLD|
> |--|--|--|--|--|
> |Acc. & std. |70.32 $\pm$ 0.44|73.8 $\pm$ 0.58|75.59 $\pm$ 0.73|74.60 $\pm$ 0.67|
> |**Method**|**FedProx**|**FedDyn**|**AFGA**|**CAFGA**|
> |Acc. & std. |70.26 $\pm$ 0.45 | 74.15 $\pm$ 2.23|74.72 $\pm$ 0.32 |**75.96**$\pm$ 0.67|
>
> We have added this experimental result in Appendix C.1 in the revision.
>
>
> ### **Q3** About the average of the last 5 global rounds to represent final accuracy
>
> In our original submission, to eliminate the effect of stochastic sampling on the final result, we take the average of the accuracy of the last 5 global rounds as the final accuracy.
>
> ### **Q4** Include error bar (std) in the reported tables.
>
> Thanks for the suggestion. We have conducted experiments with 3 different random seeds and we have updated tables with the average accuracy and the standard derivation in the revision of our main paper.
>
> ### **Q5** About the graph $G$ and the formatting for Table 5
>
> Thanks for pointing this out. We have added an introduction to the concept of graph $G$ in the method. We have fixed the formatting issue in the revision.

---

### Author Response · Authors · 2024-05-18
**Summary of Paper Revision**

We thank the reviewers for the constructive feedback. We have revised our paper accordingly. We summarize the major changes in the revision as follows:

* We reported the test accuracy with standard derivation for Tables 1, 2, 3, and 5, suggested by *Reviewer i2Nk* and *Reviewer DE59*.
* We fixed the formatting issues about Table 5, and we clarified several necessary notations, pointed out by *Reviewer i2Nk*, *Zydi*, and *DE59*.
* We added discussions about Assumption 4.5, and we conducted additional experiments using the ResNet-18 model in Appendix C.1, as suggested by *Reviewer i2Nk*.
* We reorganized the convergence analysis in Corollary 4.7 and Remark 4.8, as suggested by *Reviewer Zydi*.
* We added discussions about the client re-sampling in section 3.2, we discussed the impact of data heterogeneity from a theoretical perspective in Remark 4.9, and from an empirical perspective in Section 6.3, as suggested by *Reviewer DE59*.

---

### Decision · Action_Editor_CT9N · 2024-07-10

**Recommendation:** Accept as is

**Comment:**

This paper studies the convergence behavior of adaptive methods in federated optimization. It is stated that the convergence of earlier proposed methods, such as FedAMSGrad, can be slowed down in the presence of data heterogeneity. To alleviate this, two changes are proposed to FedAMSGrad: a subsampling strategy among the active clients and global gossip averaging to improve mixing. Experiments have shown that the proposed method can enhance convergence.

The reviewers found that the contribution is well within the scope of TMLR. Additionally, the authors have thoroughly addressed all comments and suggestions provided during the review process in their revisions.

Although the convergence theorems have a weaker dependence on data heterogeneity measures than similar statements for the referenced baselines, the paper does not discuss the tightness of the theoretical results. This is a minor limitation of the work.

**Audience:**

Adaptive methods and federated learning are two topics that interest the TMLR audience. The techniques proposed in this paper to make adaptive methods more amenable to federated optimization settings can be particularly interesting to parts of the TMLR audience.

**Claims And Evidence:**

The reviewers found that the claims in the paper are supported by clear evidence. In particular, the assumptions for the theoretical analysis are clearly stated, and the results are precise and appear correct. The setting for the numerical experiments is well described, and the results support the claims, although some reviewers would have wished to see a more thorough exploration of the impact of data heterogeneity on the convergence.